# Structural insights into the cross-exon to cross-intron spliceosome switch

Zhenwei Zhang[1,2,7], Vinay Kumar[3,7], Olexandr Dybkov[3,4], Cindy L. Will[3], Jiayun Zhong[2], Sebastian E. J. Ludwig[3,6], Henning Urlaub[4,5], Berthold Kastner[3], Holger Stark[1✉] & Reinhard Lührmann[3✉]

Early spliceosome assembly can occur through an intron-defined pathway, whereby U1 and U2 small nuclear ribonucleoprotein particles (snRNPs) assemble across the intron[1]. Alternatively, it can occur through an exon-defined pathway[2–5], whereby U2 binds the branch site located upstream of the defined exon and U1 snRNP interacts with the 5' splice site located directly downstream of it. The U4/U6.U5 tri-snRNP subsequently binds to produce a cross-intron (CI) or cross-exon (CE) pre-B complex, which is then converted to the spliceosomal B complex[6,7]. Exon definition promotes the splicing of upstream introns[2,8,9] and plays a key part in alternative splicing regulation[10–16]. However, the three-dimensional structure of exon-defined spliceosomal complexes and the molecular mechanism of the conversion from a CE-organized to a CI-organized spliceosome, a pre-requisite for splicing catalysis, remain poorly understood. Here cryo-electron microscopy analyses of human CE pre-B complex and B-like complexes reveal extensive structural similarities with their CI counterparts. The results indicate that the CE and CI spliceosome assembly pathways converge already at the pre-B stage. Add-back experiments using purified CE pre-B complexes, coupled with cryo-electron microscopy, elucidate the order of the extensive remodelling events that accompany the formation of B complexes and B-like complexes. The molecular triggers and roles of B-specific proteins in these rearrangements are also identified. We show that CE pre-B complexes can productively bind *in trans* to a U1 snRNP-bound 5' splice site. Together, our studies provide new mechanistic insights into the CE to CI switch during spliceosome assembly and its effect on pre-mRNA splice site pairing at this stage.

We aimed to elucidate the molecular architecture of exon-defined spliceosomal complexes. To that end, we first determined the three-dimensional (3D) structure of a human CE pre-B complex (previously denoted the 37S exon complex) assembled in HeLa nuclear extract on a MINX exon-containing RNA (Fig. 1a) that was previously shown to undergo exon definition[7]. To enhance the stability of the CE pre-B complex, we isolated complexes at low salt conditions, which led to the formation of predominantly pre-B dimers. Affinity-purified complexes contained stoichiometric amounts of the MINX exon RNA and U1, U2, U4, U5 and U6 snRNAs (Extended Data Fig. 1a). They also contained essentially all known snRNPs and U1-related and serine-arginine-rich (SR) proteins (Supplementary Table 1).

## Cryo-EM structure of CE pre-B complexes

Single-particle cryo-electron microscopy (cryo-EM) followed by 3D reconstructions of the purified CE pre-B complexes revealed that one of the major dimer classes comprised two similar units (protomers) aligned in an antiparallel manner, whereby each of the protomers resembles a CI pre-B complex (Fig. 1b and Extended Data Fig. 1b). We next determined the structure of individual CE pre-B protomers at 3.5 Å resolution at their tri-snRNP core (Extended Data Fig. 1c–h). The molecular architecture of the tri-snRNP in the CE pre-B complex was highly similar if not identical to that in human CI pre-B complexes and isolated tri-snRNPs[17–19] (Fig. 1c). For example, in both types of pre-B complexes, PRP8 has an open conformation and the BRR2 helicase domain is located close to the PRP8 reverse transcriptase-like (RT) domain (PRP8[RT]). Moreover, the BRR2 helicase domain has not yet translocated to the PRP8 endonuclease-like (En) domain (PRP8[En]) nor docked with its U4 snRNA substrate (Fig. 1c). Furthermore, U4/U6 stem III and the so-called U4 snRNA quasi-pseudoknot are present (Fig. 1c and Extended Data Fig. 1i–k). In CE pre-B, the tri-snRNP interacts with the U2 snRNP through three main bridges that are also observed in CI pre-B complexes[18,19] (Extended Data Fig. 2a–c). Thus, our structure

[1]Department of Structural Dynamics, Max-Planck-Institute for Multidisciplinary Sciences, Göttingen, Germany. [2]State Key Laboratory of Biotherapy and Department of Rheumatology and Immunology, West China Hospital, Sichuan University, Chengdu, China. [3]Cellular Biochemistry, Max-Planck-Institute for Multidisciplinary Sciences, Göttingen, Germany. [4]Bioanalytical Mass Spectrometry, Max-Planck-Institute for Multidisciplinary Sciences, Göttingen, Germany. [5]Bioanalytics Group, Institute for Clinical Chemistry, University Medical Center Göttingen, Göttingen, Germany. [6]Present address: Vincerx Pharma, Monheim am Rhein, Germany. [7]These authors contributed equally: Zhenwei Zhang, Vinay Kumar. ✉e-mail: hstark1@gwdg.de; reinhard. luehrmann@mpinat.mpg.de

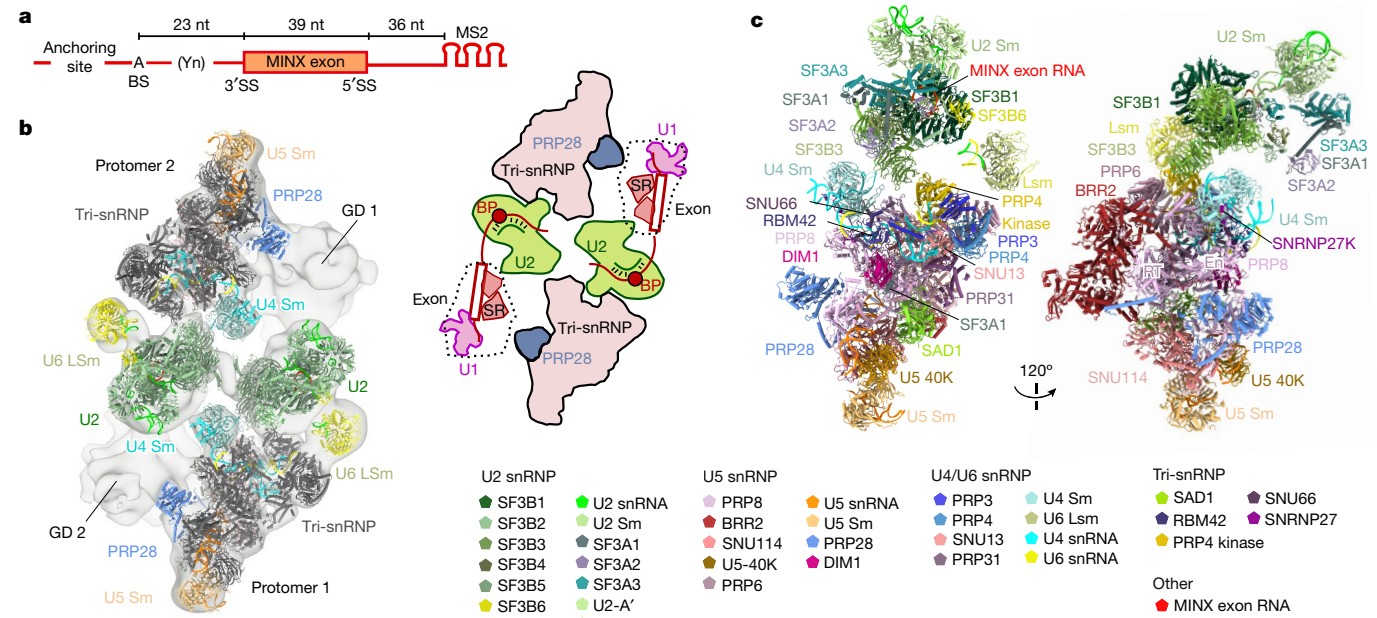

**Fig. 1 | 3D structure of human CE pre-B complexes. a**, Schematic of the MINX exon RNA. Yn, polypyrimidine tract. **b**, Left, fit of two monomeric, CE pre-B molecular models into the EM density of the class 1, pre-B dimer. Right, cartoon of the structural organization of the CE pre-B complex dimer. BP, branch point. **c**, Top, two different views of the molecular architecture of the human CE pre-B complex. Bottom, summary of all modelled proteins and RNAs with colour code.

shows that tri-snRNP recruitment is achieved through similar molecular interactions. Moreover, its 3D structure and orientation relative to U2 are highly similar in both exon-defined and intron-defined pre-B complexes.

We previously showed that the U1 snRNA is base-paired to the 5′ splice site (5′ss) of the MINX exon RNA in CE pre-B complexes[7]. In the latter, a poorly resolved, globular density (GD) is connected on one side of the U2-SF3B1 HEAT domain (SF3B1$^{HEAT}$) by a thin density element (Extended Data Fig. 2d) that protrudes from SF3B1$^{HEAT}$ repeats H7–H9. This result indicates that it contains intron or exon nucleotides near the MINX 3′ ss. Consequently, the MINX exon together with U1 snRNP and exon-binding proteins are located in the adjacent GD (Extended Data Fig. 2d–f). Owing to the poor resolution of the CE pre-B structure in this region, U1 could not be clearly localized, and the nature of the molecular bridge linking U2 at the branch site (BS) to U1 snRNP at the downstream 5′ss could not be discerned by cryo-EM. However, protein crosslinking coupled with mass spectrometry (Extended Data Fig. 2g and Supplementary Table 2) indicated that U1 and U1-related proteins communicate indirectly with U2 components, mainly through SR proteins that interact with the U2AF1 and U2AF2 proteins and SF3B1 that are located near the 3′ end of the intron. This result is consistent with the idea that U2 and U1 do not directly interact with one another. Taken together, the molecular architecture of the CE pre-B complex indicates that no major rearrangements in the pre-B complex are required for the tri-snRNP to subsequently interact across an intron with a U1 snRNP bound to an upstream 5′ss during the switch from a CE to CI complex.

## 3D structure of the B-like complex

CI pre-B complexes are converted to CI B complexes following PRP28 helicase-mediated handover of the 5′ss from U1 to the U6 snRNA, thereby forming the U6/5′ss helix[20,21]. This helix is present in an extended form (that is, extended U6/5′ss helix) in CI B complexes. This leads to multiple, major structural rearrangements in the CI pre-B complex[18,19,22,23] and the recruitment of so-called B-specific proteins (including, SMU1, RED, FBP21, SNU23, MFAP1, PRP38A and UBL5)[6]. We previously showed that addition of an excess of a 5′ss-containing RNA oligonucleotide (5′ss oligo) to CE pre-B complexes in the presence of nuclear extract and ATP bypasses the requirement for PRP28 (refs. 7,24). The CE pre-B complex is converted into a CE B-like complex that undergoes structural rearrangements that are currently poorly characterized[7,24]. This simplified system was therefore proposed to mimic the interaction of a bona fide upstream 5′ss with the tri-snRNP during the CE to CI transition. We sought to determine whether CE B-like complexes undergo structural rearrangements similar to those that occur during the conversion of a CI pre-B complex to a CI B complex. We affinity-purified CE B-like complexes (generated as described above) and subsequently determined their cryo-EM structure. Single-particle cryo-EM of the affinity-purified CE B-like complexes revealed dimers comprising two B-like complexes aligned in a parallel manner, similar to the overall organization of human CI B dimers isolated under low salt conditions[23] (Extended Data Fig. 3a–i).

We subsequently determined the structure of the B-like protomers at 3.1 Å resolution in their tri-snRNP core (Fig. 2a). In the B-like complex, G$^{+1}$ and U$^{+2}$ of the 5′ss of the exogenously added RNA oligonucleotide are exclusively recognized by DIM1 and PRP8 in the same manner as the 5′ss of the pre-mRNA in CI B complexes[19,22,23] (Fig. 2b–d and Extended Data Fig. 3j). Nucleotides downstream of U$^{+2}$ base pair with nucleotides at/or near the U6 ACAGA box, forming a U6/5′ss helix, similar to the situation in CI B complexes (Fig. 2b,e and Extended Data Fig. 3k). Notably, a second copy of the 5′ss oligo (denoted oligo 2) also binds, which extends the U6/5′ss helix through additional non-canonical base pairs and base-stacking interactions. This structure therefore mimics the extended U6/5′ss found in CI B complexes (Fig. 2e,f). The zinc finger (ZnF) regions of the B-specific proteins FBP21 and SNU23 bind side by side to the backbone of 5′ss oligo 1 and 5′ss oligo 2, respectively, stabilizing their base-pairing interactions with the U6 snRNA in the B-like complex, analogous to their spatial organization and function in the human CI B complex (Fig. 2f). Indeed, nearly all of the B-specific proteins localized in the CE B-like complex are spatially organized in an analogous manner to that of the human CI B complex[23] (Extended Data Fig. 4a–c).

Structural comparisons of the CE B-like complex with the CE pre-B complex revealed that the majority of tri-snRNP remodelling events that

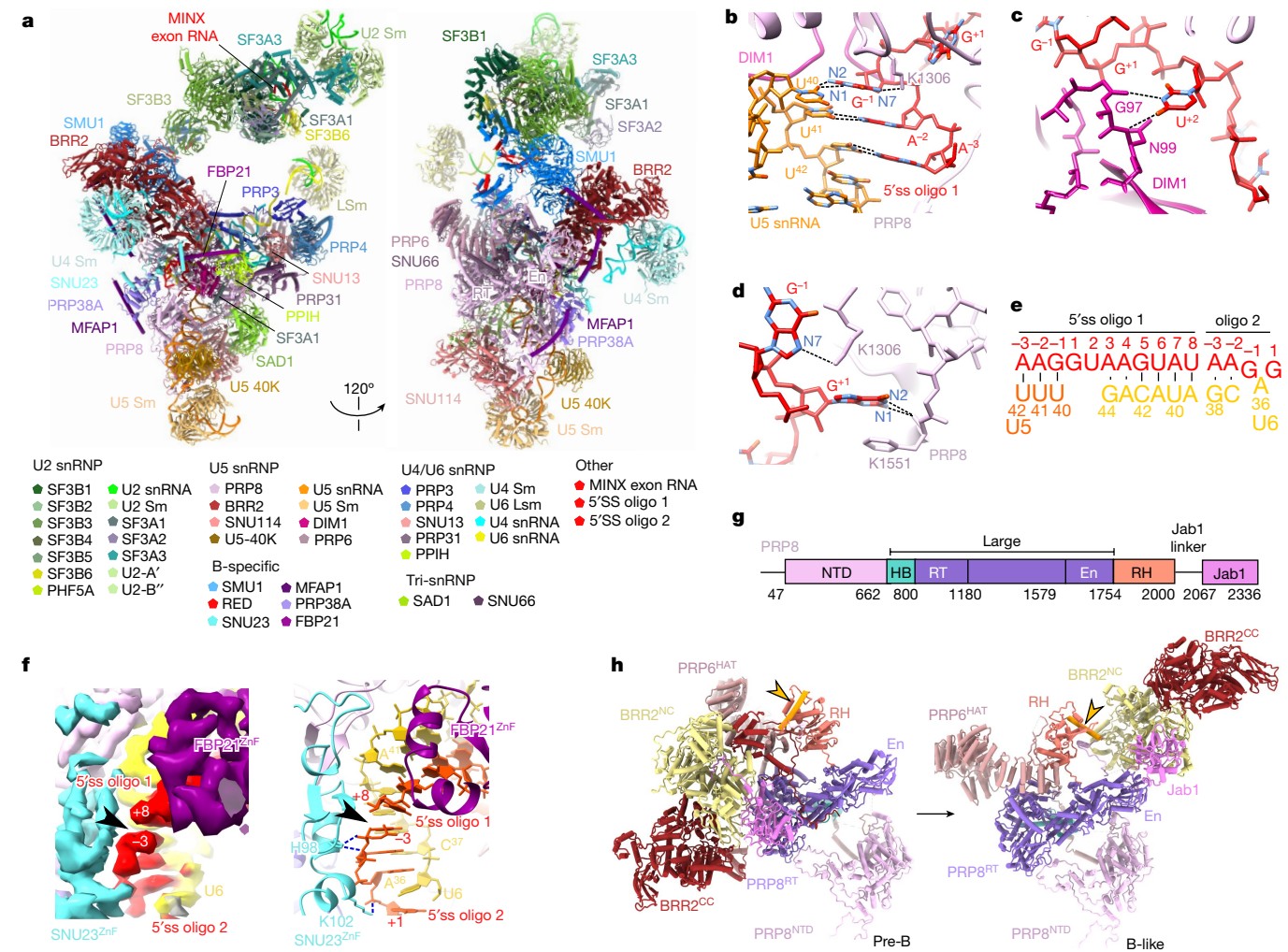

**Fig. 2 | 3D structure of human CE B-like complexes. a**, Top, two different views of the molecular architecture of the human CE B-like complex. Bottom, summary of all modelled proteins and RNAs with colour code. **b–d**, Interactions of the 5′ss oligo with U5 snRNA loop 1 nucleotides and residues of PRP8 and DIM1. **e**, Schematic of the base-pairing interactions of the two copies of the 5′ss oligo that are bound in the B-like complex. 5′ss oligo 2 stacks through its 5′ terminal $A^{-3}$ on the 3′ terminal $U^{+8}$ of the first 5′ss oligo. **f**, The ZnF domains of FBP21 and SNU23 stabilize the base-pairing interactions of 5′ss oligo 1 and 5′ss oligo 2, respectively, with U6 and U5 snRNA. Left, coloured EM density. Right, molecular model. Arrowhead indicates missing EM density, which confirms that two distinct 5′ss oligos are bound. **g**, Schematic of the domain organization of PRP8. HB, helical bundle. **h**, Repositioning of the PRP8 RT/En and RH domains and BRR2 and $PRP6^{HAT}$ during the conversion of the CE pre-B complex (left) into a B-like complex (right). The 3D structures are aligned through $PRP8^{NTD}$.

occur during the CI pre-B to B complex transition also occur after addition of an excess of the 5′ss oligo to the splicing reaction. For example, in the B-like complex, PRP8 adopts a half-closed conformation and the BRR2 helicase domain is translocated to $PRP8^{En}$ (Fig. 2g,h). In addition, the BRR2 amino-terminal helicase cassette ($BRR2^{NC}$) is in an active conformation and is now bound to its U4 snRNA substrate. Moreover, essentially all of the remodelling events that have been described for the U4/U6 proteins during the formation of the CI B complex also occur in the B-like complex. These include, among others, the repositioning of the $PRP6^{HAT}$ domain ($PRP6^{HAT}$) and PRP31 (Extended Data Fig. 4d,e), and the disruption of U4/U6 stem III and the U4 snRNA quasi-pseudoknot (Extended Data Fig. 3k). Finally, the molecular architecture of the U2 snRNP and its interfaces with the tri-snRNP are also similar between B-like and CI B complexes[19,22] (Fig. 2a). Taken together, our data reveal that the tri-snRNP has nearly the same structure in human B-like and B complexes. Thus, addition of an excess of a 5′ss oligo in the presence of nuclear extract and ATP triggers tri-snRNP remodelling events highly similar to those observed during the PRP28-mediated transformation of a CI pre-B complex into a B complex. These results indicate that CE

pre-B complexes are not only structurally similar (with the exception of the position of the U1 snRNP) but also functionally equivalent to CI pre-B complexes. Therefore, the CE and CI assembly pathways converge at the pre-B stage. Furthermore, the results support the conclusion that the CE to CI switch does not require the initial conversion of a CE A-like complex into a CI A complex.

## Addition of a 5′ss triggers PRP8 remodelling

We next sought to identify the molecular triggers and the order and interdependency of the various RNP remodelling events that lead to the formation of B-like complexes. To that end, we determined cryo-EM structures of complexes formed after the addition of an excess of 5′ss oligo to affinity-purified CE pre-B dimers in the absence or presence of ATP or ATPγS. ATPγS was previously shown to support the formation of human B complexes but prevents their subsequent transformation into activated B ($B^{act}$) complexes and/or their dissociation[25]. The cryo-EM structures of the protomers of affinity-purified CE pre-B dimers incubated with 5′ss oligo (designated pre-$B^{5′ss}$) or with 5′ss oligo

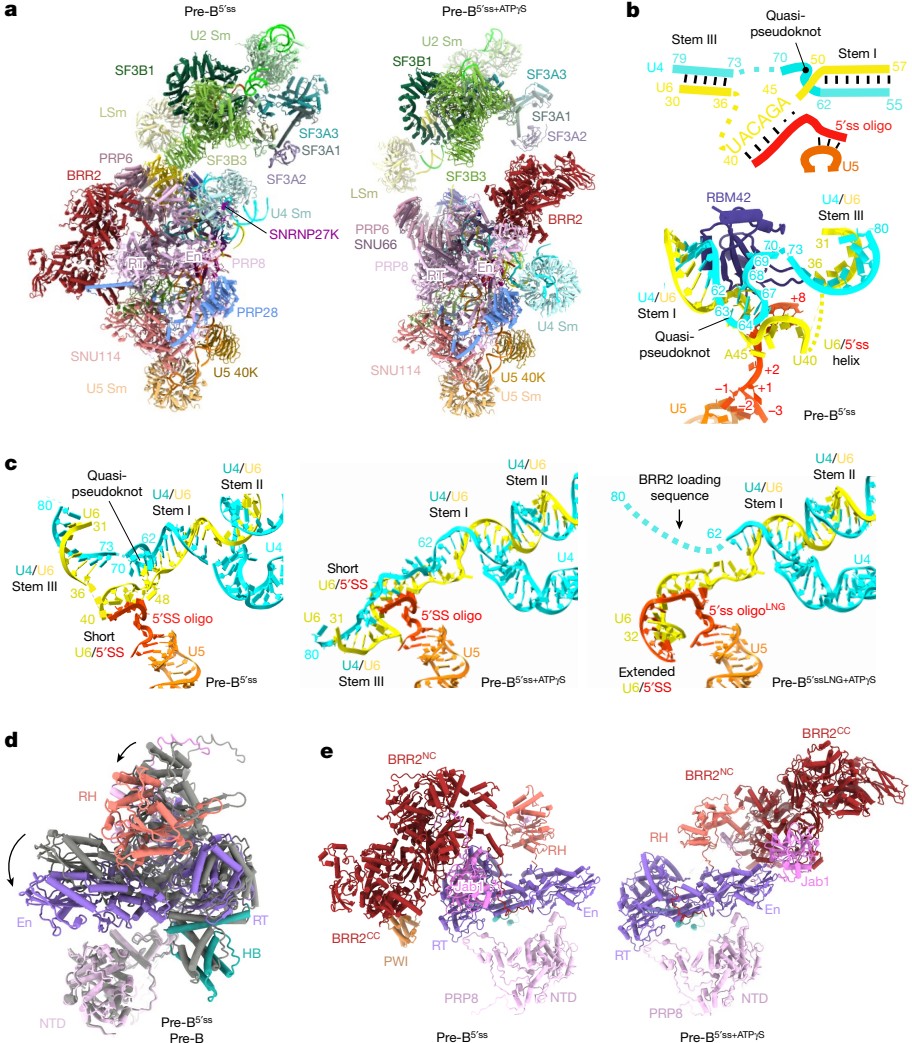

**Fig. 3 | RNP rearrangements in pre-B triggered by the addition of 5′ss oligo alone or in combination with ATPγS. a**, Molecular architecture of CE pre-B$^{5'ss}$ complexes (left) and pre-B$^{5'ss+ATPγS}$ complexes (right). Notably, PRP28 helicase is still bound to PRP8 in CE pre-B$^{5'ss}$ in essentially the same manner as in CE pre-B (compare with Fig. 1). This is presumably due to the absence of the PRP38A–SNU23–MFAP1 complex, which helps to displace PRP28 during the formation of B complexes and B-like complexes owing to their overlapping (that is, mutually exclusive) binding sites. **b**, Top, schematic of the U4/U6 snRNA helices and base-pairing interactions of the 5′ss oligo in pre-B$^{5'ss}$. Bottom, 3D organization showing the position of RBM42. **c**, 3D organization of the U4/U6 helices and interactions of the 5′ss oligo with U6 snRNA in pre-B$^{5'ss}$ (left), pre-B$^{5'ss+ATPγS}$ (middle) and pre-B$^{5'ssLNG+ATPγS}$ (right). **d**, 5′ss oligo binding triggers PRP8$^{RT/En}$ movement towards the PRP8$^{NTD}$ and concomitant movement of the PRP8 RH and HB domains and other bound proteins. An overlay of the indicated PRP8 domains in pre-B (grey) and pre-B$^{5'ss}$ (various colours) is shown. Structures are aligned through PRP8$^{NTD}$. **e**, Large-scale translocation of BRR2 requires ATP. Comparison of the location of the BRR2 helicase cassettes and the PRP8 RH, JAB1 and RT/En domains in pre-B$^{5'ss}$ versus pre-B$^{5'ss+ATPγS}$. Aligned through PRP8$^{NTD}$.

plus ATPγS (designated pre-B$^{5'ss+ATPγS}$) were determined at 4.2 Å and 3.1 Å, resolution, respectively, at their tri-snRNP cores (Fig. 3a and Extended Data Fig. 5).

Owing to the absence of the B-specific protein SNU23, which stabilizes the binding of a second 5′ss oligo in B-like complexes, only one 5′ss oligo is bound in pre-B$^{5'ss}$ in a manner similar to the 5′ss oligo 1 in B-like complexes. Thus, an extended U6/5′ss helix does not form (Fig. 3b,c and Extended Data Fig. 6a). As a consequence, U4/U6 stem III, the formation of which is mutually exclusive of that of an extended U6/5′ss helix, is still present, as well as the U4 quasi-pseudo knot and its associated proteins RBM42 and SNRNP27K (Fig. 3a–c and Extended Data Fig. 6a). In pre-B$^{5'ss}$ complexes, 5′ss oligo binding triggers the movement of the PRP8 RT/En domain (PRP8$^{RT/En}$) towards the PRP8 N-terminal domain (PRP8$^{NTD}$). This action generates a half-closed PRP8 conformation that is also found in B-like and B complexes (Fig. 3d, Extended Data Fig. 6b and Supplementary Video 1). However, no large-scale translocation of BBR2 or substantial repositioning of other tri-snRNP components like those in CE pre-B is observed (Extended Data Fig. 6b). However, PRP4 kinase (PRP4K) and the SF3B3 WD40B domain (SF3B3$^{WD40B}$) are located farther apart in pre-B$^{5'ss}$ complexes compared with pre-B complexes, and PRP4K interacts more extensively with PRP6$^{HAT}$ (Extended Data Fig. 6c–e and Supplementary Video 2), which may potentially contribute to activation of its kinase activity. The movement of PRP8$^{RT/En}$ that is triggered by the formation of the short U6/5′ss helix also destabilizes interactions between SAD1 and the BRR2 PWI domain (BBR2$^{PWI}$) and between the U4 Sm core and the PRP8 RNase H-like (RH) domain (PRP8$^{RH}$) (Extended Data Fig. 6c,d,f and Supplementary Videos 1 and 3). These contacts help to tether BRR2 and PRP8$^{RH}$ to their docking sites in the pre-B complex.

## BRR2 translocation is driven by ATP

A single 5′ss oligo is also bound in pre-B$^{5'ss+ATPγS}$ complexes (Fig. 3c and Extended Data Fig. 6g,h). Notably, the addition of ATPγS or ATP to

pre-B[5′ss] complexes leads to repositioning of the BRR2 helicase domain and to substantial restructuring of other tri-snRNP components. These include PRP31, PRP6 and PRP8[RH], which are also repositioned in B-like complexes (Fig. 3e, Extended Data Fig. 6i,j and Supplementary Videos 4 and 5). Thus, most structural rearrangements within the tri-snRNP that occur during B-like and B complex formation are driven by ATP. Notably, however, they do not require B-specific proteins, which are under-represented or absent in purified CE pre-B complexes. In pre-B[5′ss+ATPγS], BRR2 and the tightly bound PRP8 Jab1/MPN domain (PRP8[Jab1]) undergo a large-scale translocation (Fig. 3e and Supplementary Video 5). BRR2–PRP8[Jab1] is now anchored to its new position at the tip of PRP8[En] through the PRP8[RH]–PRP8[Jab1] linker, the ends of which are stably bound to PRP8[En] at the same position as in the B-like complex (Extended Data Fig. 7a–c). However, in pre-B[5′ss+ATPγS], BRR2–PRP8[Jab1] does not reach the conformational state and position that it adopts in B-like complexes, which would require a rotation of BRR2 around its long axis and substantial structural rearrangement in its helicase domains (Extended Data Fig. 7d,e and Supplementary Video 6). Furthermore, in contrast to the situation in the B-like complex, in pre-B[5′ss+ATPγS], BRR2[NC] is in an inactive conformation, in which the separator loop tightly interacts with the Sec63 domain, thereby blocking access of the U4 snRNA to the RNA-binding region of the BRR2[NC] RecA domains (Extended Data Fig. 7e). Thus, the ATP-triggered remodelling of BRR2 during the transition of pre-B[5′ss] to pre-B[5′ss+ATPγS] and then to the B-like complex is an intricate process that occurs in a stepwise manner. Consistent with the idea that a key event for the restructuring of the tri-snRNP during the pre-B to B-like to B transition is the PRP4K-mediated phosphorylation of PRP6 and PRP31 (ref. 26), both proteins are phosphorylated in pre-B[5′ss+ATP], but not in pre-B or pre-B[5′ss] (Extended Data Fig. 7f).

The U4 snRNA quasi-pseudoknot is dissolved in pre-B[5′ss+ATPγS], accompanied by displacement of RBM42 and SNRNP27K, but U4/U6 stem III is still present and, together with the U4 Sm core, adopts a new position (Fig. 3a,c and Extended Data Fig. 7g). A previous model proposed that dissociation of U4/U6 stem III is the driving force for the substantial restructuring of the tri-snRNP during the formation of human B complexes[18]. Our data indicate that most of these remodelling events do not require disruption of U4/U6 stem III or the formation of an elongated U6/5′ss helix. Instead, U4/U6 stem III is stabilized in pre-B[5′ss+ATPγS] by both RNA and protein interactions, in particular by the capture of U6 C37 by a newly identified structural element of U2 SF3A1 (Extended Data Fig. 7h,i). In addition, the absence of the B-specific protein SNU23 prevents stable binding of a second 5′ss oligo that competes with stem III formation. To elucidate whether disruption of U4/U6 stem III is a prerequisite for the final positioning of BRR2, we extended the 5′ss oligo (henceforth called 5′ss[LNG]) by 8 additional nucleotides that are fully complementary to U6 snRNA nucleotides 31–38 that form part of U4/U6 stem III. Cryo-EM of the complexes formed after incubation of affinity-purified CE pre-B with 5′ss[LNG], followed by ATPγS (designated pre-B[5′ssLNG+ATPγS]) (Extended Data Fig. 8a–g) revealed the formation of an extended U6/5′ss and the absence of U4/U6 stem III (Fig. 3c and Extended Data Fig. 7i). In addition, the U4 Sm core has moved upwards by about 5 nm (compared with pre-B[5′ss+ATPγS]) and now contacts BRR2 (Fig. 3c and Extended Data Fig. 7h,i). However, the position of BRR2 and the conformation of the BRR2[NC] RecA domains are not substantially different from pre-B[5′ss+ATPγS] (Extended Data Fig. 8h,i). These results point towards an essential role for one or more of the B-specific proteins in these BRR2 structural rearrangements.

## B-specific proteins aid BRR2 positioning

We next compared the structure of pre-B[5′ssLNG+ATPγS] with that of the B-like complex. The results indicated that the rotational movement of BRR2 and the opening of BRR2[NC] required for B-like formation are probably facilitated, among others, by the SNU23 loop 62–74 and MFAP1 helix 215–255, which bind in the B-like structure to the BRR2[NC] Sec63

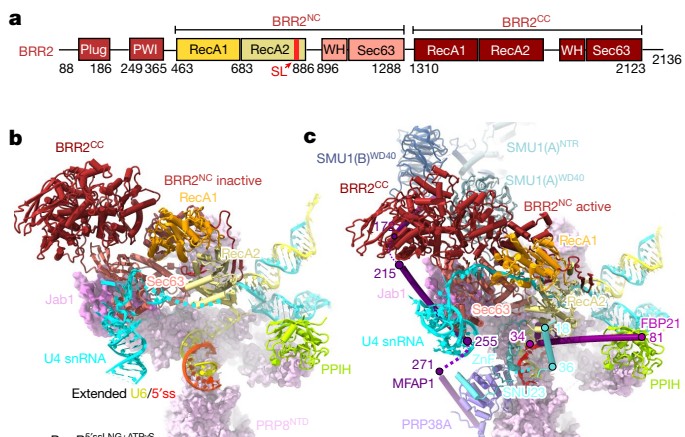

**Fig. 4 | B-specific proteins are required for the final positioning of BRR2 and the docking of its U4 snRNA substrate. a**, Schematic of the domain organization of BRR2. PWI, domain with a PWI tri-peptide located within its N-terminal region; SL, separator loop; WH, winged helix. **b,c**, Multiple B-specific proteins contact BRR2 and tether it to its activation position. Comparison of the location and conformation of BRR2 and its U4 snRNA substrate in pre-B[5′ssLNG+ATPγS] (**b**), which is formed in the absence of the B-specific proteins, and the B-like complex (**c**). In pre-B[5′ssLNG+ATPγS], the BRR2[NC] has a closed, inactive conformation and is located about 15 Å away from its final position observed in the B-like complex. In the presence of B-specific proteins, BRR2[NC] has an open, active conformation and is bound to the U4 snRNA.

domain. The binding of the long α-helix of FBP21 and α-helix 18–36 of SNU23, which both interact with RecA2 of BRR2[NC], also has an important role (Fig. 4 and Extended Data Fig. 8j–n). In B-like complexes, SNU23 also tethers BRR2[NC] to PRP8[NTD], bridging it to the U6/5′ss helix (Fig. 4 and Supplementary Video 7). The rotation of BRR2 towards PRP8[NTD] is probably further facilitated by the SMU1–RED complex, which stabilizes the new position of BRR2 in the B-like complex (Fig. 4). In B-like complexes, the BRR2 carboxy-terminal helicase cassette (BRR2[CC]) also moves to a new position that seems to be stabilized by its interaction with MFAP1 α-helix 141–172 and by SMU1 and RED (Fig. 4 and Extended Data Fig. 8k). Finally, SNU23, promotes the formation of the extended U6/5′ss helix, which frees up the U4 snRNA for its interaction with BRR2[NC]. Thus, numerous B-specific proteins are required for the final positioning of BRR2 and probably cooperate to facilitate or stabilize the active, more open conformation of BRR2, thereby allowing the docking of U4 snRNA across the BRR2[NC] RecA domains.

## CE pre-B interacts with a U1-bound 5′ss

The antiparallel orientation of the two CE pre-B complexes in the purified pre-B dimers (Fig. 1b) suggests that U1 snRNP bound to the 5′ss of the exon of one protomer may transiently interact *in trans* with PRP28 of the adjacent protomer. PRP28 is in the vicinity of the U6 ACAGA box in the pre-B protomers. This configuration therefore raises the possibility that addition of ATP alone might trigger PRP28-mediated transfer of the 5′ss of the MINX exon of one protomer to the U6 ACAGA box of the adjacent protomer. Cryo-EM of purified pre-B complexes incubated solely with ATP (denoted pre-B[ATP]) revealed highly similar molecular architectures between pre-B[ATP] protomers and pre-B[5′ss+ATPγS] (Fig. 5a–c and Extended Data Fig. 8o–v), which indicated that a U6/5′ss helix has formed. Indeed, in the pre-B[ATP] dimers, nucleotides encompassing the 5′ss directly downstream of the MINX exon RNA base pair with U5 loop 1 and the U6 ACAGA box to form a short U6/5′ss helix (Fig. 5d). Owing to structural constraints, PRP28 cannot interact within the same protomer with the U1 snRNP-bound 5′ss (Extended Data Fig. 2e). Furthermore, stoichiometric amounts of full-length MINX are present in the peak

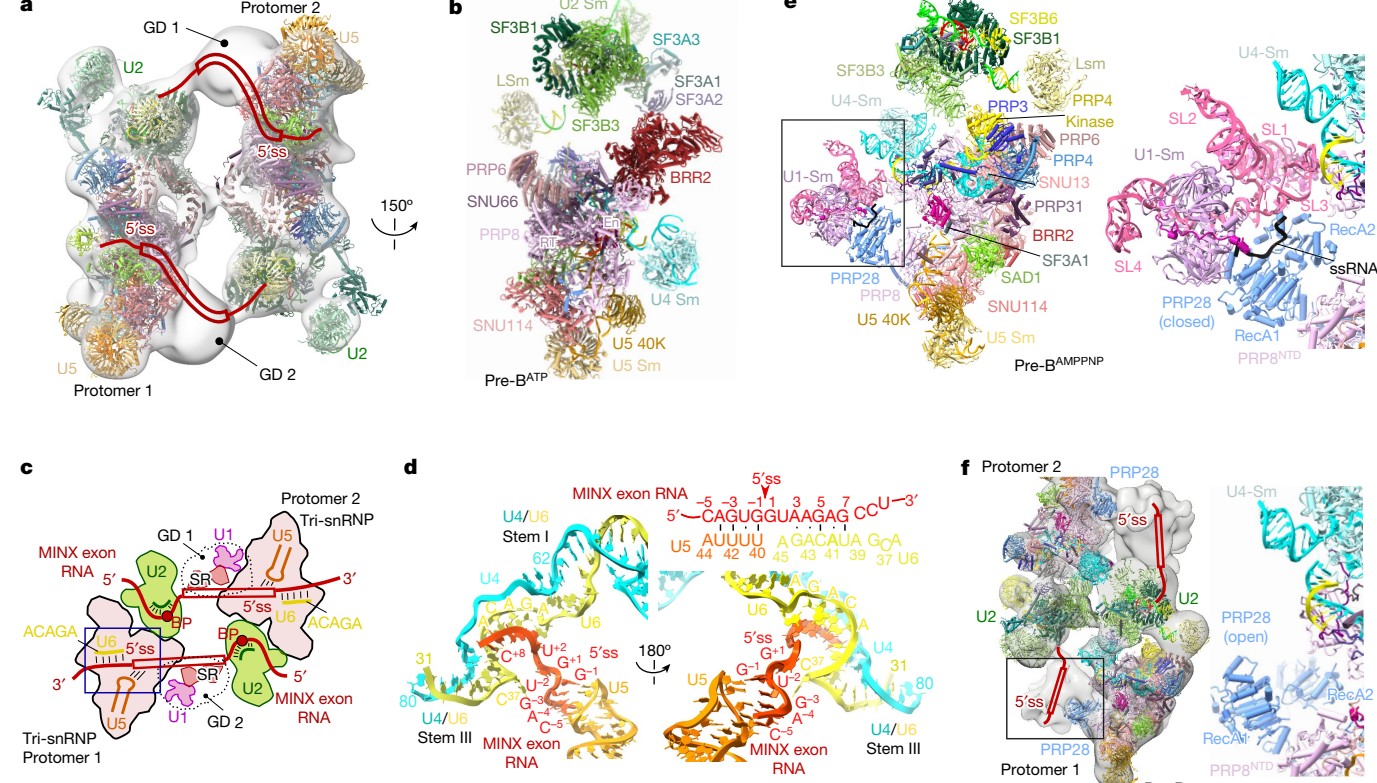

**Fig. 5 | ATP-dependent transfer *in trans* of the 5′ss from U1 snRNP to the U6 ACAGA box by PRP28. a**, Structure of purified pre-B complexes incubated with ATP alone (pre-B^ATP). Fit of the pre-B^ATP molecular model into the EM density of the pre-B^ATP dimer. **b**, Molecular architecture of the pre-B^ATP complex. The view shown in **b** is obtained if the structure of protomer 1 in **a** is rotated 150°. **c**, Cartoon showing the structural organization of the CE pre-B^ATP complex dimer. **d**, The 5′ss of the MINX exon RNA from one pre-B monomer interacts with U5 loop 1 nucleotides and the U6 ACAGA box of the adjacent monomer, forming a short U6/5′ss helix. U4/U6 stem III is still present. Top right, schematic of the

base-pairing interactions with the MINX exon RNA near or/at the 5′ss. Nucleotides at the 3′ end of the exon ($C^{-5}$ to $G^{-1}$) base pair with nucleotides of U5 loop 1. **e**, Molecular architecture of a pre-B^AMPPNP monomer bound *in trans* to the U1 snRNP of an adjacent monomer of the pre-B^AMPPNP dimeric complex. Right, zoomed-in region that is boxed on the left, showing the molecular architecture of U1 snRNP and its interaction with PRP28. ssRNA, single-stranded RNA. **f**, Comparison of the corresponding regions shown in **e** with those of the pre-B complex dimer.

gradient fractions of the pre-B^ATP dimers analysed by cryo-EM (Extended Data Fig. 8o). This finding supports the conclusion that the MINX 5′ss that is bound by U6 comes from intact MINX RNA present in the adjacent pre-B protomer; therefore, a RNA–RNA network that links both protomers is formed (Fig. 5a,c and Extended Data Fig. 9a). Notably, the PRP28 RecA domains are no longer visible in pre-B^ATP (Extended Data Fig. 8v,w), which indicates that they have been destabilized. This result is consistent with the idea that the pre-B^ATP structure is formed through ATP-dependent, PRP28-mediated transfer of the 5′ss of the MINX exon from U1 snRNA in one CE pre-B protomer to the U6 snRNA of the neighbouring protomer.

To test the latter conclusion, we incubated purified CE pre-B dimers with the non-hydrolysable ATP analogue AMPPNP and subsequently determined their structure by cryo-EM (Extended Data Fig. 9b–h). The spatial organization of the tri-snRNP proteins in pre-B^AMPPNP complexes was the same as in the CE pre-B protomers (Fig. 5e,f and Extended Data Fig. 9i,j). Notably, in the presence of AMPPNP, U1 snRNP, presumably from the adjacent protomer, was now stably docked to the PRP28 helicase domain of the pre-B protomer, as evidenced by the improved resolution in this region of the pre-B^AMPPNP complexes compared with pre-B complexes (Fig. 5e,f and Extended Data Fig. 9i,j). This enabled the localization of U1 snRNP components (Fig. 5e and Extended Data Fig. 9i) and further supports the idea that the poorly resolved density element of the CE pre-B complex indeed contains U1. In contrast to pre-B complexes, in which the RecA domains of PRP28 are in an inactive, open conformation (Fig. 5f and Extended Data Fig. 9j, l–o), in pre-B^AMPPNP

they exhibit a closed conformation (Fig. 5e and Extended Data Fig. 9i,k). Consistent with the absence of additional structural rearrangements in tri-snRNP components, the 5′ss of the adjacent protomer is not base paired to the U6 ACAGA box (Extended Data Fig. 9p,q). Indeed, the release and transfer of the 5′ss to the U6 ACAGA box would require ATP hydrolysis and lead to the release of the RecA domains of PRP28 from U1 snRNP and PRP8, as observed in the pre-B^ATP complex. Taken together, our data demonstrate that the tri-snRNP of a CE pre-B complex is capable of interacting *in trans* with a U1 snRNP bound to a 5′ss, which leads to the handover of this 5′ss from U1 to U6 by PRP28. This result further indicates that the latter handover does not require that the CE U2–U1 molecular bridge in a CE pre-B complex is first converted into a CI U1–U2 interaction, as the former bridge is still intact in pre-B^ATP. Our data indicate that the conversion of a CE pre-B to a CI B complex can occur within the context of a CE pre-B dimer. However, formation of the latter is probably not a pre-requisite for the CE to CI switch within the cell, where CE pre-B complexes probably exist predominantly as monomers and can therefore productively engage with an upstream, U1-bound 5′ss *in cis* (see below).

## Discussion

The cryo-EM structures of the CE pre-B and B-like complexes, together with our cryo-EM coupled, add-back experiments using purified CE pre-B complexes, provide new insights into the switch from a CE to CI organized spliceosome. We also provide new information on the

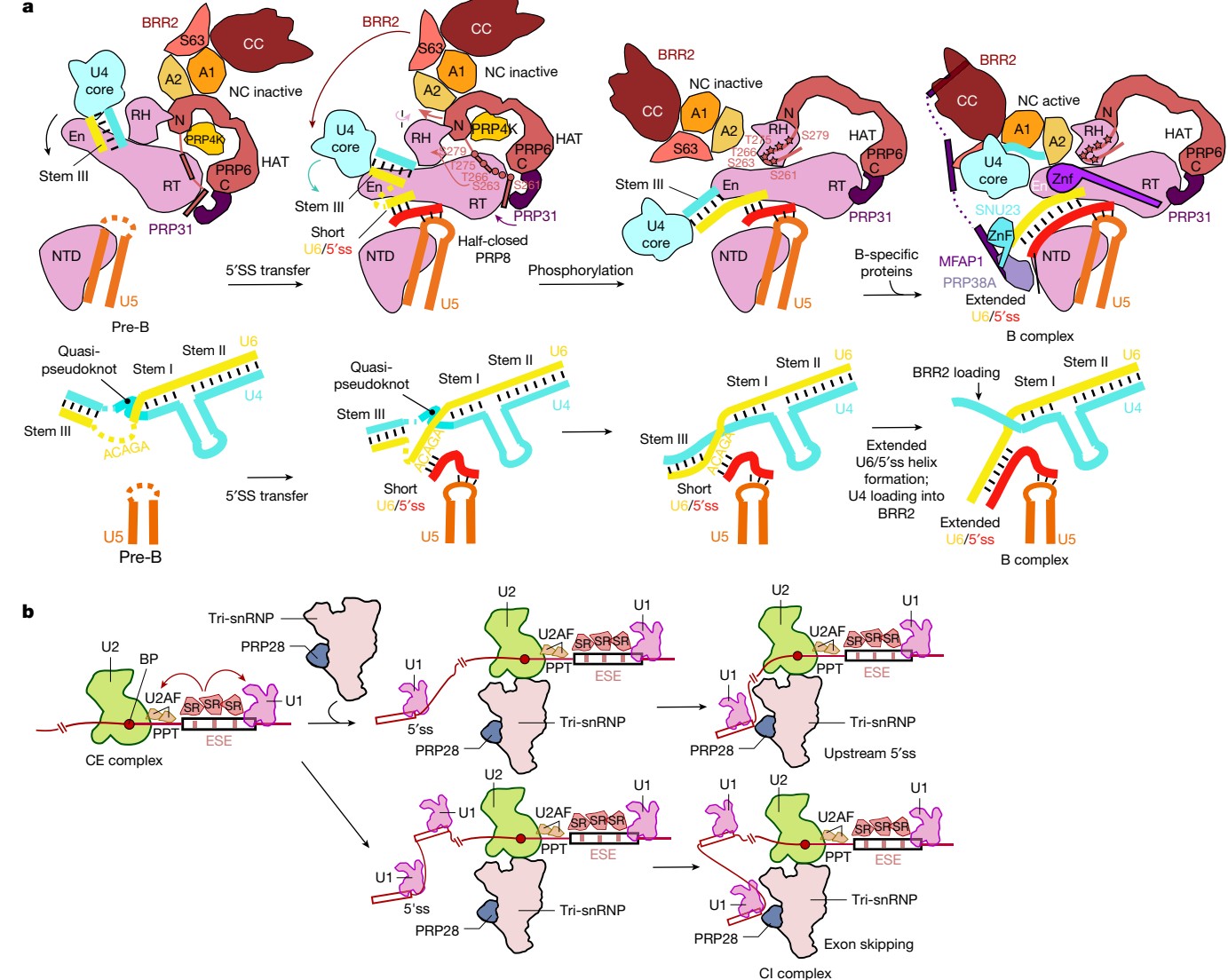

**Fig. 6 | Structural model of the conversion of a spliceosomal CE pre-B complex to a CI B complex. a**, Schematic of RNA remodelling (bottom) and rearrangement and repositioning of tri-snRNP proteins (top) during the pre-B to B-like to B complex transition. First, binding of a 5′ss to the U6 ACAGA box leads to the formation of a short U6/5′ss helix and rearrangements in PRP8 to produce a half-closed conformation. Following phosphorylation of PRP6 and PRP31 (indicated by stars), major structural remodelling of the tri-snRNP occurs, including the large-scale movement of the BRR2 helicase cassettes, and the U4/U6 quasi-pseudoknot is dissolved. B-specific proteins are recruited to the remodelled complex and tether BBR2 to its activation position. At the same time, they facilitate the formation of the extended U6/5′ss helix, freeing the U4 nucleotides that are subsequently bound by the BRR2 helicase. **b**, Model of the conversion of a CE to CI spliceosome, including alternative 5′ss/U1 snRNP choices.

intricate RNP remodelling events that occur during the transformation of a CE pre-B complex into a CI B complex. Given the structural similarities of a CE and CI pre-B complex, the order of the latter rearrangements and the intermediate structural states that our studies uncovered, as well as their molecular triggers, should also apply to the conversion of a CI pre-B complex into a CI B complex. We showed that the formation of a U6/5′ss helix induces a structural change in PRP8 such that it now adopts the PRP8 conformation found in B complexes and B-like complexes (Fig. 6a). This PRP8 conformational change destabilizes the SAD1–BBR2[PWI] and U4 Sm core–PRP8[RH] interactions, which are probably prerequisites for BRR2 translocation, and leads to the repositioning of PRP4K that might help activate its kinase activity. Thus, U6/5′ss helix formation seems to be a prerequisite for several subsequent, ATP-dependent RNP rearrangements, and at the same time ensures that the 5′ss is stably bound before the more extensive structural rearrangements that subsequently occur during formation of B complexes.

Our studies revealed that BRR2 translocation and multiple other tri-snRNP rearrangements that accompany the formation of B complexes and B-like complexes are driven by ATP. Moreover, PRP4K-mediated phosphorylation of PRP6 and PRP31 triggers a cascade of structural rearrangements in the pre-B complex that lead to BRR2 translocation (Fig. 6a). These include, among others, the repositioning of PRP31, PRP6[HAT] and PRP8[RH], and the displacement of SNRNP27K, RBM42 and PRP4K, which in some cases involve a mutually exclusive interaction or location. For example, PRP6[HAT] can only be repositioned following the translocation of BRR2, and SNRNP27K has to be displaced from the tip of PRP8[En] to allow BRR2–PRP8[Jab1] to dock there following its translocation (Extended Data Fig. 7a,b). Thus, protein phosphorylation coupled with structural rearrangements that generate mutually exclusive positions or protein-binding sites are likely to be the main driving forces for the repositioning of BRR2, a prerequisite for the subsequent spliceosome activation step. Indeed, mutually exclusive RNP interactions are a general mechanism that the spliceosome uses

to regulate major structural transitions and to ensure progression of not only spliceosome assembly but also its catalytic activation and splicing catalysis.

B-specific proteins play a key part in stabilizing the extended form of the U6/5′ss helix, as we demonstrated that it does not form in their absence. Our studies also showed that formation of the short U6/5′ss helix is not sufficient to disrupt U4/U6 helix III, which is a prerequisite for binding of U4 snRNA by BRR2 (Fig. 6a). Notably, our add-back experiments showed that the majority of the other RNP remodelling events that occur during the pre-B to B complex transition can occur in the absence of the B-specific proteins. Thus, all of the proteins needed for the release of BRR2 from its pre-B binding site and for its large-scale translocation and docking to its new position at PRP8^En are already present in CE and CI pre-B complexes. However, our studies revealed a key role for the B-specific proteins during the final positioning of BRR2 and the opening of the RNA-binding channel of its N-terminal helicase cassette. Most of the binding sites of the B-specific proteins are created during or after the translocation of BRR2 to its pre-activation position, which is consistent with these proteins functioning primarily first after BRR2 translocation, and further indicates that their final positioning and that of BRR2 probably occurs in a coordinated manner. This supports the conclusion that the intermediate conformation of BRR2 that we observed in our purified system in the absence of the B-specific proteins is indeed physiologically relevant.

Finally, our structural data provided new mechanistic insights into the conversion of CE splicing complexes into those assembled across an intron (Fig. 6b). In contrast to previous models of the CE to CI switch, our studies clearly demonstrated that there is no need to establish a CI U1–U2 interaction during the conversion from a CE to CI organized complex. Instead, we showed that the tri-snRNP of a CE pre-B complex has the potential to interact with an upstream U1/5′ss complex, without previous formation of a CI A complex. It subsequently undergoes the extensive rearrangements that generate a B complex, which in the presence of B-specific proteins can then be converted into a catalytically active spliceosome. As the CE and CI pre-B complexes seem to be functionally equivalent, the two assembly pathways mechanistically converge at the pre-B stage. Our structural data further suggest that it may not be necessary to break contacts across the downstream exon to establish a CI interaction with an upstream 5′ss once U2 snRNP and tri-snRNP interactions are in place. This in turn would indicate that the molecular bridge connecting U2 and U1 across an exon has no apparent effect on subsequent spliceosome assembly across the upstream intron. This further supports the idea that one of the primary functions of a CE complex is to promote U2AF–U2 snRNP binding to weak BS or polypyrimidine tracts. The structure of a CE pre-B complex that we reveal here suggests that the pre-B complex can engage in different forms of pre-mRNA splicing depending on which 5′ss it subsequently interacts with. When the upstream U1–5′ss of the same pre-mRNA interacts, canonical splicing that leads to the ligation of the two exons separated by a single intron would occur (Fig. 6b). It is also possible that different U1–5′ss complexes may compete for binding with the U2–tri-snRNP complex, leading to alternative 5′ss usage or exon skipping, or other less frequently occurring forms of splicing, such as *trans*-splicing or back-splicing. Our studies are consistent with the idea that commitment to the usage of a particular 5′ss and its functional pairing to a given 3′ss occurs primarily during the pre-B to B complex transition when the 5′ss is handed over from U1 to the U6 snRNA. Thus, the regulation of the usage of alternative 5′ss, which would affect the level of exon skipping, could be achieved by modulating the initial interaction with the tri-snRNP and/or productive interaction of PRP28 with the bound U1/5′ss.

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

## Methods

### MS2 affinity selection of CE complexes

HeLa S3 cells were obtained from the Helmholtz Zentrum für Infektionsforschung, Braunschweig and tested negative for mycoplasma. Cells were not authenticated. HeLa nuclear extracts were prepared according to a previously published method[27] and were dialysed twice for 2.5 h against 50 volumes of Roeder D buffer (20 mM HEPES-KOH, pH 7.9, 0.2 mM EDTA, pH 8.0, 1.5 mM MgCl$_2$, 100 mM KCl, 10% (v/v) glycerol, 0.5 mM DTT and 0.5 mM PMSF). For both pre-B and B-like complexes, 10 nM m$^7$G(5')ppp(5')G-capped MINX exon RNA containing 3 MS2 aptamers at its 3' end[7] was pre-incubated with 100 nM MS2–MBP fusion protein for 40 min on ice before addition to the splicing reaction. Splicing reactions were carried out at 30 °C with 50% (v/v) nuclear extract in splicing buffer (1.5 mM MgCl$_2$, 65 mM KCl, 20 mM HEPES-KOH pH 7.9, 2 mM ATP and 20 mM creatine phosphate). For pre-B complexes, splicing was performed for 20 min. To obtain B-like complexes, a 100-fold molar excess of a 5'ss oligo (5'-AAG/GUAAGUAU-3', where / indicates the exon–intron boundary) was added after allowing pre-B complex formation, and the reaction was incubated for an additional 10 min at 30 °C. Splicing reactions were then chilled on ice for 10 min, centrifuged 15 min at 18,000$g$ to remove aggregates and loaded onto a MBP Trap HP column (GE Healthcare). The column was washed with G-75 buffer (20 mM HEPES-KOH pH 7.9, 1.5 mM MgCl$_2$ and 75 mM NaCl) and complexes were eluted with G-75 buffer containing 15 mM maltose. Eluted complexes were loaded onto a linear 10–30% (v/v) glycerol gradient prepared in G-75 buffer, centrifuged at 17,500 r.p.m. for 18 h at 4 °C in a TST41.14 rotor (Thermo Fisher Scientific), and fractions were collected from the bottom of the gradient. RNA from complexes in peak gradient fractions was separated on a denaturing 4–12% NuPAGE gel (Life Technologies) and visualized by staining with SYBR Gold (Thermo Fisher Scientific). For cryo-EM analysis, eluted complexes were subjected to gradient fixation (GRAFIX)[28] and further processed as described below.

### Add-back experiments using purified pre-B

CE pre-B complexes were MS2 affinity-purified as described above. To obtain pre-B$^{5'ss}$ complexes, affinity-purified complexes were subsequently incubated for 10 min at 0 °C or 30 °C with a 100-fold molar excess of the 5'ss oligo alone. To generate pre-B$^{5'ss+ATP/ATPγS}$ complexes, after incubation with the 5'ss oligo at 0 °C, complexes were subsequently incubated for 30 min at 30 °C after addition of 2 mM ATP or ATPγS. To generate pre-B$^{5'ssLNG+ATPγS}$ complexes, purified pre-B complexes were incubated with a 100-fold molar excess of an elongated 5'ss oligo (5'-AAG/GUAAGUAUCGUUCCAA-3') for 10 min at 0 °C, followed by the addition of 2 mM ATPγS and incubation for an additional 30 min at 30 °C. To obtain pre-B$^{ATP}$ or pre-B$^{AMPPNP}$ complexes, affinity-purified pre-B complexes were incubated with 2 mM ATP or AMPPNP, respectively, for 30 min at 30 °C. All complexes were then loaded onto a linear 10–30% (v/v) glycerol gradient prepared in G-75 buffer, and centrifuged and analysed as described above. For cryo-EM analyses, eluted complexes were subjected to gradient fixation (GRAFIX) and further processed as described below.

### Western blotting

For western blot analysis of purified pre-B, pre-B$^{5'ss}$, pre-B$^{5'ss+ATP}$ and pre-B$^{ATP}$ complexes, 200 fmoles of each complex was separated on 4–12% NuPAGE gels and transferred to a Hybond P membrane. Membranes were first blocked with 5% milk in 1× TBS-T buffer (20 mM Tris-HCl, pH 7.5, 150 mM NaCl and 0.1% Tween 20) and then incubated with rabbit antibodies against human phospho-PRP6 (1:1,000 dilution) and phospho-PRP31 (1:500 dilution), followed by antibodies against human SF3B1 (1:700 dilution), PRP31 (1:500 dilution) and PRP6 (1:1,000 dilution; AB99292, Abcam). Subsequent to incubation with the primary antibodies, membranes were washed with TBS-T buffer and incubated with HRP-conjugated goat anti-rabbit IgG (1:30,000 dilution; 111-035-144, Jackson Immunoresearch). After washing, membranes were immunostained using an enhanced chemiluminescence detection kit (GE Healthcare) and the signal was visualized using an Amersham Imager 680.

### Protein–protein crosslinking

CE pre-B complexes were MS2 affinity-purified as described above with the following modifications: after affinity purification, eluted complexes were crosslinked with 400 µM BS3 for 45 min at 18 °C in a total volume of 1.4 ml. Crosslinked complexes were loaded onto a linear 10–30% (v/v) glycerol gradient and subjected to centrifugation at 17,500 r.p.m. for 18 h at 2 °C in a TST41.14 rotor. Four peak fractions containing dimeric pre-B complexes were pooled and ultracentrifuged in a S100-AT4 rotor (Thermo Fisher Scientific). The pelleted, crosslinked dimeric CE pre-B complexes (approximately 20 pmol) were dissolved in 50 mM ammonium bicarbonate buffer containing 4 M urea, reduced with dithiothreitol, alkylated with iodoacetamide and, after diluting the urea to 1 M, in-solution digested with trypsin. Peptides were reverse-phase extracted using Sep-Pak Vac tC18 1cc cartridges (Waters), lyophilized and subsequently dissolved in 40 µl 2% acetonitrile (ACN) and 20 mM ammonium hydroxide. Peptides were separated on an xBridge C18 3.5 µm 1 × 150 mm reverse-phase column (Waters) using a 4–48% gradient of ACN in 10 mM ammonium hydroxide over 45 min at a flow rate of 60 µl min$^{-1}$. One-minute fractions of 60 µl were collected, pooled in a step of 12 min (resulting in 12 pooled fractions in total), vacuum dried and dissolved in 5% ACN and 0.1% trifluoroacetic acid (TFA) for subsequent uHPLC-ESI–MS/MS analysis that was performed in triplicate on an Orbitrap Exploris 480 (Thermo Scientific). The mass spectrometer was coupled to a Dionex UltiMate 3000 uHPLC system (Thermo Scientific) with a custom 35 cm C18 column (75 µm inner diameter packed with ReproSil-Pur 120 C18-AQ beads, 3 µm pore size (Dr. Maisch)). The MS1 and MS2 resolutions were set to 120,000 and 30,000, respectively. Only precursors with a charge state of 3–8 were selected for MS2. MS data were acquired using Thermo Scientific Xcalibur (v.4.4.16.14) software.

B-like complexes were MS2 affinity-purified as described above with the following modifications: after affinity purification, eluted spliceosomal complexes were crosslinked with 350 µM BS3 for 30 min at 18 °C in a total volume of 2 ml. Crosslinked complexes were loaded onto a linear 10–30% (v/v) glycerol gradient and subjected to centrifugation at 21,000 r.p.m. for 12 h at 2 °C in a TST41.14 rotor. Peak fractions containing dimeric B-like complexes (about 12 pmol) were pooled, pelleted, digested and the peptides were reverse-extracted as described above for the pre-B complexes. Peptides were fractionated by gel filtration using a Superdex Peptide PC3.2/30 column (GE Healthcare) in 30% ACN and 0.1% TFA. Fifty-microlitre fractions corresponding to an elution volume of 1.2–1.8 ml were analysed in duplicate on a Thermo Scientific Orbitrap Fusion Lumos Tribrid mass spectrometer coupled to Ultimate 3000 uHPLC (Thermo Scientific). MS data acquisition was performed using Thermo Scientific Xcalibur (v.4.5.445.18) software.

The protein composition of the spliceosomal complexes was determined in a search with MaxQuant (v.2.4.2.0) against a UniProt human reference proteome using the same samples but before pre-fractionation by either offline reverse phase (pre-B) or size exclusion (B-like) chromatography. Based on the MaxQuant results, restricted protein databases were compiled and used for protein–protein crosslink identification by searching the Thermo raw files with pLink (v.2.3.11) for pre-B and pLink (v.2.3.9) for B-like complexes[29]. For model building, a maximum distance of 30 Å between the Cα atoms of the crosslinked lysine residues was allowed.

### EM sample preparation and imaging

For cryo-EM samples, spliceosomal complexes were loaded onto a linear 10–30% (v/v) glycerol gradient prepared in G-75 buffer containing 0–0.1% glutaraldehyde (GRAFIX) and centrifuged at 17,500 r.p.m.

for 18 h at 4 °C in a TST41.14 rotor. Fractions were collected from the bottom of the gradient and were quenched with 120 mM Tris-HCl pH 7.5 on ice. Complexes in the peak gradient fractions were pooled, buffer-exchanged and concentrated in an Amicon 50 kDa cut-off unit. Complexes were then adsorbed for 20 min to a thin layer carbon film that was subsequently attached to R2/2 UltrAuFoil grids (Quantifoil). A volume of 3.8 μl of double-distilled water was applied to the grids and excess water was blotted away using a FEI Vitrobot loaded with pre-wet filter paper, with the following settings: blotting force of 11 and blotting time of 7.5 s at 4 °C and 100% humidity. Samples were subsequently vitrified by plunging into liquid ethane cooled to liquid nitrogen temperature. Cryo-EM grids of the pre-B, pre-B[5′ss], pre-B[5′ss+ATPγS], pre-B[5′ssLNG+ATPγS] and pre-B[ATP] were imaged in a Titan Krios 1 (Thermo Fisher Scientific), equipped with a Cs corrector, operated at 300 kV, on a Falcon III detector in linear mode at a calibrated pixel size of 1.16 Å at the specimen level (see Extended Data Table 1 for a summary of EM statistics). Cryo-EM grids of B-like and pre-B[AMPPNP] were imaged in a Titan Krios 3 (Thermo Fisher Scientific), operated at 300 kV, on a Falcon III detector in linear mode at a calibrated pixel size of 1.35 Å at the specimen level. Krios1 and Krios3 cryo-EM images were acquired using Thermo Fisher EPU2.1 with an exposure time of 1.02 s (40 movie frames), with a total dose of 60 e⁻ Å[-2] and 48 e⁻ Å[-2], respectively.

### EM data processing
For all of the cryo-EM datasets, frames were aligned, dose-weighted and summed using MotionCor (v.2.0)[30]. Defocus values were estimated using Gctf[31]. Particle picking was performed using crYOLO[32]. For each sample, approximately 800–1,000 particles were manually picked from 30–50 micrographs and used to train a neural network model, which was then used to automatically pick particles for the corresponding dataset. All subsequent processing was performed using RELION 3.1 (http://www2.mrc-lmb.cam.ac.uk/relion/index.php/Main_Page) unless otherwise specified. Cryo-EM data were split randomly into two halves for gold-standard FSC determination in RELION 3.1.

For the pre-B complex, 777,350 particles were picked from 25,904 micrographs, extracted and binned to 200 × 200 pixels (3× binned, pixel size of 3.48 Å). After reference-free two-dimensional (2D) classification, 586,198 particles were retained for further processing, from which 100,000 particles were used for ab initio reconstruction in cryoSPARC[33]. The ab initio model showed a 3D structure resembling that of a CI pre-B complex but with additional fuzzy densities (which later turned out to be the second protomer). The fuzzy density, as well as the unstable U2 density, was erased using Chimera[34], and the resulting 3D structure was low-pass filtered to 40 Å resolution to prevent model bias, which was then used for 3D classification in RELION 3.1. The 586,198 particles were 3D classified into 5 classes that contained 2 major types of particles. In class 1, both protomers were well-defined, whereas in class 2, only one well-defined pre-B protomer was observed. To improve the resolution of the pre-B protomer, the two protomers were separately re-extracted, re-centred in a box of 160 × 160 pixels (3× binned, pixel size of 3.48 Å) using the alignment parameters from the first round of 3D classification. The resulting particles were then 3D classified separately, and the good particles were combined and subjected to a masked 3D classification, focusing on the tri-snRNP density. The 279,781 particles showing a well-defined tri-snRNP density were then re-extracted in the original pixel size in a 480 × 480 box. 3D refinement, CTF refinement and Bayesian polishing were performed in two rounds. In the final round of 3D refinement, soft masks around the tri-snRNP core—encompassing PRP8[NTD], the PRP8 Large domain (PRP8[Large]), U4/U6 stem I and stem III, U5 snRNA, SNU114, SAD1, DIM1, the PRP6 N-terminal domain (PRP6[NTD]), SF3A1 C-terminal region (SF3A1[CT]), the PRP28 N-terminal domain (PRP28[NTD]) and the BRR2 N-terminal domain (BRR2[NTD])—and the BRR2 region (encompassing the BRR2 helicase domain and PRP8[Jab1]) were applied, producing two 3D reconstructions at nominal resolution of 3.5 Å and 4.2 Å, respectively.

Focused classification without alignment was applied to improve the U4 core region (encompassing the U4 Sm domain, SNU66, U4/U6 stem I and stem III, RBM42, PRP8[RH] and PRP8[En]) and the U2 region (encompassing the U2 5′ domain comprising SF3b proteins, U2/U6 helix II and U6 Lsm proteins). After masked refinement, the U4 core and the U2 region were resolved to nominal resolutions of 6.1 Å and 12 Å, respectively.

For the B-like complex, 488,598 particles from 14,665 micrographs were picked, extracted and binned to 200 × 200 pixels (3× binned, pixel size of 4.05 Å). A total of 389,830 particles were retained after reference-free 2D classification, from which 100,000 particles were used for ab initio reconstruction in cryoSPARC[33]. The ab initio model showed a dimeric structure. The less well-defined protomer was erased using Chimera[34], and the better-defined protomer was low-pass filtered to 40 Å and used as the starting model for 3D classification of the entire dataset. For the first round of 3D classification, a soft mask around one protomer was applied, so that all the particles were forced to align to only one protomer. This separated particles that had at least one well-defined protomer from the bad particles. To investigate whether the good particles contain monomeric B-like complexes, after the first round of 3D classification the particles were further 3D classified into four classes without a mask. All of the 3D classes showed well-defined dimeric complexes, which suggested that all of the good particles are dimeric B-like complexes. To improve the resolution of the B-like protomers, particles were re-centred and re-extracted in the original pixel size in a 480 × 480 pixels box. Two rounds of 3D refinement, CTF refinement and Bayesian polishing were performed. In the final round of 3D refinement, soft masks around the tri-snRNP core (encompassing PRP8, the 5′ss oligo, U4/U6 stem I and stem II, U5 snRNA, PRP3, SNU114, SAD1, DIM1, PRP6[NTD], SF3A1[CT], PRP28[NTD], SNU13, FBP21, SNU23, MFAP1 and PRP38A), the BRR2 region (encompassing the BRR2 helicase domain and PRP8[Jab1]) and the U4/U6 region (encompassing U4/U6 stem I and stem II, SNU13, PRP3, PRP4, PRP31, PPIH, PRP6[HAT] and PRP8[RH]) were applied, producing three 3D reconstructions with nominal resolutions of 3.1 Å, 4.3 Å, and 3.3 Å, respectively. Focused classification without alignment was applied to improve the U2 region (encompassing the U2 5′ region, U2/U6 helix II and SMU1). After masked refinement, the U2 region was improved to about 12 Å resolution.

For the pre-B[5′ss] complex, 1,283,541 particles from 23,372 micrographs were picked, extracted and binned to 200 × 200 pixels (3× binned, pixel size of 3.48 Å). Overall, 944,381 particles were retained after reference-free 2D classification and subjected to 3D classification using the low-pass filtered tri-snRNP part of the pre-B complex or the tri-snRNP core of the B-like complex (excluding the BRR2 region) as the starting model. Both starting models generated the same result, with one 3D class containing a well-defined protomer. No class resembling the B-like complex was found even when the tri-snRNP core of the B-like complex was used as the starting model. To separate the class 1 and class 2 dimers, the good 3D class was further classified into nine classes. To improve the resolution of the pre-B[5′ss] protomer, particles were re-centred and re-extracted in the original pixel size in a 480 × 480 pixels box, and another round of 3D classification was performed with a soft mask around the tri-snRNP region. The good class was selected and 3D refined, followed by one round of CTF refinement and Bayesian polishing. The final 176,879 particles were 3D refined with a soft mask around the tri-snRNP core (encompassing 5′ss oligo, PRP8[NTD], PRP8[Large], U4/U6 stem I and stem III, U5 snRNA, SNU114, SAD1, DIM1, PRP6[NTD], SF3A1[CT], PRP28[NTD] and BRR2[NTD]), resulting in a 3D reconstruction at a nominal resolution of 4.2 Å.

For the pre-B[5′ss+ATPγS] complex, 791,079 particles were picked, extracted and binned to 200 × 200 pixels (3× binned, pixel size of 3.48 Å). As the ab initio reconstruction from cryoSPARC largely resembles the B-like complex, the tri-snRNP core (excluding BRR2) of the B-like complex was low-pass filtered to 40 Å and used as the starting model for 3D classification. The good classes were combined,

re-centred and re-extracted in the original pixel size in a 480 × 480 pixels box. Two rounds of 3D refinement, CTF refinement and Bayesian polishing were performed, and the final 411,185 particles were refined with a soft mask around the tri-snRNP core (encompassing PRP8, 5′ss oligo, U4/U6 stem I and stem II, U5 snRNA, PRP3, SNU114, SAD1, DIM1, PRP6$^{NTD}$, SF3A1$^{CT}$, PRP28$^{NTD}$ and SNU13), producing a 3D reconstruction at a nominal resolution of 3.1 Å. Focused classification without alignment followed by a masked refinement was applied to improve the BRR2 region (encompassing the BRR2 helicase domain, PRP8$^{En}$ and PRP8$^{Jab1}$) to a nominal resolution of 4.0 Å.

For the pre-B$^{5′ssLNG+ATPγS}$ complex, 541,230 particles from 13,740 micrographs were picked, extracted and binned to 200 × 200 pixels (3× binned, pixel size of 3.48 Å). Using the low-pass filtered tri-snRNP core of the pre-B$^{5′ss+ATPγS}$ complex as a starting model, the particles were 3D classified, and particles from the best class were re-centred and re-extracted in the original pixel size in a 480 × 480 pixels box. After two rounds of 3D refinement, CTF refinement and Bayesian polishing, the final 136,333 particles were refined with a soft mask around the tri-snRNP core (encompassing PRP8, the long 5′ss oligo, U4/U6 stem I and stem II, U5 snRNA, PRP3, SNU114, SAD1, DIM1, PRP6$^{NTD}$, SF3A1$^{CT}$, PRP28$^{NTD}$ and SNU13), producing a 3D reconstruction at a nominal resolution of 3.7 Å.

For the pre-B$^{ATP}$ complex, 757,260 particles from 11,752 micrographs were picked, extracted and binned to 200 × 200 pixels (3× binned, pixel size of 3.48 Å). A total of 499,792 particles were retained after 2D classification. Various starting models were tested for 3D classification, including the tri-snRNP core from the pre-B, pre-B$^{5′ss}$ and pre-B$^{5′ss+ATPγS}$ complexes. The density of BRR2 was erased from all of the starting models to prevent model bias, and the low-pass filtered tri-snRNP core from the pre-B$^{5′ss+ATPγS}$ complex worked best for 3D classification. No class resembling pre-B or pre-B$^{5′ss}$ was detected even when the two complexes were used as the starting model. The particles from the best 3D class (94,460 particles) were further 3D classified with a resolution limit of 30 Å, which showed that 25.4% of the particles contain a well-resolved second protomer. The rest (74.6%) of the particles showed a poorly resolved second protomer, which was due to either the flexibility of the second protomer or the lack of stable tri-snRNP integration in the second protomer (that is, it consists of a CE A-like complex). Given that all of the 94,460 particles contained at least one good protomer, for 3D reconstruction of the high-resolution core, all of these particles were re-centred and re-extracted in the original pixel size in a 480 × 480 pixels box. After two rounds of 3D refinement, CTF refinement and Bayesian polishing, the final 3D refinement was performed with a soft mask around the tri-snRNP core (encompassing PRP8, the 5′ss region of the MINX exon RNA, U4/U6 stem I and stem II, U5 snRNA, PRP3, SNU114, SAD1, DIM1, PRP6$^{NTD}$, SF3A1$^{CT}$, PRP28$^{NTD}$ and SNU13), producing a 3D reconstruction at a nominal resolution of 3.7 Å. Subsequent local 3D classification around the tri-snRNP core did not reveal further structural heterogeneity, which suggested that the tri-snRNP core remains identical regardless of the presence or absence of a stable second protomer.

For the pre-B$^{AMPPNP}$ complex, 619,945 particles from 20,337 micrographs were picked, extracted and binned to 200 × 200 pixels (3× binned, pixel size of 4.05 Å). In total, 371,919 particles were retained after 2D classification. The same set of starting models prepared for the pre-B$^{ATP}$ complex was used for 3D classification of the pre-B$^{AMPPNP}$ complex, with the low-pass filtered tri-snRNP core from the pre-B complex working best. No class resembling pre-B$^{5′ss+ATPγS}$ was detected even when it was used as the starting model. The particles from the best 3D class were re-centred and re-extracted in the original pixel size in a 480 × 480 pixels box. After one round of 3D refinement, CTF refinement and Bayesian polishing, the final 53,422 particles were refined with a soft mask around the tri-snRNP core (encompassing PRP8$^{NTD}$, PRP8$^{Large}$, U4/U6 stem I and stem III, U5 snRNA, SNU114, SAD1, DIM1, PRP6$^{NTD}$, SF3A1$^{CT}$, PRP28$^{NTD}$ and BRR2$^{NTD}$), producing a 3D reconstruction at a

nominal resolution of 4.1 Å. Focused classification without alignment followed by a masked refinement was applied to improve the PRP28/U1 snRNP region (encompassing PRP8, U5 snRNA, PRP28 and U1 snRNP) to a nominal resolution of 6.1 Å.

## Model building and refinement

Model building was carried out by docking cryo-EM, crystal and AlphaFold2 structures into EM density and adjusting in COOT[35]. A list of modelled protein and RNA components, as well as their corresponding model templates, is provided in Supplementary Table 3. In brief, the CE pre-B complex was modelled by fitting the tri-snRNP and U2 snRNP parts of the CI pre-B complex (Protein Data Bank (PDB) identifier 6QX9) into the EM density as rigid bodies. The U2 part (including the SF3B core complex, the SF3A core complex, U2 Sm, U2-A′ and U2-B″) was truncated to a polyalanine chain without further adjustment. The SF3B6 protein was modelled based on its position relative to the SF3B1 C-terminal HEAT domain in the A-like complex (PDB 7Q4O) without further adjustment, consistent with crosslinks (Supplementary Table 2 and Extended Data Fig. 2). For the high-resolution tri-snRNP part, each individual component and its side chains were adjusted manually in COOT. The B-like complex was modelled by fitting the B complex model (PDB 8Q7N) into the EM density, and the parts that are absent in B-like (that is, UBL5, an extended U6/5′ss helix, TCERG1 and BUD31) were deleted from the model. Two copies of the 5′ss oligo were modelled de novo, and the high-resolution tri-snRNP part was manually adjusted in COOT. The pre-B$^{5′ss}$ complex was modelled by fitting individual components of the CE pre-B complex into the EM density as rigid bodies. PRP8, along with 5′ss oligo 1, was taken from the B-like complex and fit into the EM density as a rigid body. The side chains were initially truncated to a polyalanine chain owing to the relatively lower resolution, and the carbon backbones were manually adjusted in COOT. The side chains were then added back manually at the positions where the local resolution allows. The pre-B$^{5′ss+ATPγS}$) complex was modelled by fitting individual components of the B-like complex into the EM density. The high-resolution tri-snRNP part was adjusted in COOT. The crystal structure of the BRR2 helicase (PDB 4F91) was truncated into a polyalanine chain and docked into the density as a rigid body. The BRR2$^{CC}$ was not further adjusted, and the N-terminal cassette was manually adjusted into the density in COOT. The C-terminal part of SF3A1 (amino acids 496–521) was predicted by AlphaFold2 and docked into the density and adjusted in COOT. U6 nucleotides between the U6/5′ss helix and U4/U6 stem III (nucleotides 35–39), and U4 nucleotides between U4/U6 stem I and stem III (nucleotides 63–74) were de novo modelled in COOT. The pre-B$^{5′ssLNG+ATPγS}$ complex was modelled by fitting the pre-B$^{5′ss+ATPγS}$ complex into the EM density, and the flexible U4 snRNA strand (nucleotides 62–85) was deleted. The U4 Sm core was fit into the density as a rigid body. The extended U6/5′ss helix was modelled as a A-form helix and fit into the density. The pre-B$^{ATP}$ complex was modelled by fitting the pre-B$^{5′ss+ATPγS}$ complex into the EM density, and changing the 5′ss oligo sequence into the MINX exon sequence. A$^{-4}$ and C$^{-5}$ of the MINX exon were de novo modelled into the EM density in COOT. The PRP28 RecA domains were deleted owing to the absence of EM density at the corresponding position. The pre-B$^{AMPPNP}$ complex was modelled by fitting the CE pre-B complex into the EM density. The U1 Sm core, U1 snRNA and U1-70K were taken from the CI pre-B complex (PDB 6QX9) and docked into the EM density as a rigid body. The closed RecA domains of PRP28, together with the unknown single-stranded RNA, were modelled based on the crystal structure of the closed Mss116p DEAD-box helicase bound to AMP-PNP and a single-stranded RNA (PDB 3I5X). Coordinates of the tri-snRNP parts of the various complexes were refined in real space using PHENIX[36].

## Reporting summary

Further information on research design is available in the Nature Portfolio Reporting Summary linked to this article.

## Data availability

The cryo-EM maps and coordinates have been deposited into the Electron Microscopy Data Bank (EMDB) and the PDB as follows: pre-B protomer (EMD-18718; PDB 8QXD); B-like protomer (EMD-18781; PDB 8QZS); pre-B$^{5'ss}$ protomer at 0 °C (EMD-18788; PDB 8R0A); pre-B$^{5'ss}$ protomer at 30 °C (EMD-19847); pre-B$^{5'ss+ATPγS}$ protomer (EMD-18787; PDB 8R09); pre-B$^{5'ss+ATP}$ protomer (EMD-19848); pre-B$^{5'ssLNG+ATPγS}$ protomer (EMD-19349; PDB 8RM5); pre-B$^{ATP}$ protomer (EMD-18789; PDB 8R0B); pre-B$^{AMPPNP}$ protomer (EMD-18786; PDB 8R08); pre-B dimer (EMD-19594); B-like dimer (EMD-19595); pre-B$^{5'ss}$ dimer at 0 °C (EMD-19868); pre-B$^{5'ss+ATPγS}$ dimer (EMD-19596); pre-B$^{5'ssLNG+ATPγS}$ dimer (EMD-19597); pre-B$^{ATP}$ dimer (EMD-19598); the tri-snRNP core of pre-B (EMD-18544; PDB 8QP8); the tri-snRNP core of B-like complex (EMD-18548; PDB 8QPE); the tri-snRNP core of pre-B$^{5'ss}$ at 0 °C (EMD-18555; PDB 8QPK); the tri-snRNP core of pre-B$^{5'ss+ATPγS}$ (EMD-18542; PDB 8QOZ); the tri-snRNP core of pre-B$^{5'ssLNG+ATPγS}$ (EMD-18546; PDB 8QPA); the tri-snRNP core of pre-B$^{ATP}$ (EMD-18547; PDB 8QPB); the tri-snRNP core of pre-B$^{AMPPNP}$ (EMD-18545; PDB 8QP9); and the tri-snRNP core plus U1 snRNP of pre-B$^{AMPPNP}$ (EMD-18727).

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

**Acknowledgements** The authors thank T. Conrad for HeLa cell production in a bioreactor, H. Kohansal for preparing HeLa nuclear extract, and W. Lendeckel, M. Raabe and R. Pflanz for excellent technical assistance. This work was supported by funding from the Max Planck Society (to R. L.). H.U was supported by grants from the Deutsche Forschungsgemeinschaft (SFB860, project number 105286809 and SFB1565 project number 469281184). H.S. was supported by a grant from the Deutsche Forschungsgemeinschaft (SFB1565).

**Author contributions** Z.Z. prepared grids, collected EM data, performed EM data processing and refinement, and built the models. J.Z. helped refine the models. Z.Z. analysed the structure, with input from B.K., H.S. and R.L. V.K. purified and characterized all spliceosomal complexes, with initial help from S.E.J.L. O.D. and H.U. analysed protein–protein crosslinking and mass spectrometry data. All authors were involved in data interpretation. The manuscript was written by R.L., Z.Z. and C.L.W., with input from all authors. R.L. and H.S. supervised the project.

**Funding** Open access funding provided by Max Planck Society.

**Competing interests** The authors declare no competing interests.

**Additional information**
**Correspondence and requests for materials** should be addressed to Holger Stark or Reinhard Lührmann.

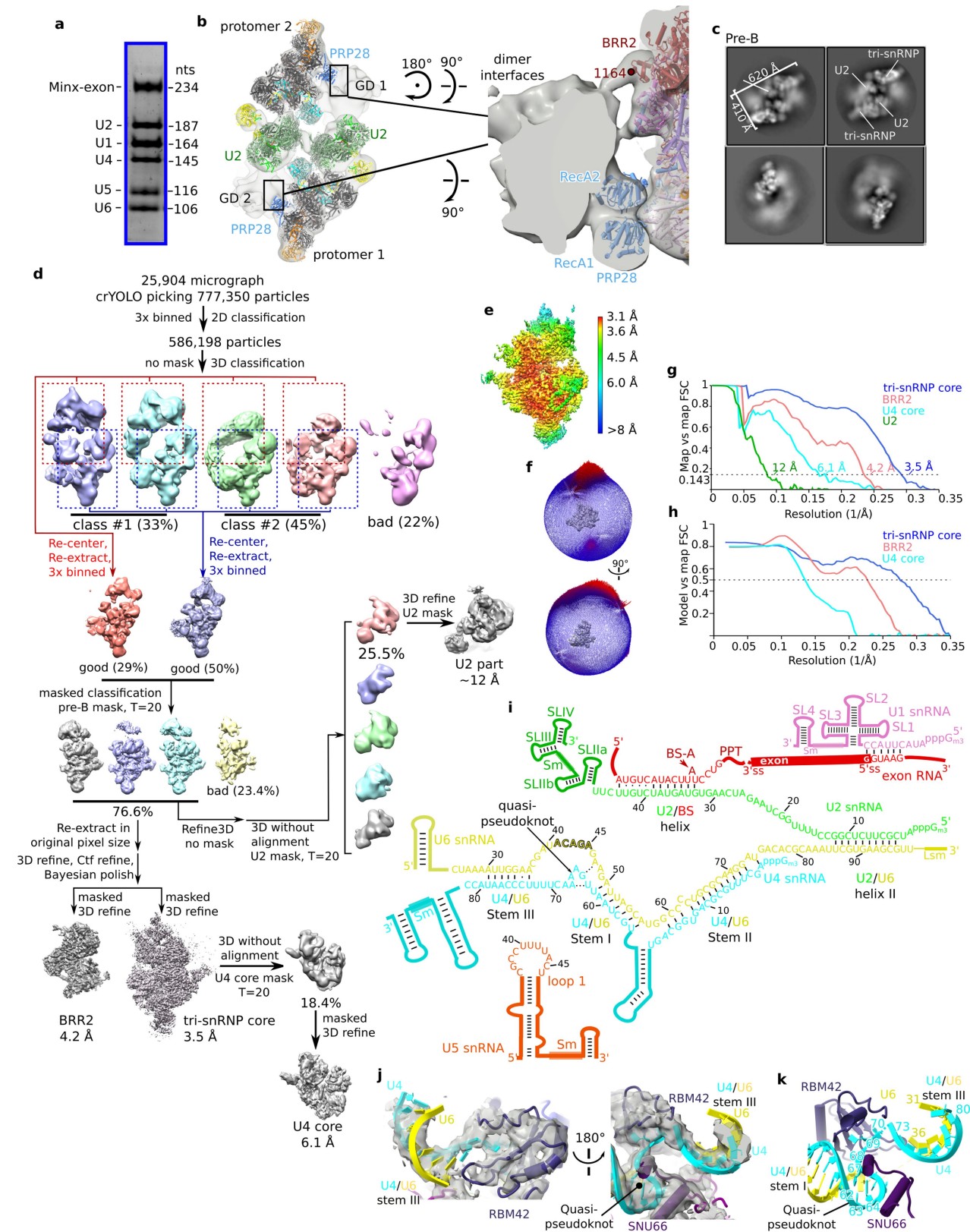

**Extended Data Fig. 1** | See next page for caption.

**Extended Data Fig. 1 | Cryo-EM and image-processing of the cross-exon pre-B complex. a**, RNA composition of purified CE pre-B complex dimers. For gel source data, see Supplementary Fig. 1. Pre-B complexes formed on the MINX exon RNA (Fig. 1a) were affinity-purified and RNA from fractions of the fastest sedimenting peak (typically fractions 14-18) was isolated, separated on a NuPAGE gel, and visualized by staining with SyBr gold. The nucleotide (nts) lengths of the snRNAs and MINX exon RNA are indicated on the right. The RNA composition was analysed from three independent pre-B complex purifications with similar results. The MINX exon RNA[7] was generated from the MINX pre-mRNA[37], which is a derivative of the Adenovirus Major Late (ADML) pre-mRNA. In the MINX exon RNA, the 5′ exon and adjacent downstream intron nucleotides of the MINX pre-mRNA have been deleted, leaving the 3′ exon (a truncated version of the ADML exon 2) and 64 nts of the 3′ end of the upstream, adjacent ADML intron, which contains an anchoring site followed by the branch site, polypyrimidine tract and 3′ss AG. A 5′ss was introduced at the 5′ end of the truncated exon, by adding the last 6 nts of the wildtype ADML exon 2 plus 22 nts of the adjacent downstream ADML intron, which are followed by a short linker and three RNA stem-loops that bind the MS2 protein (see Fig. 1a). Previous studies in our lab showed that cross-exon A-like complex formation on the MINX exon RNA is enhanced by the presence of the downstream 5′ss[7]. Furthermore, depletion of either U1 or U2 from the nuclear extract also led in each case to substantial reduction in the formation of the cross-exon A-like complex[7]. Thus, the complexes formed on the exon-containing substrate used in this study are exon-defined. **b**, Close up of the interfaces of the pre-B dimer. An expanded view of the boxed regions at the interfaces of the pre-B dimer is shown at the right. The two protomers contact each other via BRR2 and the RecA domains of PRP28 of one protomer, and the globular density of the other protomer, which contains U1 snRNP bound to the 5′ss and exon-binding proteins (see Extended Data Fig. 2). The functional relevance of these interfaces, as well as dimer formation in general, is currently not known. **c**, Representative cryo-EM 2D class averages of the pre-B dimers, where the top two represent class 1 dimers and the bottom two, class 2 dimers (see below). **d**, Cryo-EM computation sorting scheme. All major image-processing steps are depicted. For a more detailed explanation, see the EM data processing section in the Methods. Two major classes of the pre-B dimers are detected. In the class 2 dimer, the structure of only one pre-B complex is well-defined, whereas in class 1 both protomers are well-defined. The poorly resolved protomer in class 2 could potentially be a CE A-like complex. The tri-snRNP core is comprised of all tri-snRNP components excluding the U4 Sm core, U5 Sm core and BRR2. **e**, Local resolution estimation of the tri-snRNP region of the pre-B complex. **f**, Orientation distribution plot for the particles contributing to the reconstruction of the tri-snRNP region. **g**, Fourier shell correlation (FSC) values for the listed parts of the CE pre-B complex indicate a resolution of 3.5 Å for the tri-snRNP core, 4.2 Å for BRR2, 6.1 Å for the U4 core and 12 Å for the U2 snRNP. **h**, Map versus model FSC curves generated for the tri-snRNP core, BRR2 and U4 core regions of CE pre-B using PHENIX mtriage. **i**, Schematic of the RNA-RNA interaction network in the human CE pre-B protomer. The U1/5′ss base pairing interaction is inferred from previous biochemical characterization of CE pre-B complexes (previously denoted 37 S exon complexes)[7]. A dot between two nucleotides indicates that they do not base pair, but that a helix involving these nucleotides is formed (e.g., the extended U2/BS helix). **j-k**, U4/U6 stem III and the quasi pseudoknot are present in CE pre-B. Panel j, fit of the U4/U6 stem III and quasi-pseudoknot, as well as RBM42, to the CE pre-B EM density. Panel k, 3D molecular model corresponding to the right view in panel j.

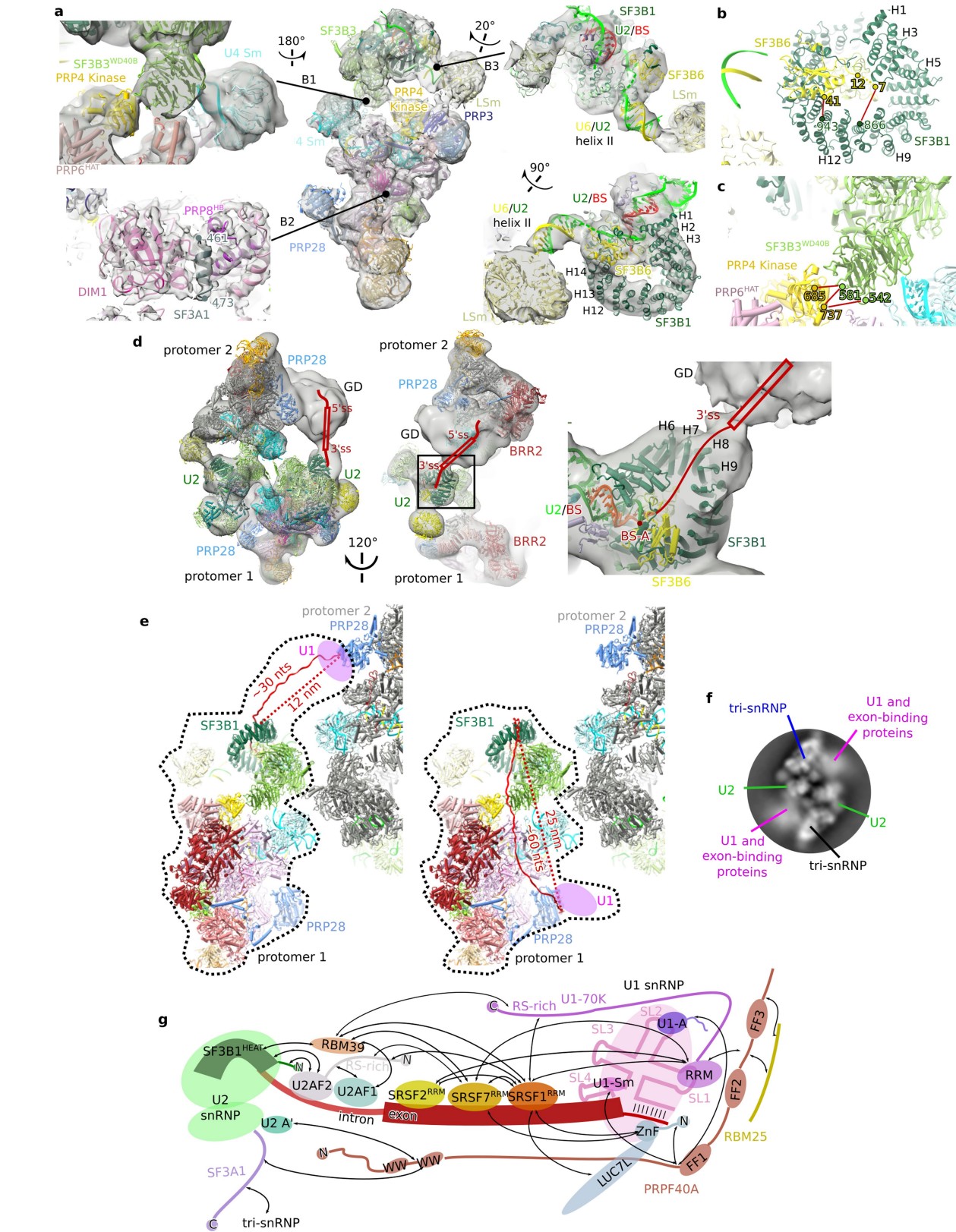

**Extended Data Fig. 2 |** See next page for caption.

**Extended Data Fig. 2 | U2 snRNP is connected to the U4/U6.U5 tri-snRNP in CE pre-B by three major bridges, and localization of U1 and the MINX exon.** **a**, Fit of the pre-B model into the EM density of one pre-B protomer. On the left and right, enlargements of the major U2-tri-snRNP bridges are shown. Upper left, PRP4 kinase (PRP4K) bridges SF3B3$^{WD40B}$ with the HAT repeats of the U5 protein PRP6. Lower left, interaction of an α-helix of SF3A1 (aa 461-473) with the U5 snRNP protein DIM1 and the PRP8 helical bundle (PRP8$^{HB}$). Right panels, two views of the U2/U6 helix II bridge with SF3B1 and SF3B6 at one end and the U6 Lsm ring at the other. In CI pre-B[18,19] the tri-snRNP and U2 snRNP are also connected via the U2/U6 helix II and the SF3A1-DIM1/PRP8 interaction. PRP4K is docked to the tri-snRNP via U4/U6 stem II and PRP6$^{HAT}$ in both complexes, but its interaction with U2 SF3B3$^{WD40B}$ appears to be more flexible in CI pre-B, as EM density connecting the two proteins is observed in only a subset of CI pre-B complexes[18]. Guided by crosslinks (see panel b) we can place the RRM domain of SF3B6 at the C-terminal HEAT repeats of SF3B1 directly adjacent to U2/U6 helix II, consistent with the idea that SF3B6 may help to facilitate the formation of U2/U6 helix II during the initial interaction of the tri-snRNP with the spliceosome, in both the cross-intron and cross-exon assembly pathways. **b**, Cross-links of SF3B6 with SF3B1. Crosslinked residues are depicted by circles connected by a reddish-brown line, where the numbers indicate the positions of the crosslinked amino acids. **c**, Crosslinks of SF3B3$^{WD40B}$ with PRP4K. Labeling as in panel b. **d**, Left and middle, U2 snRNP is connected to the poorly-resolved, globular EM density by a thin density element. Right, the latter protrudes from SF3B1$^{HEAT}$ at the position where the 3' end of the intron was previously shown to exit the HEAT domain. Thus, this density is predicted to contain intron nucleotides between the BS and 3'ss, and the adjacent, poorly-defined globular domain highly likely contains the MINX exon and bound SR proteins, as well as U1 snRNP. **e**, Distance constraints exclude an alternative pre-B protomer organization. The distance between U1 and U2 SF3B1$^{HEAT}$ in pre-B protomers organized as shown in Fig. 1 (left), and in an alternatively-organized protomer (right) are shown. For simplicity, only one of the U1 snRNPs is shown in each cartoon. Panel 2d shows that there is continuous density that links U2 SF3B1 of one promoter to the exon/U1 snRNP-containing globular density (GD) that is attached to PRP28 of the other protomer. Following this density, the distance between SF3B1 of one protomer and PRP28 of the other promoter would require ca 30 nts of RNA to cover. A similar RNA length would be required to reach the U1 bound to the 5'ss in the adjacent GD. Thus, our placement of the U1 snRNP is consistent with the 39 nt length of the MINX exon plus ca 10 nts of the PPT/3'ss that extend beyond SF3B1$^{HEAT}$. In the alternative organization, the same U2 snRNP, but instead the U1 from the other GD, would

bind to the same MINX exon RNA substrate (and thus in this case U1 would interact *in cis* with PRP28). This would require that after exiting SF3B1$^{HEAT}$, the remaining PPT/3'ss nts plus the MINX exon would wrap back across SF3B1$^{HEAT}$ and extend to PRP28 (and to the adjacent U1) within the same protomer. However, this would require more than 60 nts to cover this distance, without clashing with any tri-snRNP proteins, which is much longer than the length of our exon (39 nts) plus ca 10 nts of the PPT/3'ss that extend beyond SF3B1$^{HEAT}$. Alternative RNA paths that extend along the other side of the complex would be even longer. Therefore, an exon much longer than 60 nts would be required to form such an intra-protomer complex. Thus, in our CE pre-B complexes, PRP28 cannot interact within the same protomer with the U1 snRNP-bound 5'ss. **f**, Representative EM 2D class of the CE pre-B complex dimer. The "fuzzy" nature of the globular domains containing U1 and the exon binding proteins is likely due to the transient interaction of various SR proteins with the exon. **g**, Model of cross-exon, protein-protein interactions that indirectly bridge the U2 and U1 snRNPs in the CE pre-B complex based on protein crosslinking of the CE pre-B complex (see also Supplementary Table S2). Crosslinked residues are connected by black arrows. Although the cryo-EM structure of a cross-exon pre-A complex was recently reported[38], the nature of the bridge connecting the U2 and U1 snRNPs could not be discerned due to the poor resolution in this region of the complex. SR proteins are likely recruited to the defined exon by exonic splicing enhancers, as well as to the U1 snRNP[39], and have long been proposed to establish a network of protein-protein interactions across the exon that link the U2 and U1 snRNPs[40–42]. The number and identity of the SR proteins interacting with the exon likely varies from one cross-exon complex to the next[43]. Thus, there may be different combinations of SR proteins bound compared to those depicted in the model. The RRMs of SRSF1 crosslink with the U1-70K RRM, consistent with previous biochemical studies[44]. Likewise, RBM39 crosslinks to U2AF, and the LUC7L paralogs LUC7L2 and LUC7L3 (labeled LUC7L) crosslink to several SR proteins. The U1-related protein PRPF40A crosslinks to U1-70K, U1-A and LUC7L3, consistent with previous studies revealing similar interactions of PRP40 in the yeast U1 snRNP in early splicing complexes[45,46]. With the exception of a crosslink between SF3A1 and PRPF40A, crosslinks between U1 and U2 snRNP proteins are not observed, supporting the conclusion that U2 and U1 do not directly contact one another in cross-exon complexes, as previously proposed. The U1 snRNA stem-loop 4 was previously shown to interact with the U2 SF3A1 protein during the early stages of cross-intron spliceosome assembly[47]. However, it is not clear whether this RNA-protein interaction, which would directly connect U1 and U2 snRNP, contributes to the molecular bridge linking U1 and U2 in cross-exon complexes.

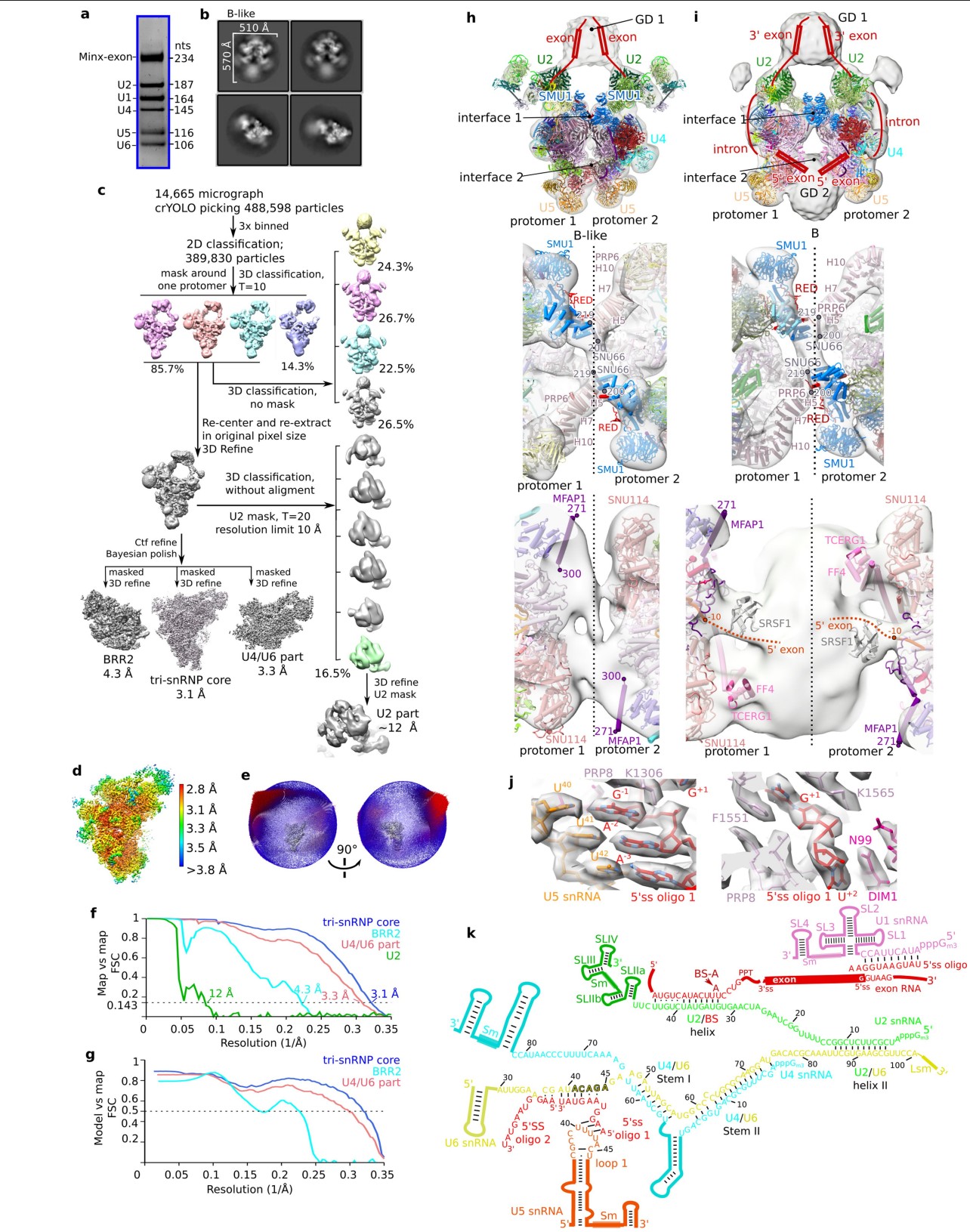

**Extended Data Fig. 3** | See next page for caption.

**Extended Data Fig. 3 | Cryo-EM and image-processing of the cross-exon B-like complex. a**, RNA composition of the purified B-like dimers. B-like complexes were affinity-purified and RNA from the fastest-sedimenting gradient peak was isolated, separated on a NuPAGE gel, and visualized by staining with SyBr gold. For gel source data, see Supplementary Fig. 1. The nucleotide (nts) lengths of the snRNAs and MINX exon RNA are indicated on the right. The RNA composition was analysed from four independent B-like complex purifications with similar results. U1 snRNA, which due to the presence of an excess of the 5′ss oligo should no longer be base paired to the 5′ss of the MINX exon RNA (see panel I) is still present, together with U2, U4, U5 and U6 snRNA, and the MINX exon RNA. It is very likely that, under our low salt purification conditions, U1 snRNP remains bound via protein-protein interactions presumably in the vicinity of the MINX 5′ss (i.e., in the upper poorly-resolved, globular EM density). Substantial amounts of U1 are also observed in all other complexes incubated in the presence of an excess of the 5′ss oligo (see below). **b**, Representative cryo-EM 2D class averages of the CE B-like dimers. **c**, Cryo-EM computation sorting scheme for CE B-like complexes. All major image-processing steps are depicted. The tri-snRNP core is comprised of all tri-snRNP components excluding the U4 Sm core, U5 Sm core and BRR2. The U4/U6 part consists of the U4/U6 stems I and II and associated proteins. **d**, Local resolution estimation of the tri-snRNP core region of the B-like complex. **e**, Orientation distribution plot for the particles contributing to the reconstruction of the tri-snRNP core region. **f**, Fourier shell correlation (FSC) values for the listed parts of the B-like complex, indicate a resolution of 3.1 Å for the tri-snRNP core, 4.3 Å for BRR2, 3.3 Å for the U4/U6 snRNP and 12 Å for the U2 snRNP. **g**, Map versus model FSC curves generated for the tri-snRNP core, BRR2 and U4/U6 regions of B-like using PHENIX mtriage. **h-i**, Structure of the CE B-like dimers (panel h) and CI B dimers[23] (panel i). Upper panels, overview of the 3D organization of the dimers. Middle and lower panels, expanded views of

interface 1 and 2, respectively. Note that the view of the interfaces is from the top of the complexes shown in the upper panels. As in the CI hB dimer[23], the B-like dimer is organized in a parallel manner with U5 located at the bottom of both protomers. Both dimers contain an upper, poorly defined globular density. Based on the position of the thin density bridges adjacent to SF3B1[HEAT], in both dimers the upper globular density appears to contain the MINX exon and associated proteins, and may additionally contain loosely-associated U1 snRNP. The two protomers in the B-like and B dimers are connected in the middle by an interface involving SMU1 and its binding partner RED, the PRP6 N-terminal HAT domains, and SNU66 α−helix[200-219]. The CI B dimers contain a large globular density element that is located at the very bottom, close to the exit site of the 5′ exon in each protomer, suggesting that this density element is comprised of the 5′ exons and associated proteins[23]. Consistent with this idea, CE B-like dimers, which lack an upstream 5′ exon, do not possess this large globular density. This supports the idea that the poorly-defined globular densities indeed are comprised of exons and exon-interacting proteins. In B-like, the lower interface 2 is comprised of unassigned density that separates SNU114 and MFAP1 from one protomer with SNU114 and MFAP1 from the other protomer. The functional relevance of these interfaces, as well as dimer formation in general, is currently not known. **j**, Fit of the nucleotides and protein side chains shown in Fig. 2b (left) and 2d (right) into the B-like EM density. **k**, Schematic of the RNA-RNA interaction network in the B-like complex. Two copies of the 5′ss-containing RNA oligonucleotide (5′ss oligo), which is added in excess, base pair not only with U1 (disrupting the U1/5′ss helix), but also with the U6 snRNA at/near the ACAGA box and with U5 snRNA loop 1. The resulting extended U6/5′ss helix is not as long as the *bona fide* extended U6/5′ss helix that is formed in the human B complex[19,22]. A dot between two nucleotides indicates that they do not base pair, but that a helix involving these nucleotides is formed (e.g., the extended U2/BS or U6/5′ss helices).

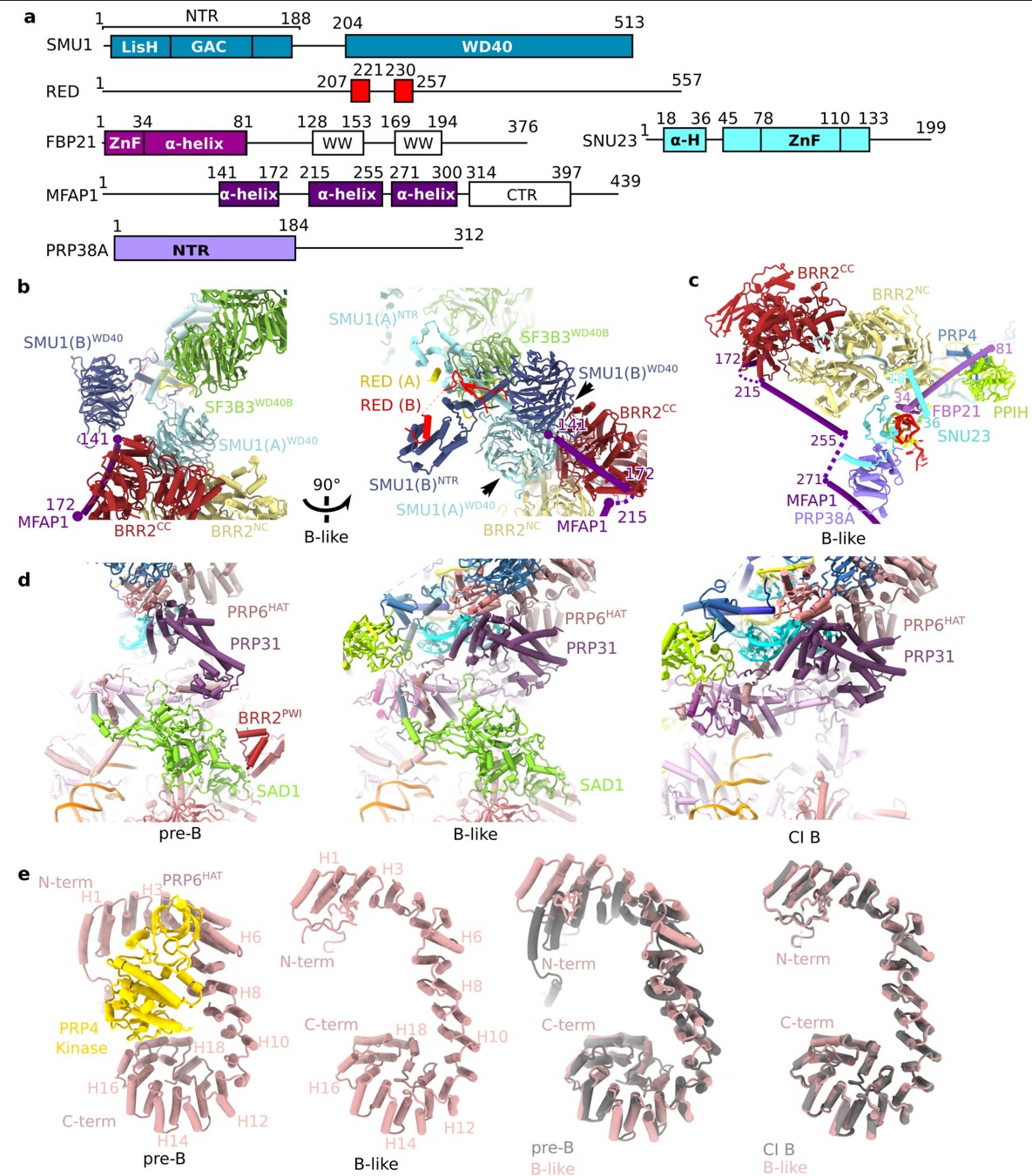

**Extended Data Fig. 4** | See next page for caption.

**Extended Data Fig. 4 | Spatial organization of B-specific proteins in B-like complexes, and structural comparisons of pre-B, B-like and CI B.**
**a**, Schematic of the domain organization of SMU1, RED, FBP21, SNU23, MFAP1 and PRP38A. Domains localized in the B-like cryo-EM structure are colored. NTR, N-terminal region; LisH, Lissencephaly type 1-like homology; GAC, globular α-helical core; WD40, β-propeller-like domain comprised of ca 40 amino acids that often contains a C-terminal tryptophan-aspartic acid (W-D) dipeptide; ZnF, Zinc Finger; WW; ca 40 amino acid-long domain containing two tryptophan residues; CTR, C-terminal region. **b**, U2 snRNP, whose molecular architecture is similar in B-like and B complexes, is stably-attached to the remodeled tri-snRNP, not only via U2/U6 helix II and interactions involving SF3A1, but also by the B-specific proteins SMU1 and RED. The latter form a the hetero-tetrameric SMU1-RED complex, whose domains are located at the same positions in the B-like and CI B complexes[23]. That is, the SF3B3-WD40B domain interacts with the WD40 domain of one of the SMU1 subunits (SMU1-B$^{WD40}$) and also the SMU1 NTRs, while SMU1-A$^{WD40}$ is also docked to BRR2 at the interface between both helicase domains. As in the CI human B complex, SMU1 forms a homodimer that forms primarily via the interaction of the LisH domain in the NTR of each SMU1 molecule, and each SMU1 additionally interacts with one copy of the RED protein[23]. The identity of the individual SMU1 NTR domains (A or B) cannot be determined unambiguously. **c**, In addition to SMU1, several other B-specific proteins, including FBP21, SNU23 and MFAP1, interact with BRR2 at its new position in B-like, as they do in the CI B complex[23]. Aside from a potential role in helping to tether BRR2 to its new activation position, several of these proteins may thus additionally, or instead, regulate BRR2 activity in both B-like and B complexes. Our data do not directly implicate the B-specific proteins in regulating alternative splicing events. However, as our studies indicate that splice site pairing occurs at the B complex stage, proteins that play a role in B complex formation (which include the B-specific proteins) are clearly regulatory candidates. Indeed, RNA-mediated knockdowns of several B-specific proteins have revealed that they modulate multiple alternative splicing events in the cell. For example, Papasaikas et al showed that knockdown of SMU1 and RED leads to alternative splice site usage and exon skipping[48]. In addition, the *C. elegans* homologue of MFAP1 was shown to affect alternative splicing[49]. **d**, Comparison of the position of PRP31 and SAD1 in CE pre-B (left), B-like (middle) and CI B (PDB 6AHD)[19] (right) complexes. The structures are aligned via PRP8$^{NTD}$. Surprisingly, SAD1, which is displaced in the CI B complex concomitant with the translocation of the BRR2 helicase domain from PRP8$^{RT}$ to PRP8$^{EN19,22}$, is still stably bound to SNU114 and PRP8$^{RT}$ in the B-like complex. This appears to be due to structural differences in B versus B-like, and might arise due to the slightly different conformation of PRP8$^{En}$ because a 5' exon binding channel is not formed by UBL5 and MFAP1$^{CTR}$ in B-like complexes due to the absence of a 5' exon. **e**, Conformational change in the PRP6 HAT repeats during the conversion of the CE pre-B complex (left) into a B-like complex (middle left). Overlays of the PRP6 HAT repeats in CE pre-B versus B-like (middle right) and B-like versus CI B (right) (PDB 6AHD)[19]. The structures were aligned via the C-terminal HAT repeats.

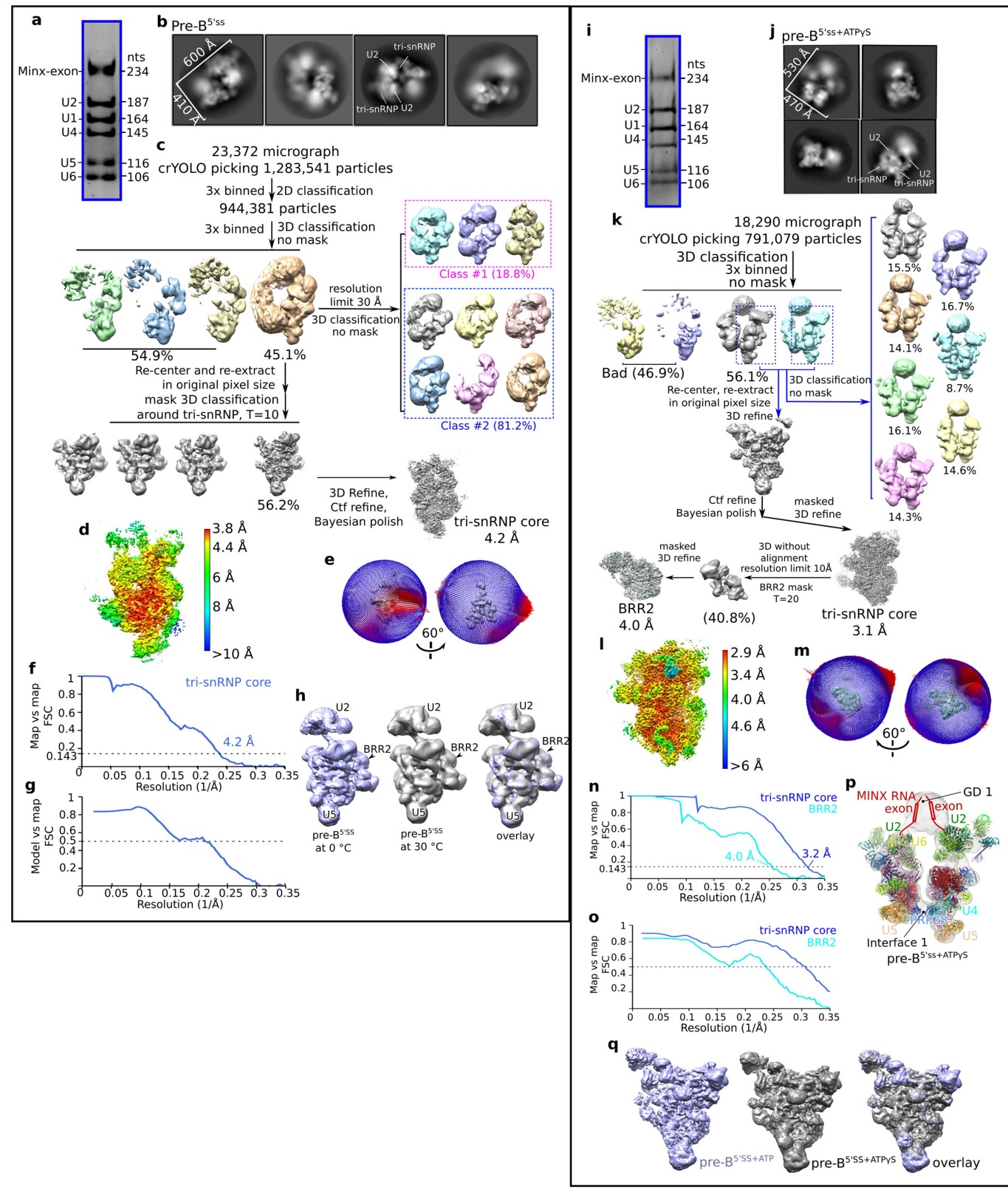

**Extended Data Fig. 5 | See next page for caption.**

**Extended Data Fig. 5 | Cryo-EM and image-processing of the pre-B$^{5'ss}$ and pre-B$^{5'ss+ATPγS}$ complexes. a**, RNA composition of pre-B$^{5'ss}$ complexes. Purified pre-B complexes were incubated with an excess of the 5'ss RNA oligonucleotide and subjected to glycerol gradient centrifugation. RNA from the fastest-sedimenting gradient peak was isolated, separated on a NuPAGE gel, and visualized by staining with SyBr gold. For gel source data, see Supplementary Fig. 1. The nucleotide (nts) lengths of the snRNAs and MINX exon RNA are indicated on the right. The RNA composition was analysed from three independent pre-B$^{5'ss}$ purifications with similar results. **b**, Representative cryo-EM 2D class averages of the pre-B$^{5'ss}$ dimers. **c**, Cryo-EM computation sorting scheme for pre-B$^{5'ss}$ complexes. All major image-processing steps are depicted. Two major classes of the pre-B$^{5'ss}$ dimers are detected, which are organized in an antiparallel manner and are highly similar to the class 1 and class 2 dimers of the CE pre-B complex (Fig. 1a and Extended Data Fig. 1). In the class 2 dimer, the structure of only one pre-B complex is well-defined, whereas in class 1 both protomers are well-defined. **d**, Local resolution estimation of the tri-snRNP core region of the pre-B$^{5'ss}$ complex. **e**, Orientation distribution plot for the particles contributing to the reconstruction of the tri-snRNP core region. **f**, Fourier shell correlation (FSC) values indicate a resolution of 4.2 Å for the tri-snRNP core in pre-B$^{5'ss}$. **g**, Map versus model FSC curves generated for the tri-snRNP core region of pre-B$^{5'ss}$. **h**, Comparison of the structure of pre-B$^{5'ss}$ complexes formed by incubating purified pre-B with the 5'ss oligo at 0 °C or 30 °C. An overlay of the EM densities (low-pass filtered to ~20 Å resolution) is shown on the right. Pre-B$^{5'ss}$ complexes with a nearly identical 3D structure were obtained if incubation with the 5'ss oligo was carried out at 0 °C or 30 °C. **i**, RNA composition of pre-B$^{5'ss+ATPγS}$ complexes. Purified pre-B complexes were incubated with an excess of the 5'ss RNA oligonucleotide followed by ATPγS, and then subjected to glycerol gradient centrifugation. RNA from the fastest-sedimenting gradient peak was isolated, separated on a NuPAGE gel, and visualized by staining with SyBr gold. For gel source data, see Supplementary Fig. 1. The nucleotide (nts) lengths of the snRNAs and MINX exon RNA are indicated on the right. The RNA composition was analysed from two independent pre-B$^{5'ss+ATPγS}$ purifications with similar results. **j**, Representative cryo-EM 2D class averages of the pre-B$^{5'ss+ATPγS}$ dimers. **k**, Cryo-EM computation sorting scheme. All major image-processing steps are depicted. **l**, Local resolution estimation of the tri-snRNP core region of the pre-B$^{5'ss+ATPγS}$ complex. **m**, Orientation distribution plot for the particles contributing to the reconstruction of the tri-snRNP core region. **n**, Fourier shell correlation (FSC) values indicate a resolution of 3.1 Å for the tri-snRNP core and 4.0 Å for BRR2. **o**, Map versus model FSC curves generated for the tri-snRNP core and BRR2 regions of pre-B$^{5'ss+ATPγS}$. **p**, Fit of two monomeric pre-B$^{5'ss+ATPγS}$ complex molecular models into the EM density of the pre-B$^{5'ss+ATPγS}$ dimer. As in B-like, the two pre-B$^{5'ss+ATPγS}$ complexes are organized in a parallel manner, with their U5 Sm cores at the bottom. In addition, there is an upper globular density that, like in the B-like dimers, likely contains the exons and associated proteins. However, consistent with the absence of the B-specific proteins, the middle interface present in the B-like dimers that is formed in part by SMU1 and RED, is missing. Furthermore, the lower bridge that contained MFAP1 in B-like complexes, now appears to form mainly between the PRP28 helicase of each protomer. **q**, Comparison of the structure of complexes formed by incubating purified pre-B with the 5'ss oligo plus ATP (pre-B$^{5'ss+ATP}$) or ATPγS (pre-B$^{5'ss+ATPγS}$). An overlay of their EM densities is shown on the right, confirming that they possess a highly similar structure.

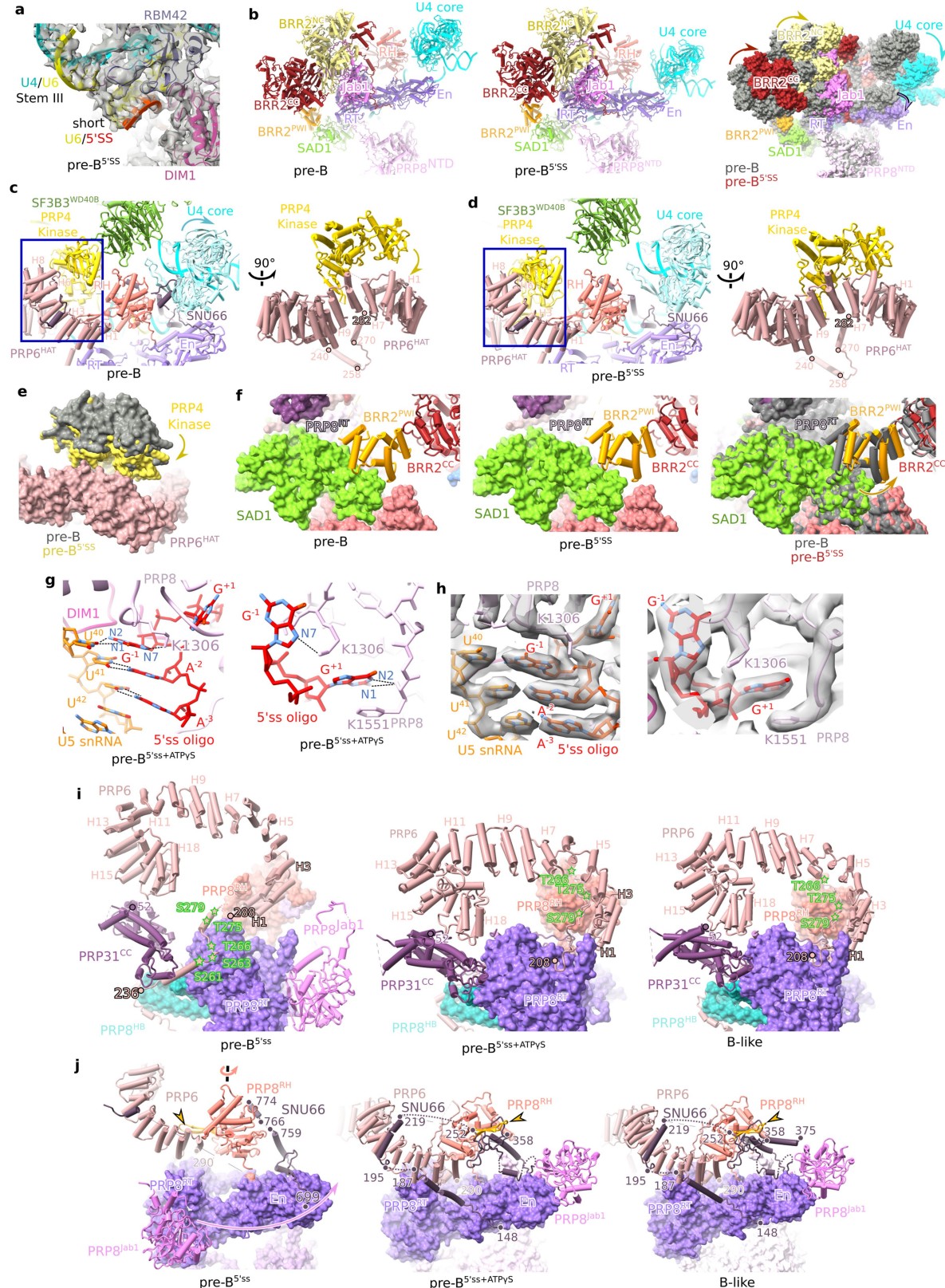

**Extended Data Fig. 6 |** See next page for caption.

**Extended Data Fig. 6 | Structural changes observed after addition of the 5'ss oligo alone or the 5'ss oligo plus ATPγS. a,** Fit of a single 5'ss oligo to the pre-B$^{5ss}$ EM density. **b,** The movement of the PRP8$^{RT/En}$ domain toward the PRP8$^{NTD}$ upon addition of the 5'ss oligo leads to concomitant movements of other tri-snRNP components that interact with PRP8$^{RT/En}$. Comparison of the molecular architecture of BRR2, PRP8, SAD1 and the U4 Sm core domain in pre-B (left) and pre-B$^{5ss}$ complexes (middle). Right, overlay of the corresponding surface representations in pre-B (grey) versus pre-B$^{5ss}$ (various colors), with arrows indicating the concomitant movements of the indicated domains with PRP8$^{RT/En}$. These movements do not involve any major structural changes such as translocation of BRR2 to its activation position. **c,d,** Comparison of the molecular architecture of PRP4 kinase, SF3B3, PRP6$^{HAT}$, SNU66 and the U4 Sm core domain in pre-B (panel c, left) and pre-B$^{5ss}$ complexes (panel d, left). The boxed regions are expanded and rotated 90° in the corresponding panels shown at the right. Aligned via the PRP8$^{RT/En}$ domain. In pre-B$^{5ss}$, PRP4K and SF3B3$^{WD40B}$ are located further apart, and PRP4K is more closely associated with PRP6$^{HAT}$. Moreover, the U4 Sm core has moved away from PRP8$^{RH}$ in pre-B$^{5ss}$ (as indicated by the arrow in panel c). **e,** Overlay of pre-B and pre-B$^{5ss}$ showing differences in the position of PRP4K. **f,** In pre-B$^{5ss}$, BRR2$^{PWI}$ is detached from SAD1. A comparison of the molecular architecture of SAD1, BRR2$^{CC}$ and BRR2$^{PWI}$ in pre-B (panel f, left) and pre-B$^{5ss}$ complexes (panel f, middle). An overlay of the

structures is shown in panel f (right); alignment on PRP8$^{NTD}$. **g,** Interactions of the 5'ss-containing RNA oligonucleotide (5'ss oligo) with U5 snRNA loop 1 nucleotides, and residues of PRP8 and DIM1 in pre-B$^{5ss+ATPγS}$. **h,** Fit of the nucleotides and protein side chains shown in panel g into the pre-B$^{5ss+ATPγS}$ EM density. **i,** Rearrangements in the NOP and adjacent coiled-coil domains of PRP31 (aa 52-331) and repositioning of the PRP6 HAT domain and its more N-terminal phosphorylated region upon addition of ATP to pre-B$^{5ss}$ complexes. The coiled/coil (CC) domains of PRP31 rotate by ca 45° relative to the PRP31 NOP domain after addition of ATP. A comparison of the molecular architecture of the indicated proteins/protein domains in pre-B$^{5ss}$ (left), pre-B$^{5ss+ATPγS}$ (middle) and B-like complexes (right). Serine and threonine residues of PRP6 that could be modelled previously[18,19] and that were previously shown to be phosphorylated by PRP4K[26] are indicated by green stars. The region of PRP31 that is known to be phosphorylated could not be modelled. **j,** ATP addition to pre-B$^{5ss}$ leads to the ca 180° rotation of PRP8$^{RH}$, which is coordinated with the repositioning of BRR2/PRP8$^{Jab1}$, PRP6 and SNU66, among others. The β-hairpin loop of PRP8$^{RH}$ is indicated in yellow and by an arrowhead. The SNU66 domain comprised of aa 252-358 is structured first in pre-B$^{5ss+ATPγS}$ where it is docked to the β-hairpin loop region of the repositioned PRP8$^{RH}$ domain (compare left and middle panels). The C-terminal SNU66$^{630-774}$ domain, which binds across SNRP27K and the U4 Sm core in pre-B$^{5ss}$, is also repositioned in pre-B$^{5ss+ATPγS}$.

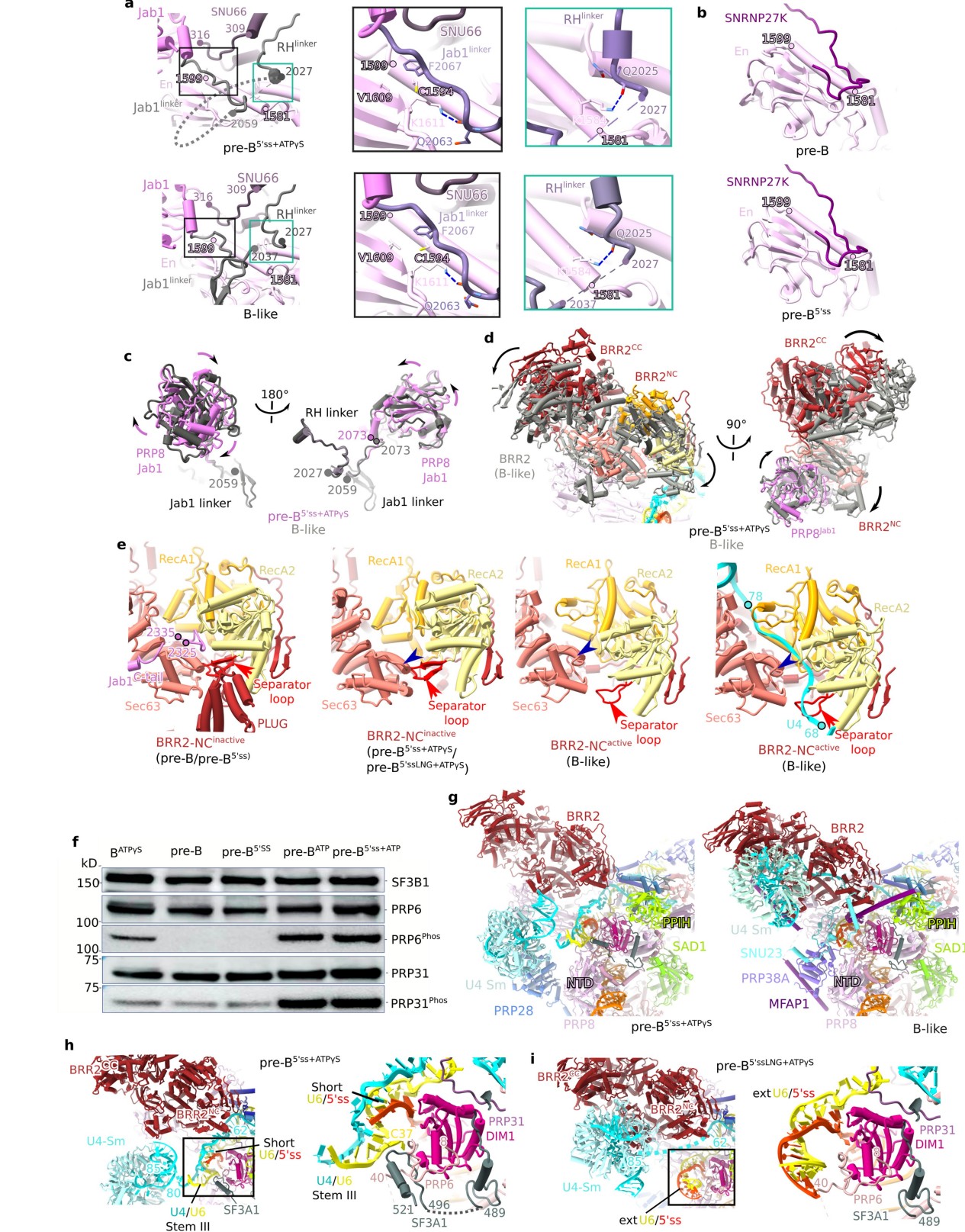

**Extended Data Fig. 7** | See next page for caption.

**Extended Data Fig. 7 | Structural differences between pre-B^5'ss, pre-B^5'ss+ATPγS and B-like complexes. a**, BRR2 is anchored to its new position in pre-B^5'ss+ATPγS (top) and the B-like complex (bottom) via the PRP8^RH-PRP8^Jab1- linker whose ends are now stably bound to PRP8^En. The boxed regions are shown in expanded form at the right. Jab1 linker, the region of the RH-Jab1 linker proximal to the PRP8 Jab1 domain. RH linker, the region of the RH-Jab1 linker proximal to the PRP8 RH domain. **b**, Interaction of SNRNP27K with PRP8^En in the pre-B (top) and pre-B^5'ss (bottom) complexes. **c**, Rotation of PRP8^Jab1 around the RH-Jab1 linker is required to reach its B-like position. An overlay shows the position of the Jab1 domain in pre-B^5'ss+ATPγS (purple) and B-like (grey). PRP8^Jab1 appears to rotate around the amino acids of the RH-Jab1 liker directly upstream of the Jab1 domain, which may act as a hinge. **d**, Two different views of an overlay of BRR2 in pre-B^5'ss+ATPγS (multi-colored) and B-like (grey), showing that BRR2 in pre-B^5'ss+ATPγS must still rotate in order to reach its position in the B-like complex. **e**, Stepwise opening of the BBR2^NC. In pre-B and pre-B^5'ss (left panel), the opening of the RNA binding channel is blocked by the BRR2 PLUG domain and C-terminal (CT) tail of the PRP8 Jab1 domain. In pre-B^5'ss+ATPγS, pre-B^5'ssLNG+ATPγS and the B-like complex (middle and right panels), the BRR2^PLUG and PRP8 Jab1 CT tail are displaced. However, in pre-B^5'ss+ATPγS, pre-B^5'ssLNG+ATPγS (middle left panel), the separator loop (marked by a red arrowhead) of the RecA2 domain contacts an α-helix of the Sec63 domain (blue arrowhead), blocking the RNA binding channel. Thus, in the absence of the B-specific proteins, a new intermediate BRR2^NC state (i.e., with a free RNA channel entry site but a blocked RNA channel) is observed. Due to the presence of the U4 Sm domain, U4 snRNA cannot be threaded into the RecA domains. Instead, the latter must to be opened to allow U4 snRNA binding. Downward movement of the RecA1 and RecA2 domains of BRR2^NC and the concomitant movement of the separator loop, positions it further away from the BRR2^NC Sec63 domain, leading to an open RNA binding channel (middle right panel), allowing the binding of the single-stranded region of U4 snRNA (right panel) (See also Supplementary Video 6). **f**, PRP6 and PRP31 are phosphorylated in pre-B^ATP and pre-B^5'ss+ATP complexes. Proteins were isolated from the indicated complexes and the phosphorylation status of PRP6

and PRP31 determined by immunoblotting with antibodies specific for phosphorylated PRP6 or PRP31. The less intense signals in the B complex lane obtained with the anti-phosphorylated PRP6 and PRP31 antibodies is due to the fact that these proteins are thiophosphorylated in the B complexes, which were isolated in the presence of ATPγS, and thiophosphorylated proteins are poorly recognized by the anti-phospho antibodies. For blot source data, see Supplementary Fig. 1. Similar blot results where obtained with two independent experiments. **g**, Comparison of the position of BRR2 and the U4 Sm core in pre-B^5'ss+ATPγS (left) compared to the B-like complex (right). The structures are aligned via the PRP8^NTD. In pre-B^5'ss+ATPγS, the BRR2 helicase domain has not reached its final position. In addition, BRR2^NC exhibits a closed, inactive conformation and has not yet bound the U4 snRNA, and the U4 Sm core domain has also not reached its final, B-like position. Although BRR2's PLUG and PWI domains are not visible, the N-terminal ca 50 amino acids of BRR2 still wrap around PRP8^Large at the same position as in the pre-B^5'ss complex. These data strongly support the idea that during the translocation of BRR2's helicase domain across the large domain of PRP8, BRR2 does not dissociate and subsequently reassociate, but rather remains attached via its N-terminal region to the spliceosome. **h**, BRR2 is translocated, but the U4 Sm core is not docked to BRR2 in pre-B^5'ss+ATPγS. Right, zoomed-in view of the boxed region. U6-C37 is inserted into a protein pocket formed by SF3A1 aa 496-521 and PRP6, stabilizing the new position of U4/U6 stem III. Tethering of U6-C37 by SF3A1 and PRP6, may help to stabilize the short U6/5'ss helix after release of RBM42, which appears to contact U6 near C37, and after the repositioning of U4/U6 helix III, which prior to its movement likely constrains the position of U6 nucleotides in this region. In addition, capture of this U6 nucleotide fixes the path of the adjacent U6 snRNA region, which may facilitate the subsequent formation of the extended U6/5'ss helix upon disruption of U4/U6 stem III and release of U6-C37. **i**, The U4 Sm core contacts BRR2 in pre-B^5'ssLNG+ATPγS, and U4/U6 stem III is dissociated by binding of the long 5'ss oligo to U6 snRNA. Right, zoomed-in view of the boxed region. An extended (ext) U6/5'ss helix is formed, and aa 496-521 of SF3A1 are destabilized.

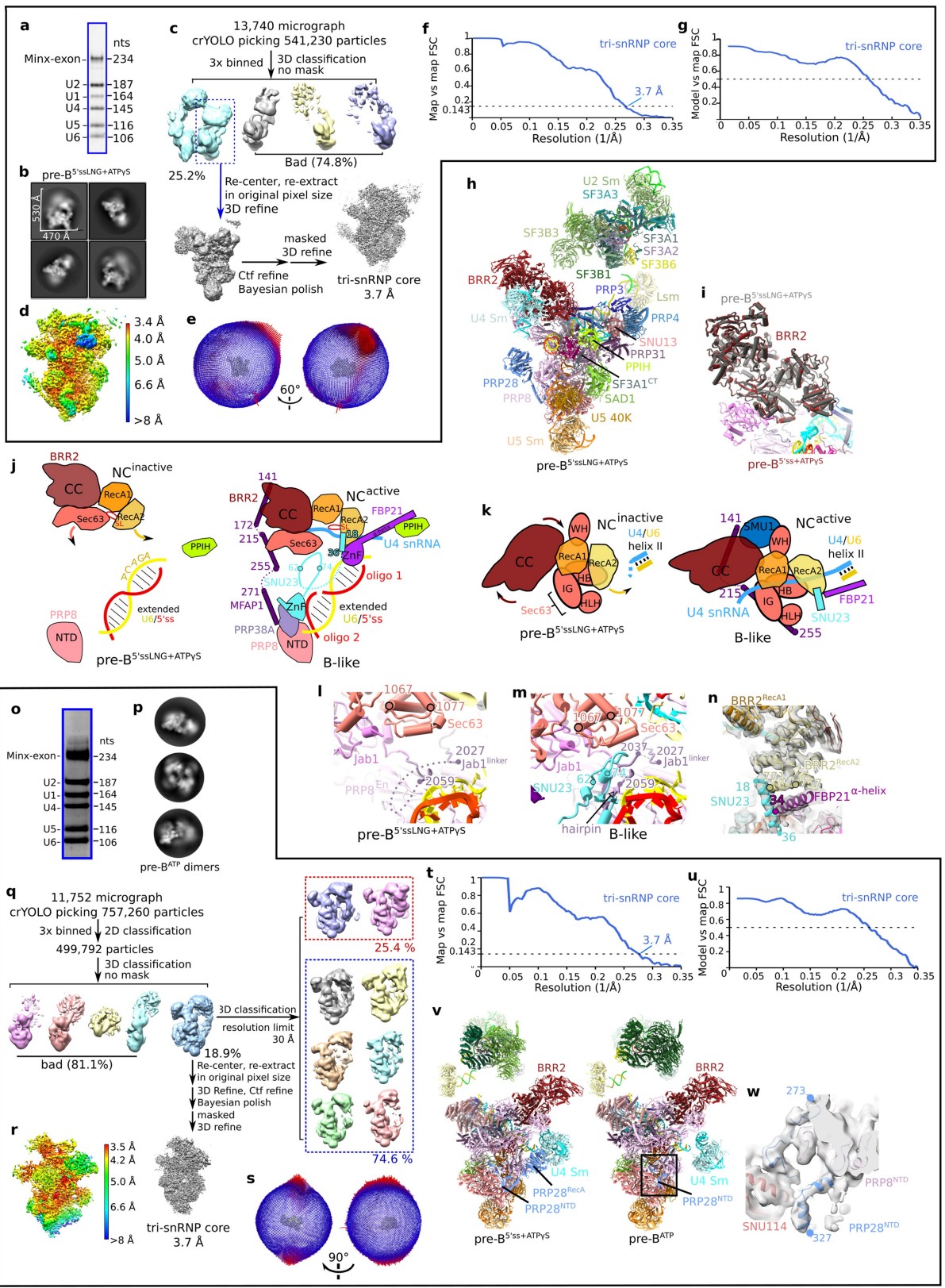

**Extended Data Fig. 8** | See next page for caption.

**Extended Data Fig. 8 | Cryo-EM of the pre-B$^{5'ssLNG+ATPγS}$ and pre-B$^{ATP}$ complexes and structural comparisons with the B-like complex. a**, RNA composition of pre-B$^{5'ssLNG+ATPγS}$ complexes. Purified pre-B complexes were incubated with an excess of the long 5'ss RNA oligonucleotide followed by ATPγS, and then subjected to glycerol gradient centrifugation. RNA from the fastest-sedimenting gradient peak was isolated, separated on a NuPAGE gel, and visualized by staining with SyBr gold. For gel source data, see Supplementary Fig. 1. The nucleotide (nts) lengths of the snRNAs and MINX exon RNA are indicated on the right. The RNA composition was analysed twice from the same pre-B$^{5'ssLNG+ATPγS}$ purification with similar results. **b**, Representative cryo-EM 2D class averages of the pre-B$^{5'ssLNG+ATPγS}$ dimers. All "good" particles show dimerized complexes, in which both protomers are well-defined and aligned in a parallel manner. **c**, Cryo-EM computation sorting scheme. **d**, Local resolution estimation of the tri-snRNP core region of the pre-B$^{5'ssLNG+ATPγS}$ complex. **e**, Orientation distribution plot for the particles contributing to the reconstruction of the tri-snRNP core region. **f**, Fourier shell correlation (FSC) values indicate a resolution of 3.7 Å for the tri-snRNP core of pre-B$^{5'ssLNG+ATPγS}$. **g**, Map versus model FSC curves generated for the tri-snRNP core region of pre-B$^{5'ssLNG+ATPγS}$. **h**, Molecular architecture of the pre-B$^{5'ssLNG+ATPγS}$ complex. **i**, Overlay of BRR2 in pre-B$^{5'ssLNG+ATPγS}$ (grey) and pre-B$^{5'ss+ATPγS}$ (brown), revealing that the position of the BRR2 helicase domain does not change substantially upon addition of the long 5'ss oligonucleotide. Thus, the BRR2 helicase domain must still be rotated to reach its B-like position, even though the U4/U6 helix III is dissolved and an extended U6/5'ss helix has formed. Aligned via PRP8$^{NTD}$. **j**, Cartoon showing that the movement and opening of BRR2 are facilitated by SNU23 loop 62-74 and MFAP1 helix 215-255, which are missing in pre-B$^{5'ssLNG+ATPγS}$ but bind in B-like to the BRR2$^{NC}$ Sec63 domain, and also by the binding of FBP21's long α-helix and SNU23's α-helix 18-36, which interact both with RecA2 of BRR2$^{NC}$. SNU23 also tethers BRR2$^{NC}$ to the PRP8$^{NTD}$, bridging it to the U6/5'ss helix. **k**, Rotation of BRR2 towards PRP8$^{NTD}$ is likely further facilitated by the SMU1/RED complex which stabilizes the new position of BRR2 in the B-like complex. **l-n**, Zoomed-in view of the structure of the PRP8 RH-Jab1 linker (designated Jab1$^{Linker}$) in pre-B$^{5'ssLNG+ATPγS}$ (l) versus B-like (m) complexes, and the interaction of SNU23 with BRR2$^{SEC63}$ and the Jab1$^{Linker}$, leading to the stabilization of a hairpin structure (aa 2037-2058) in the linker in B-like. **n**, An α-helix of RecA2 of the BRR2$^{NC}$ is contacted by both SNU23 α-helix 18-36 and by the long α-helix of FBP21 in B-like. Most of the binding sites of the B-specific proteins are created during or after the translocation of BRR2 to its pre-activation position. For example, MFAP1 and SNU23 are able to interact with BRR2 helicase elements first after BRR2 translocation to its pre-final site. In addition, a binding site for FBP21 is generated by PPIH and an N-terminal region of PRP4 first upon their stabilization during BRR2 translocation/remodeling. **o**, RNA composition of pre-B$^{ATP}$ complexes. As in all other complex purifications, described above, the MINX exon RNA remains intact during incubation of pre-B complexes with ATP. Affinity-purified pre-B complexes were incubated with ATP and then subjected to glycerol gradient centrifugation. RNA from the fastest-sedimenting gradient peak was isolated, and separated on a NUPAGE gel and visualized by staining with Sybr gold. For gel source data, see Supplementary Fig. 1. The nucleotide (nts) lengths of the snRNAs and MINX exon RNA are indicated on the right. The RNA composition was analysed from two independent pre-B$^{ATP}$ purifications with similar results. **p**, Representative cryo-EM 2D class averages of the pre-B$^{ATP}$ dimers. **q**, Cryo-EM computation sorting scheme. **r**, Local resolution estimation of the tri-snRNP core region of the pre-B$^{ATP}$ complex. **s**, Orientation distribution plot for the particles contributing to the reconstruction of the tri-snRNP core region. **t**, Fourier shell correlation (FSC) values indicate a resolution of 3.7 Å for the tri-snRNP core of pre-B$^{ATP}$. **u**, Map versus model FSC curves generated for the tri-snRNP core region of pre-B$^{ATP}$. **v**, The molecular architecture of the pre-B$^{5'ss+ATPγS}$ (left) and the pre-B$^{ATP}$ (right) complex is very similar. One notable exception is that the PRP28 RecA domains are destabilized in pre-B$^{ATP}$, consistent with PRP28 action. In pre-B$^{5'ss+ATPγS}$, a previously unknown intermediate, spliceosome assembly state in which the U4 Sm core and U4/U6 stem III have moved about 5 nm downwards from their location in pre-B is observed. This intermediate position of U4/U6 stem III is not only observed in pre-B$^{5'ss+ATPγS}$ complexes, but also in pre-B$^{ATP}$ complexes, where it is also stabilized by an interaction between U6-C37 and SF3A1 and PRP6$^{NTR}$. This suggests that the transient docking of stem III at this position also occurs when the U6/5'ss helix is formed by the PRP28-mediated interaction of the U6 ACAGA box with a *bona fide* 5'ss. **w**, Fit of the PRP28$^{NTD}$ into the pre-B$^{ATP}$ EM density (expanded view of the boxed region in panel h). In pre-B$^{ATP}$ aa 273 to 327 of the PRP28$^{NTD}$ remain associated with PRP8$^{NTD}$ presumably due to the absence of MFAP1 whose CTR binds in a mutually exclusive manner with the PRP28$^{NTD}$.

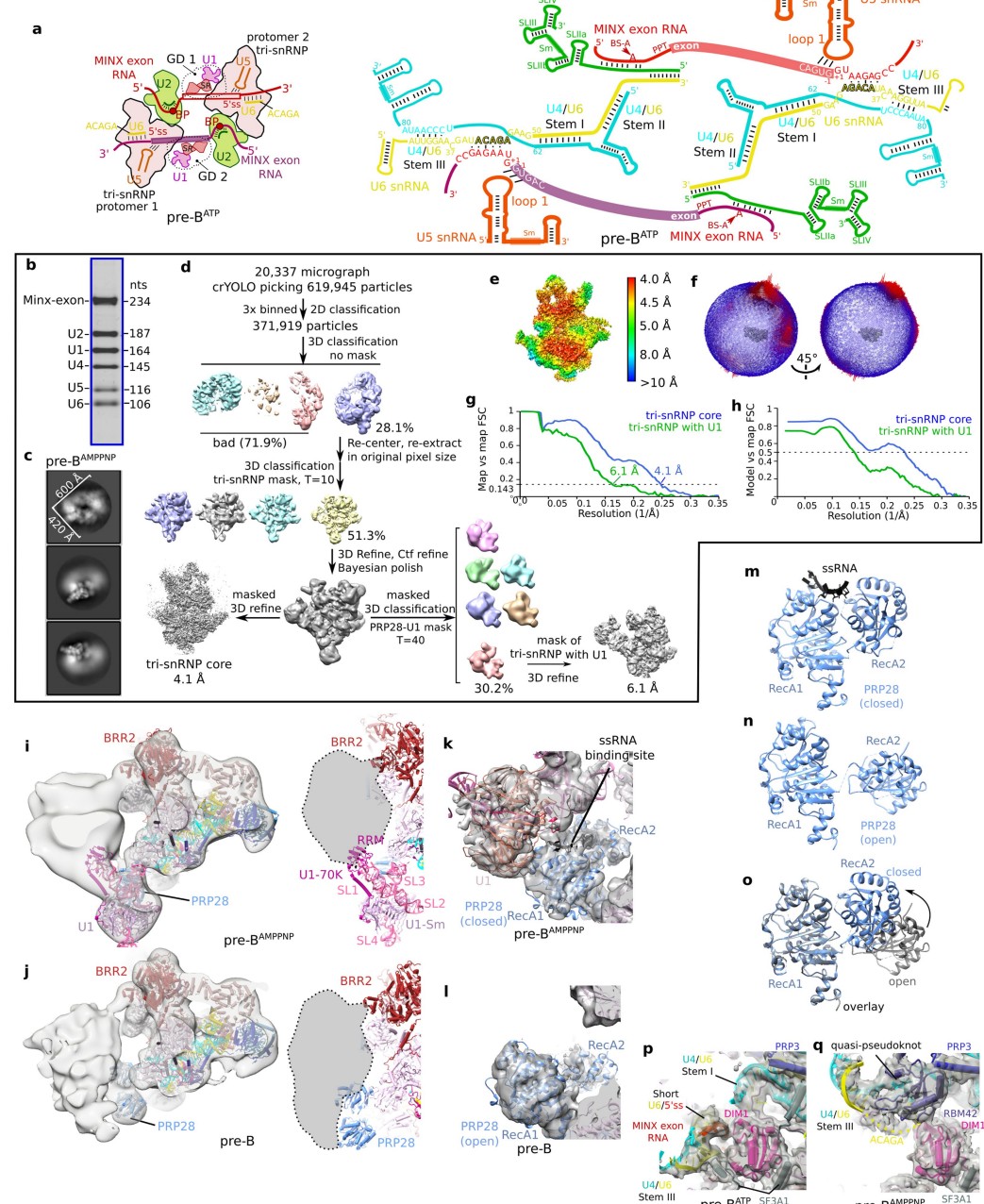

**Extended Data Fig. 9** | See next page for caption.

**Extended Data Fig. 9 | Cryo-EM and image-processing of the pre-B^AMPPNP complex, and localization of the U1 snRNP. a**, Cartoon showing organization of the protomers in the pre-B^ATP dimers (left). Right, Schematic of the RNA-RNA network in the pre-B^ATP dimer. The protomers in the pre-B (Fig. 1a) and pre-B^ATP dimers, are organized in an anti-parallel manner, in contrast to the parallel organization of the protomers in the pre-B^5'ss+ATPγS dimers (Extended Data Fig. 5p). Addition of ATP and the 5'ss oligo, thus not only triggers rearrangements within the protomers, foremost the large-scale translocation of BRR2, but also a reorganization of the protomers relative to each other from an anti-parallel orientation to a parallel one. Although the mechanism for this reorganization is not clear, in particular whether it involves partial or complete detachment of the protomers comprising the dimer, it would in both cases likely require the dissociation of protein-protein contacts involving the PRP28 region of the tri-snRNP of one protomer and components of the globular domain encompassing the exon of the adjacent protomer, such as SR proteins and U1 snRNP. During formation of pre-B^5'ss+ATPγS, the 5'ss oligo disrupts the U1 base pairing interaction with the 5'ss of the MINX exon RNA and at the same time prevents U6 from interacting with the latter, freeing up the MINX exon RNA. In contrast, in pre-B^ATP, formation of the base pairing interaction of U6 with the 5'ss of the MINX exon RNA would stabilize the anti-parallel orientation and thus hinder the change in polarity of the dimer during the BRR2 translocation events in the two pre-B protomers. **b**, RNA composition of pre-B^AMPPNP complexes. Purified pre-B complexes were incubated with the non-hydrolyzable ATP analog AMPPNP, and then subjected to glycerol gradient centrifugation. RNA from the fastest-sedimenting gradient peak was isolated, separated on a NuPAGE gel, and visualized by staining with SyBr gold. For gel source data, see Supplementary Fig. 1. The nucleotide (nts) lengths of the snRNAs and MINX exon RNA are indicated on the right. The RNA composition was analysed from two independent pre-B^AMPPNP purifications with similar results. **c**, Representative cryo-EM 2D class averages of the pre-B^AMPPNP dimers. **d**, Cryo-EM computation sorting scheme of the pre-B^AMPPNP. **e**, Local resolution estimation of the tri-snRNP core region of the pre-B^AMPPNP complex. **f**, Orientation distribution plot for the particles contributing to the reconstruction of the tri-snRNP region in of the pre-B^AMPPNP. **g**, Fourier shell correlation (FSC) values indicate a resolution of 4.1 Å for the tri-snRNP core and 6.1 Å for the tri-snRNP plus adjacent U1 snRNP. **h**, Map versus model FSC curves generated for the tri-snRNP core and tri-snRNP plus adjacent U1 snRNP regions of pre-B^AMPPNP. **i**, Fit into the EM density (low-pass filtered to -20 Å resolution) of the molecular model of a pre-B^AMPPNP complex monomer interacting *in trans* with a U1 snRNP bound to a second MINX exon RNA. On the right, the molecular architecture of the U1 snRNP and adjacent BRR2 and PRP28 proteins without EM density, as well as a cartoon of the unassigned EM density adjacent to U1, is shown. In pre-B^AMPPNP, U1 snRNA stem-loop III is located close to PRP8^En of the adjacent protomer, and the U1 Sm domain contacts the PRP28 RecA1 domain. **j**, Fit into the EM density (low-pass filtered to -20 Å resolution) of the molecular model of a pre-B complex monomer interacting *in trans* with PRP28 from the adjacent pre-B monomer. On the right, the molecular architecture of BRR2 and PRP28 without EM density, as well as a cartoon of the unassigned EM density that likely contains the U1 snRNP, is shown. **k**, Close up of PRP28 docked to U1 of the adjacent monomer in the pre-B^AMPPNP dimer. A molecular model of PRP28 with a closed conformation and bound with a single-stranded RNA (based on the crystal structure of Mss116p) can be fit into the pre-B^AMPPNP EM density. In pre-B^AMPPNP, there is density between the two PRP28 RecA domains that appears to accommodate a single-stranded (ssRNA) but not double stranded RNA. Given this is indeed the case, our data would thus suggest that in the presence of AMPPNP, PRP28 has disrupted the U1/5'ss helix of the adjacent protomer, but due to the lack of ATP hydrolysis, its RecA domains are trapped in a closed conformation. **l**, EM density fit of PRP28 with an open conformation in the corresponding region of the cross-exon pre-B complex. In the CE pre-B complex, PRP28 adopts an open conformation, and there is no stable docking of U1. **m-o**, The PRP28 RecA domains undergo a major conformational change. Spatial organization of the PRP28 RecA domains in the closed (m) and open (n) conformations, with an overlay shown in panel o. The open conformation of a DEAD-box helicase such as PRP28 is its default state when it is not bound to a substrate[50]. When the helicase encounters a double-helical RNA in the presence of ATP, it transitions to the closed conformation, where the subsequent closure of the two RecA domains physically separates the two strands[50], resulting in closed RecA domains bound to a single-stranded RNA (ssRNA). **p**, Fit of the U6/5'ss helix into the EM density of pre-B^ATP. **q**, A U6/5'ss helix is not formed in pre-B^AMPPNP. EM density that accommodates a U6/5'ss helix (as seen in panel p) is absent in pre-B^AMPPNP, confirming that this helix does not form.

## Extended Data Table 1 | Cryo-EM data collection, refinement and validation statistics

| | Human pre-B complex | | | Human B-like complex | | | Human pre-B+5'ss (0 °C) | | | Human pre-B +5'ss (30 °C) | Human pre-B+5'ss +ATPγS | | | Human pre-B +5'ss +ATP | Human pre-B +5'ssLNG+ATPγS | | | Human pre-B+ATP | | | Human pre-B +AMPPNP | | |
|---|---|---|---|---|---|---|---|---|---|---|---|---|---|---|---|---|---|---|---|---|---|---|---|
| | dimer | protomer | tri-snRNP core | dimer | protomer | tri-snRNP core | dimer (0 °C) | protomer (0 °C) | tri-snRNP core (0 °C) | protomer (30 °C) | dimer | protomer | tri-snRNP core | protomer | dimer | protomer | tri-snRNP core | dimer | protomer | tri-snRNP core | protomer | tri-snRNP core | core with U1 snRNP |
| EMDB | 19594 | 18718 | 18544 | 19595 | 18781 | 18548 | 19868 | 18788 | 18555 | 19847 | 19596 | 18787 | 18542 | 19848 | 19597 | 19349 | 18546 | 19598 | 18789 | 18547 | 18786 | 18545 | 18727 |
| PDB | N.A. | 8QXD | 8QP8 | N.A. | 8QZS | 8QPE | N.A. | 8R0A | 8QPK | N.A. | N.A. | 8R09 | 8QOZ | N.A. | N.A. | 8RM5 | 8QPA | N.A. | 8R0B | 8QPB | 8R08 | 8QP9 | N.A. |
| **Data collection and processing** | | | | | | | | | | | | | | | | | | | | | | | |
| Magnification | 120,700 | | | 120,700 | | | 120,700 | | | 120,700 | 120,700 | | | 120,700 | 120,700 | | | 120,700 | | | 103,700 | | |
| Voltage (kV) | 300 kV | | | 300 kV | | | 300 kV | | | 300 kV | 300 kV | | | 300 kV | 300 kV | | | 300 kV | | | 300 kV | | |
| Electron exposure (e–/Å²) | 60 | | | 60 | | | 60 | | | 60 | 60 | | | 60 | 60 | | | 60 | | | 48 | | |
| Defocus range (µm) | 1.5-4 | | | 1.5-4 | | | 1.5-4 | | | 1.5-4 | 1.5-4 | | | 1.5-4 | 1.5-4 | | | 1.5-4 | | | 1.5-4 | | |
| Pixel size (Å) | 1.16 | | | 1.16 | | | 1.16 | | | 1.16 | 1.16 | | | 1.16 | 1.16 | | | 1.16 | | | 1.35 | | |
| Symmetry imposed | C1 | | | C1 | | | C1 | | | C1 | C1 | | | C1 | C1 | | | C1 | | | C1 | | |
| Initial particle images (no.) | 777,350 | | | 488,598 | | | 1,283,541 | | | 283,599 | 791,079 | | | 309,120 | 541,230 | | | 757,260 | | | 619,945 | | |
| Final particle images (no.) | 55,845 | 279,781 | 279,781 | 80,810 | 334,084 | 334,084 | 33,997 | 176,879 | 176,879 | 65,433 | 62,020 | 411,185 | 411,185 | 71,742 | 29,993 | 136,333 | 136,333 | 12,355 | 94,460 | 94,460 | 53,422 | 53,422 | 53,422 |
| Map resolution (Å) | -- | 4.5 | 3.5 | -- | 4.1 | 3.1 | -- | 5.8 | 4.2 | 8 | -- | 4.3 | 3.1 | 3.9 | -- | 4.5 | 3.7 | -- | 4.4 | 3.7 | 8 | 4.1 | 6.1 |
| FSC threshold | | 0.143 | 0.143 | | 0.143 | 0.143 | | 0.143 | 0.143 | 0.143 | | 0.143 | 0.143 | 0.143 | | 0.143 | 0.143 | | 0.143 | 0.143 | 0.143 | 0.143 | 0.143 |
| Map resolution range (Å) | 15-30 | 4-8 | 3.1-6 | 15-30 | 3.5-8 | 2.8-3.8 | 20-40 | 4.5-15 | 3.8-10 | 6-10 | 15-30 | 4-8 | 2.9-6 | 3.5-8 | 15-30 | 4.2-8 | 3.4-8 | 15-30 | 4-8 | 3.5-8 | 6-12 | 4-10 | 4.5-12 |
| **Refinement** | | | | | | | | | | | | | | | | | | | | | | | |
| Initial model used (PDB code) | | 6QX9 | | | 8Q7N | | | -- | 6QX9; B-like (this study) | -- | | | B-like (this study) | B-like (this study) | | | B-like (this study) | | | B-like (this study) | 6QX9 | | |
| Model resolution (Å) | | | 3.46 | | | 3.11 | | | 4.39 | | | | 3.23 | | | | 3.84 | | | 3.89 | | 4.27 | |
| FSC threshold | | | 0.5 | | | 0.5 | | | 0.5 | | | | 0.5 | | | | 0.5 | | | 0.5 | | 0.5 | |
| Model resolution range (Å) | | | 3.2-6 | | | 3.0-4.2 | | | 4.2-10 | | | | 3.1-6 | | | | 3.6-8 | | | 3.6-8 | | 6-10 | |
| Map sharpening B factor (Å²) | | -100 | -80 | | -100 | -70 | | -220 | -150 | -250 | | -100 | -70 | -100 | | -130 | -100 | | -120 | -90 | -220 | -160 | -200 |
| Model composition | | | | | | | | | | | | | | | | | | | | | | | |
| Non-hydrogen atoms | | | 34526 | | | 54531 | | | 29486 | | | | 45277 | | | | 43418 | | | 44739 | | 24241 | |
| Protein residues | | | 3908 | | | 6251 | | | 3973 | | | | 5152 | | | | 4880 | | | 5088 | | 4023 | |
| Ligands | | | 1 | | | 1 | | | 1 | | | | 1 | | | | 1 | | | 1 | | 1 | |
| B factors (Å²) | | | | | | | | | | | | | | | | | | | | | | | |
| Protein | | | 73.9 | | | 57.6 | | | 100.8 | | | | 36.6 | | | | 83.0 | | | 90.0 | | 39.3 | |
| Ligand | | | 191.7 | | | 94.9 | | | 262.0 | | | | 65.6 | | | | 116.2 | | | 110.1 | | 183.7 | |
| R.m.s. deviations | | | | | | | | | | | | | | | | | | | | | | | |
| Bond lengths (Å) | | | 0.003 | | | 0.002 | | | 0.002 | | | | 0.004 | | | | 0.004 | | | 0.002 | | 0.002 | |
| Bond angles (°) | | | 0.598 | | | 0.459 | | | 0.489 | | | | 0.538 | | | | 0.751 | | | 0.513 | | 0.568 | |
| Validation | | | | | | | | | | | | | | | | | | | | | | | |
| MolProbity score | | | 1.72 | | | 1.46 | | | 1.78 | | | | 1.69 | | | | 2.02 | | | 1.74 | | 1.53 | |
| Clashscore | | | 10.10 | | | 8.44 | | | 9.66 | | | | 10.47 | | | | 13.70 | | | 11.29 | | 5.76 | |
| Poor rotamers (%) | | | 0.03 | | | 0.00 | | | 0.00 | | | | 0.00 | | | | 0.36 | | | 0.00 | | 0.00 | |
| Ramachandran plot | | | | | | | | | | | | | | | | | | | | | | | |
| Favored (%) | | | 96.79 | | | 98.10 | | | 95.98 | | | | 97.16 | | | | 94.47 | | | 96.98 | | 96.66 | |
| Allowed (%) | | | 3.18 | | | 1.90 | | | 4.02 | | | | 2.84 | | | | 5.53 | | | 3.02 | | 3.34 | |
| Disallowed (%) | | | 0.00 | | | 0.00 | | | 0.00 | | | | 0.00 | | | | 0.00 | | | 0.00 | | 0.00 | |

Data collection, refinement and validation statistics for human cross-exon (CE) pre-B, pre-B[5'ss], pre-B[ATP], pre-B[5'ss+ATPγS], pre-B[5'ssLNG+ATPγS], pre-B[AMPPNP], and B-like complexes.

Reinhard Lührmann
Holger Stark

# Reporting Summary

## Statistics

For all statistical analyses, confirm that the following items are present in the figure legend, table legend, main text, or Methods section.

| n/a | Confirmed | |
|---|---|---|
| ☐ | ☒ | The exact sample size (*n*) for each experimental group/condition, given as a discrete number and unit of measurement |
| ☒ | ☐ | A statement on whether measurements were taken from distinct samples or whether the same sample was measured repeatedly |
| ☒ | ☐ | The statistical test(s) used AND whether they are one- or two-sided<br>*Only common tests should be described solely by name; describe more complex techniques in the Methods section.* |
| ☒ | ☐ | A description of all covariates tested |
| ☒ | ☐ | A description of any assumptions or corrections, such as tests of normality and adjustment for multiple comparisons |
| ☒ | ☐ | A full description of the statistical parameters including central tendency (e.g. means) or other basic estimates (e.g. regression coefficient) AND variation (e.g. standard deviation) or associated estimates of uncertainty (e.g. confidence intervals) |
| ☒ | ☐ | For null hypothesis testing, the test statistic (e.g. $F$, $t$, $r$) with confidence intervals, effect sizes, degrees of freedom and $P$ value noted<br>*Give P values as exact values whenever suitable.* |
| ☒ | ☐ | For Bayesian analysis, information on the choice of priors and Markov chain Monte Carlo settings |
| ☒ | ☐ | For hierarchical and complex designs, identification of the appropriate level for tests and full reporting of outcomes |
| ☒ | ☐ | Estimates of effect sizes (e.g. Cohen's *d*, Pearson's *r*), indicating how they were calculated |

*Our web collection on statistics for biologists contains articles on many of the points above.*

## Software and code

Policy information about availability of computer code

| Data collection | Thermo Fischer EPU2.1, Thermo Scientific Orbitrap Fusion Lumos v. 3.4.3072, Thermo Scientific Orbitrap Exploris 480 v. 4.0.30.28, Xcalibur v.4.5.445.18, Xcalibur v.4.4.16.14 |
|---|---|
| Data analysis | Max Quant v.2.4.2.0, pLink 2.3.9 and 2.3.11, MotionCor2, Gctf v.1.06, RELION 3.1, UCSF Chimera v.1.13.1, ChimeraX v.1.3, cryolo v.1.6.1, cryoSPARC v.2.1, COOT v.0.8.92, PHENIX v.1.2 |

For manuscripts utilizing custom algorithms or software that are central to the research but not yet described in published literature, software must be made available to editors and reviewers. We strongly encourage code deposition in a community repository (e.g. GitHub). See the Nature Portfolio guidelines for submitting code & software for further information.

## Data

The cryo-EM maps and coordinates have been deposited to the Electron Microscopy Data Bank (EMDB) and Protein Data Bank (PDB) as follows: pre-B protomer (EMD-18718, PDB ID 8QXD), B-like protomer (EMD-18781, PDB ID 8QZS), pre-B5'ss protomer at 0 °C (EMD-18788, PDB ID 8R0A), pre-B5'ss protomer at 30 °C (EMD-19847), pre-B5'ss+ATPγS protomer (EMD-18787, PDB ID 8R09),  pre-B5'ss+ATP protomer (EMD-19848), pre-B5'ssLNG+ATPγS protomer (EMD-19349, PDB ID 8RM5),  pre-BATP protomer (EMD-18789, PDB ID 8R0B),  pre-BAMPPNP protomer (EMD-18786, PDB ID 8R08);  pre-B dimer (EMD-19594), B-like dimer (EMD-19595), pre-B5'ss dimer at 0 °C (EMD-19868),  pre-B5'ss+ATPγS dimer (EMD-19596),  pre-B5'ssLNG+ATPγS dimer (EMD-19597), pre-BATP dimer (EMD-19598); the tri-snRNP core of pre-B (EMD-18544, PDB ID 8QP8),  the tri-snRNP core of B-like (EMD-18548, PDB ID 8QPE),  the tri-snRNP core of pre-B5'ss at 0 °C (EMD-18555, PDB ID 8QPK),  the tri-snRNP core of pre-B5'ss+ATPγS (EMD-18542, 8QOZ), the tri-snRNP core of pre-B5'ssLNG+ATPγS (EMD-18546, PDB ID 8QPA), the tri-snRNP core of pre-BATP (EMD-18547, PDB ID 8QPB), the tri-snRNP core of pre-BAMPPNP (EMD-18545,  PDB ID 8QP9),  and the tri-snRNP core plus U1 snRNP of pre-BAMPPNP (EMD-18727).

## Research involving human participants, their data, or biological material

| | |
|---|---|
| Reporting on sex and gender | Not applicable |
| Reporting on race, ethnicity, or other socially relevant groupings | Not applicable |
| Population characteristics | Not applicable |
| Recruitment | Not applicable |
| Ethics oversight | Not applicable |

Note that full information on the approval of the study protocol must also be provided in the manuscript.

# Field-specific reporting

Please select the one below that is the best fit for your research. If you are not sure, read the appropriate sections before making your selection.

☒ Life sciences        ☐ Behavioural & social sciences        ☐ Ecological, evolutionary & environmental sciences

# Life sciences study design

All studies must disclose on these points even when the disclosure is negative.

| | |
|---|---|
| Sample size | Sample size does not apply to this study because it deals with a macromolecular structure. |
| Data exclusions | No cryo-EM micrographs were removed. Data processing was performed according to pre-established common image classification procedures in the field, in order to select particle images with the highest resolution content in the cryo-EM reconstruction process. Details of the number of selected images are reported in Extended Data Table 1. |
| Replication | Experimental replication does not apply for a macromolecular structure. |
| Randomization | Cryo-EM data were split randomly into two halves for gold-standard FSC determination in RELION 3.1. |
| Blinding | The nature of this study does not require blinding because it does not involve a clinical trial or treatment allocation. |

# Behavioural & social sciences study design

All studies must disclose on these points even when the disclosure is negative.

| | |
|---|---|
| Study description | *Briefly describe the study type including whether data are quantitative, qualitative, or mixed-methods (e.g. qualitative cross-sectional, quantitative experimental, mixed-methods case study).* |
| Research sample | *State the research sample (e.g. Harvard university undergraduates, villagers in rural India) and provide relevant demographic information (e.g. age, sex) and indicate whether the sample is representative. Provide a rationale for the study sample chosen. For studies involving existing datasets, please describe the dataset and source.* |
| Sampling strategy | *Describe the sampling procedure (e.g. random, snowball, stratified, convenience). Describe the statistical methods that were used to predetermine sample size OR if no sample-size calculation was performed, describe how sample sizes were chosen and provide a rationale for why these sample sizes are sufficient. For qualitative data, please indicate whether data saturation was considered, and what criteria were used to decide that no further sampling was needed.* |
| Data collection | *Provide details about the data collection procedure, including the instruments or devices used to record the data (e.g. pen and paper, computer, eye tracker, video or audio equipment) whether anyone was present besides the participant(s) and the researcher, and whether the researcher was blind to experimental condition and/or the study hypothesis during data collection.* |
| Timing | *Indicate the start and stop dates of data collection. If there is a gap between collection periods, state the dates for each sample cohort.* |
| Data exclusions | *If no data were excluded from the analyses, state so OR if data were excluded, provide the exact number of exclusions and the rationale behind them, indicating whether exclusion criteria were pre-established.* |
| Non-participation | *State how many participants dropped out/declined participation and the reason(s) given OR provide response rate OR state that no participants dropped out/declined participation.* |
| Randomization | *If participants were not allocated into experimental groups, state so OR describe how participants were allocated to groups, and if allocation was not random, describe how covariates were controlled.* |

# Ecological, evolutionary & environmental sciences study design

All studies must disclose on these points even when the disclosure is negative.

| | |
|---|---|
| Study description | *Briefly describe the study. For quantitative data include treatment factors and interactions, design structure (e.g. factorial, nested, hierarchical), nature and number of experimental units and replicates.* |
| Research sample | *Describe the research sample (e.g. a group of tagged Passer domesticus, all Stenocereus thurberi within Organ Pipe Cactus National Monument), and provide a rationale for the sample choice. When relevant, describe the organism taxa, source, sex, age range and any manipulations. State what population the sample is meant to represent when applicable. For studies involving existing datasets, describe the data and its source.* |
| Sampling strategy | *Note the sampling procedure. Describe the statistical methods that were used to predetermine sample size OR if no sample-size calculation was performed, describe how sample sizes were chosen and provide a rationale for why these sample sizes are sufficient.* |
| Data collection | *Describe the data collection procedure, including who recorded the data and how.* |
| Timing and spatial scale | *Indicate the start and stop dates of data collection, noting the frequency and periodicity of sampling and providing a rationale for these choices. If there is a gap between collection periods, state the dates for each sample cohort. Specify the spatial scale from which the data are taken* |
| Data exclusions | *If no data were excluded from the analyses, state so OR if data were excluded, describe the exclusions and the rationale behind them, indicating whether exclusion criteria were pre-established.* |
| Reproducibility | *Describe the measures taken to verify the reproducibility of experimental findings. For each experiment, note whether any attempts to repeat the experiment failed OR state that all attempts to repeat the experiment were successful.* |
| Randomization | *Describe how samples/organisms/participants were allocated into groups. If allocation was not random, describe how covariates were controlled. If this is not relevant to your study, explain why.* |
| Blinding | *Describe the extent of blinding used during data acquisition and analysis. If blinding was not possible, describe why OR explain why blinding was not relevant to your study.* |

Did the study involve field work? ☐ Yes ☐ No

# Field work, collection and transport

| | |
|---|---|
| Field conditions | *Describe the study conditions for field work, providing relevant parameters (e.g. temperature, rainfall).* |
| Location | *State the location of the sampling or experiment, providing relevant parameters (e.g. latitude and longitude, elevation, water depth).* |
| Access & import/export | *Describe the efforts you have made to access habitats and to collect and import/export your samples in a responsible manner and in compliance with local, national and international laws, noting any permits that were obtained (give the name of the issuing authority, the date of issue, and any identifying information).* |
| Disturbance | *Describe any disturbance caused by the study and how it was minimized.* |

# Reporting for specific materials, systems and methods

We require information from authors about some types of materials, experimental systems and methods used in many studies. Here, indicate whether each material, system or method listed is relevant to your study. If you are not sure if a list item applies to your research, read the appropriate section before selecting a response.

## Materials & experimental systems

| n/a | Involved in the study |
|---|---|
| ☐ | ☒ Antibodies |
| ☐ | ☒ Eukaryotic cell lines |
| ☒ | ☐ Palaeontology and archaeology |
| ☒ | ☐ Animals and other organisms |
| ☒ | ☐ Clinical data |
| ☒ | ☐ Dual use research of concern |
| ☒ | ☐ Plants |

## Methods

| n/a | Involved in the study |
|---|---|
| ☒ | ☐ ChIP-seq |
| ☒ | ☐ Flow cytometry |
| ☒ | ☐ MRI-based neuroimaging |

## Antibodies

| | |
|---|---|
| Antibodies used | Antibodies against human phospho-PRP6, phospho-PRP31, PRP31 and SF3B1 were prepared in house. Anti-human PRP6 antibodies were purchased from Abcam (cat# AB99292) HRP-conjugated goat anti-rabbit antibodies were purchased from Jackson Immunoresearch (cat# 111-035-144). |
| Validation | Antibodies prepared in house were validated experimentally - see Schneider et al, NSMB 17:216 (2010) and Will et al EMBO J 20:4536 (2001). |

## Eukaryotic cell lines

Policy information about cell lines and Sex and Gender in Research

| | |
|---|---|
| Cell line source(s) | HeLa S3 cells were obtained from the Helmholtz Zentrum für Infektionsforschung, Braunschweig. |
| Authentication | Cells were not authenticated |
| Mycoplasma contamination | HeLa S3 cells tested negative for mycoplasma. |
| Commonly misidentified lines (See ICLAC register) | No commonly misidentified lines were used. |

## Palaeontology and Archaeology

| | |
|---|---|
| Specimen provenance | *Provide provenance information for specimens and describe permits that were obtained for the work (including the name of the issuing authority, the date of issue, and any identifying information). Permits should encompass collection and, where applicable, export.* |
| Specimen deposition | *Indicate where the specimens have been deposited to permit free access by other researchers.* |

| Dating methods | *If new dates are provided, describe how they were obtained (e.g. collection, storage, sample pretreatment and measurement), where they were obtained (i.e. lab name), the calibration program and the protocol for quality assurance OR state that no new dates are provided.* |

☐ Tick this box to confirm that the raw and calibrated dates are available in the paper or in Supplementary Information.

| Ethics oversight | *Identify the organization(s) that approved or provided guidance on the study protocol, OR state that no ethical approval or guidance was required and explain why not.* |

Note that full information on the approval of the study protocol must also be provided in the manuscript.

# Animals and other research organisms

Policy information about studies involving animals; ARRIVE guidelines recommended for reporting animal research, and Sex and Gender in Research

| Laboratory animals | *For laboratory animals, report species, strain and age OR state that the study did not involve laboratory animals.* |

| Wild animals | *Provide details on animals observed in or captured in the field; report species and age where possible. Describe how animals were caught and transported and what happened to captive animals after the study (if killed, explain why and describe method; if released, say where and when) OR state that the study did not involve wild animals.* |

| Reporting on sex | *Indicate if findings apply to only one sex; describe whether sex was considered in study design, methods used for assigning sex. Provide data disaggregated for sex where this information has been collected in the source data as appropriate; provide overall numbers in this Reporting Summary. Please state if this information has not been collected.  Report sex-based analyses where performed, justify reasons for lack of sex-based analysis.* |

| Field-collected samples | *For laboratory work with field-collected samples, describe all relevant parameters such as housing, maintenance, temperature, photoperiod and end-of-experiment protocol OR state that the study did not involve samples collected from the field.* |

| Ethics oversight | *Identify the organization(s) that approved or provided guidance on the study protocol, OR state that no ethical approval or guidance was required and explain why not.* |

Note that full information on the approval of the study protocol must also be provided in the manuscript.

# Clinical data

Policy information about clinical studies
All manuscripts should comply with the ICMJE guidelines for publication of clinical research and a completed CONSORT checklist must be included with all submissions.

| Clinical trial registration | *Provide the trial registration number from ClinicalTrials.gov or an equivalent agency.* |

| Study protocol | *Note where the full trial protocol can be accessed OR if not available, explain why.* |

| Data collection | *Describe the settings and locales of data collection, noting the time periods of recruitment and data collection.* |

| Outcomes | *Describe how you pre-defined primary and secondary outcome measures and how you assessed these measures.* |

# Dual use research of concern

Policy information about dual use research of concern

## Hazards

Could the accidental, deliberate or reckless misuse of agents or technologies generated in the work, or the application of information presented in the manuscript, pose a threat to:

No | Yes

☐ ☐ Public health

☐ ☐ National security

☐ ☐ Crops and/or livestock

☐ ☐ Ecosystems

☐ ☐ Any other significant area

## Experiments of concern

Does the work involve any of these experiments of concern:

No | Yes

☐ ☐ Demonstrate how to render a vaccine ineffective

☐ ☐ Confer resistance to therapeutically useful antibiotics or antiviral agents

☐ ☐ Enhance the virulence of a pathogen or render a nonpathogen virulent

☐ ☐ Increase transmissibility of a pathogen

☐ ☐ Alter the host range of a pathogen

☐ ☐ Enable evasion of diagnostic/detection modalities

☐ ☐ Enable the weaponization of a biological agent or toxin

☐ ☐ Any other potentially harmful combination of experiments and agents

# Plants

**Seed stocks**
Report on the source of all seed stocks or other plant material used. If applicable, state the seed stock centre and catalogue number. If plant specimens were collected from the field, describe the collection location, date and sampling procedures.

**Novel plant genotypes**
Describe the methods by which all novel plant genotypes were produced. This includes those generated by transgenic approaches, gene editing, chemical/radiation-based mutagenesis and hybridization. For transgenic lines, describe the transformation method, the number of independent lines analyzed and the generation upon which experiments were performed. For gene-edited lines, describe the editor used, the endogenous sequence targeted for editing, the targeting guide RNA sequence (if applicable) and how the editor was applied.

**Authentication**
Describe any authentication procedures for each seed stock used or novel genotype generated. Describe any experiments used to assess the effect of a mutation and, where applicable, how potential secondary effects (e.g. second site T-DNA insertions, mosiacism, off-target gene editing) were examined.

# ChIP-seq

## Data deposition

☐ Confirm that both raw and final processed data have been deposited in a public database such as GEO.

☐ Confirm that you have deposited or provided access to graph files (e.g. BED files) for the called peaks.

**Data access links**
*May remain private before publication.*
For "Initial submission" or "Revised version" documents, provide reviewer access links. For your "Final submission" document, provide a link to the deposited data.

**Files in database submission**
Provide a list of all files available in the database submission.

**Genome browser session**
(e.g. UCSC)
Provide a link to an anonymized genome browser session for "Initial submission" and "Revised version" documents only, to enable peer review. Write "no longer applicable" for "Final submission" documents.

## Methodology

**Replicates**
Describe the experimental replicates, specifying number, type and replicate agreement.

**Sequencing depth**
Describe the sequencing depth for each experiment, providing the total number of reads, uniquely mapped reads, length of reads and whether they were paired- or single-end.

**Antibodies**
Describe the antibodies used for the ChIP-seq experiments; as applicable, provide supplier name, catalog number, clone name, and lot number.

**Peak calling parameters**
Specify the command line program and parameters used for read mapping and peak calling, including the ChIP, control and index files used.

**Data quality**
Describe the methods used to ensure data quality in full detail, including how many peaks are at FDR 5% and above 5-fold enrichment.

**Software**
Describe the software used to collect and analyze the ChIP-seq data. For custom code that has been deposited into a community repository, provide accession details.

# Flow Cytometry

## Plots

Confirm that:

☐ The axis labels state the marker and fluorochrome used (e.g. CD4-FITC).

☐ The axis scales are clearly visible. Include numbers along axes only for bottom left plot of group (a 'group' is an analysis of identical markers).

☐ All plots are contour plots with outliers or pseudocolor plots.

☐ A numerical value for number of cells or percentage (with statistics) is provided.

## Methodology

| | |
|---|---|
| Sample preparation | *Describe the sample preparation, detailing the biological source of the cells and any tissue processing steps used.* |
| Instrument | *Identify the instrument used for data collection, specifying make and model number.* |
| Software | *Describe the software used to collect and analyze the flow cytometry data. For custom code that has been deposited into a community repository, provide accession details.* |
| Cell population abundance | *Describe the abundance of the relevant cell populations within post-sort fractions, providing details on the purity of the samples and how it was determined.* |
| Gating strategy | *Describe the gating strategy used for all relevant experiments, specifying the preliminary FSC/SSC gates of the starting cell population, indicating where boundaries between "positive" and "negative" staining cell populations are defined.* |

☐ Tick this box to confirm that a figure exemplifying the gating strategy is provided in the Supplementary Information.

# Magnetic resonance imaging

## Experimental design

| | |
|---|---|
| Design type | *Indicate task or resting state; event-related or block design.* |
| Design specifications | *Specify the number of blocks, trials or experimental units per session and/or subject, and specify the length of each trial or block (if trials are blocked) and interval between trials.* |
| Behavioral performance measures | *State number and/or type of variables recorded (e.g. correct button press, response time) and what statistics were used to establish that the subjects were performing the task as expected (e.g. mean, range, and/or standard deviation across subjects).* |

## Acquisition

| | |
|---|---|
| Imaging type(s) | *Specify: functional, structural, diffusion, perfusion.* |
| Field strength | *Specify in Tesla* |
| Sequence & imaging parameters | *Specify the pulse sequence type (gradient echo, spin echo, etc.), imaging type (EPI, spiral, etc.), field of view, matrix size, slice thickness, orientation and TE/TR/flip angle.* |
| Area of acquisition | *State whether a whole brain scan was used OR define the area of acquisition, describing how the region was determined.* |

Diffusion MRI    ☐ Used    ☐ Not used

## Preprocessing

| | |
|---|---|
| Preprocessing software | *Provide detail on software version and revision number and on specific parameters (model/functions, brain extraction, segmentation, smoothing kernel size, etc.).* |
| Normalization | *If data were normalized/standardized, describe the approach(es): specify linear or non-linear and define image types used for transformation OR indicate that data were not normalized and explain rationale for lack of normalization.* |
| Normalization template | *Describe the template used for normalization/transformation, specifying subject space or group standardized space (e.g. original Talairach, MNI305, ICBM152) OR indicate that the data were not normalized.* |
| Noise and artifact removal | *Describe your procedure(s) for artifact and structured noise removal, specifying motion parameters, tissue signals and physiological signals (heart rate, respiration).* |

| Volume censoring | *Define your software and/or method and criteria for volume censoring, and state the extent of such censoring.* |

## Statistical modeling & inference

| Model type and settings | *Specify type (mass univariate, multivariate, RSA, predictive, etc.) and describe essential details of the model at the first and second levels (e.g. fixed, random or mixed effects; drift or auto-correlation).* |

| Effect(s) tested | *Define precise effect in terms of the task or stimulus conditions instead of psychological concepts and indicate whether ANOVA or factorial designs were used.* |

Specify type of analysis: ☐ Whole brain ☐ ROI-based ☐ Both

| Statistic type for inference | *Specify voxel-wise or cluster-wise and report all relevant parameters for cluster-wise methods.* |

(See Eklund et al. 2016)

| Correction | *Describe the type of correction and how it is obtained for multiple comparisons (e.g. FWE, FDR, permutation or Monte Carlo).* |

## Models & analysis

n/a | Involved in the study
☐ | ☐ Functional and/or effective connectivity
☐ | ☐ Graph analysis
☐ | ☐ Multivariate modeling or predictive analysis

| Functional and/or effective connectivity | *Report the measures of dependence used and the model details (e.g. Pearson correlation, partial correlation, mutual information).* |

| Graph analysis | *Report the dependent variable and connectivity measure, specifying weighted graph or binarized graph, subject- or group-level, and the global and/or node summaries used (e.g. clustering coefficient, efficiency, etc.).* |

| Multivariate modeling and predictive analysis | *Specify independent variables, features extraction and dimension reduction, model, training and evaluation metrics.* |

nature portfolio | reporting summary

April 2023