## [Peer Review File · Nature]

Manuscript Title: Structural insights into the cross-exon to cross-intron spliceosome switch

Reviewer Comments & Author Rebuttals

Reviewer Reports on the Initial Version:

Referees' comments:

Referee #1 (Remarks to the Author):

This manuscript is among the most important contributions to our understanding of an operational splicing mechanism, in a context of real genes, that this reviewer has ever seen. It is a technical tour-de-force, providing profound new insights into diverse aspects of spliceosomal activation and splicing mechanism. For example, it demonstrates that the core spliceosome is formed by all processes at the pre-B₂ stage, it reveals unexpected mechanical behavior Brr2, and it reveals key roles for B-specific proteins. However this paper is most significant because it sets the stage for an entirely new type of next-generation papers that help us understand how splicing works at the molecular level within the larger context of exon choice and splice site definition (much as the authors have begun to link splicing with transcriptional machineries). Ultimately, the findings can be interpreted in a manner that helps us decode the molecular basis for alternative splicing, which is central to our ability to understand and manipulate gene expression. As in any paper with this level of complexity, there are aspects that could be clarified for the reader, and in the event that it is helpful, suggestions are enumerated below. That said, aspects of the paper that this reviewer found particularly striking and important are also articulated.

Additional supportive comments:

- The similarities among CE and CI complexes are not only striking and mechanistically informative, but the fact that we now can view the various B and B-like complexes in different contexts greatly strengthens the field as a whole. It establishes that known models for spliceosomal complexes are robust across studies.

- That "pre-B complexes can productively bind in trans to a U1 snRNP-bound 5' splice site" is an extremely important finding for an operational understanding of RNA genetic architecture, and how it is mechanically defined.

- That tri-snRNP recruitment occurs in the same way, regardless of how exons and introns are defined is really important - it greatly simplifies a process for spliceosome assembly and function along a gene.

- The complexity of BRR2 motor activity is striking and unexpected, particularly in the broader context of the RNA helicase field. It provides a useful paradigm for other Superfamily II helicase systems. Movie 5 was quite helpful in this regard.

- The unexpected participation of 5'-ss oligo2 was a windfall for understanding the B-like complex, and to this reviewer, underscoring how much spliceosomal architecture can drive RNA molecular recognition and positioning.

- It was important that the authors have now established that the persistence of the U4/U6 stem III.

Specific suggestions:

1. It would be helpful if some aspects of the construct design are moved from the Extended data to the main text, as the general reader is not going to understand a lot of the experimental design here. For example, I would have liked to see ExtData Fig.1a moved to Figure1.

2. Fig 1 and ExtData Fig3: This reviewer would have liked more information on the specific dimer interfaces of these complexes, and some clear indication of any possible functional relevance. It is difficult to see from the figures, and based on the statements on p.12, line 4up (on the RNA-RNA network) and on the last few lines of page 13, this interface is probably functionally important. It would be helpful to create a specific extended data figure just on these interfaces, with the components clearly labeled. Also, it would be helpful if this could be addressed head-on in the manuscript: Are these dimer interfaces of functional importance? Aside from the models mentioned in the text above, are there cases where there could indeed be two spliceosomes aligned as shown? Why or why not?

3. ExtDataFig1f and Exdata Fig5e: There are some preferred orientation issues with both of these particle types. While this reviewer does not think it's sufficiently problematic to revisit the processing or analysis, it would have been helpful to explain explicitly how it might have impacted specific aspects of the modeling, and in turn, how it impacts final interpretation of the models and mechanism.

4. Page 5, line 6up: How does the dimer interface differ between these two complexes (see comment above)?

5. Fig.2 b-d and ExtData F6f: Would you please add a superimposed faint mesh map over these images of the interactions indicated, or provide the density map somewhere else in these figures? It is not clear to this reviewer how good the map is in this region, or how well-resolved the base interactions are. This is important for knowing if the RNA is recognized similarly in other complexes to which these structures are compared.

6. Movie 1: Prp8 domains are not labeled in a manner consistent with the text (Prp8Large is not noted - do you mean RT/En domain?)

7. Movies 1-2. This reviewer is not really seeing evidence for Prp4K moving away from SF3b, and toward Prp6 such that a specific new interface is being made during the structural transition. Indeed, in the movies, it just looks like there is some rotation of the surrounding components while Prp4K fails to make any clear new close interface with a different surface.

8. Prp4K: Relevant to the above comment: If the authors do not observe specific interfaces involving Prp4K, it does not seem appropriate for them to make specific comments about kinase activation and mechanism later in the text. To that end, if the authors have a better view of Prp4K interfaces and it would strengthen the paper if they were provided as a figure.

9. There is a typo on page 11, line 2up.

10. As a follow-up in future work, how can the complexity of the BRR2-B-specific protein network be sorted out mechanistically? The paper makes it clear that this network is very important, but it's not evident how it can be untangled. Any insights into this would be helpful.

11. Figure 4: At least for this reader, this figure does not clearly convey how the B-specific proteins are involved in any specific way. It is a complicated figure, and yet it does not really communicate the ideas that the authors seem to be stating, nor does it draw attention to specific players amidst the alphabet soup of proteins. A new type of graphic here would be helpful.

12. Final sentence of the Results section, regarding the handoff, is unclear and should be rewritten for the general reader.

13. The latter part of the discussion on alternative splicing and splice site choice was outstanding. Having stated these ideas, however, is it now possible to implicate specific B-specific proteins and begin testing their roles in splice site differentiation?

Referee #2 (Remarks to the Author):

Zhang, Kumar et al report a detailed cryo-EM analysis of spliceosomal complexes that assemble on an RNA substrate harboring a 3' splice site region followed by an exonic sequence and a 5' splice site. This cross-exon (CE) complex contains U2 snRNP bound to the 3' splice site region, U1 snRNP bound to the 5' splice site and, consistent with previous results by the Lührmann lab (Schneider et al Mol Cell 2010), it assembles also the U4/U6.U5 tri-snRNP. The authors find that the complex undergoes conformational transitions that are very similar or identical to those occurring during the assembly of CI spliceosomal complexes, including pre-B to B and extensive remodeling leading to replacement of U1 by U6 in the recognition of the 5' splice site. Strikingly, CE complexes can engage in trans with a 5' splice site bound by U1 snRNP, suggesting that the transition from CE to CI complexes -which is naturally required for intron removal, does not require the formation of new U1-U2 interactions across the intron, but can rather occur during the transition from pre-B to B in CE complexes.

This is a phenomenal piece of work, supported by remarkably extensive structural and biochemical data of the highest quality, that provides a detailed account of the composition and conformational dynamics possible in spliceosomal complexes occurring across exons, which recapitulate and expand knowledge acquired in previous studies of complexes formed across introns and provides an unprecedented model with the potential to explain the capacity of splicing factors involved in late events in spliceosome assembly to influence splice site selection, an observation without previous structural support.

In my opinion the following revisions would help to improve the manuscript:

1) The 30 years-old concept of exon definition refers to the mutual stabilization of early splicing factors interacting with the 3' and 5' splice sites across an exonic sequence, e.g. manifested in the observation that mutations in either of these splice sites lead to skipping of the exon rather than retention of the intron harboring the mutation (e.g. Berget et al 1995 PMID: 7852296). These effects are particularly noticeable in alternatively spliced exons rather than in constitutive exons, which has been associated to weaker splice sites flanking regulatable exons (e.g. Xiao et al 2007 PMID: 17998536). Even for alternative exons, the extent to which exon definition effects play a role in their recognition varies (e.g. Buratti et al 2006 PMID: 16855287; De Conti et al 2013 PMID: 23044818). The authors often refer to exon definition in their manuscript, but it is not shown that the exon and flanking splice sites used as the substrate in this study is, strictly speaking, exon defined. It would be important, in my opinion, to show that complex A formation is enhanced by the presence of the downstream 5' splice site (by comparing A complex formation on the substrate with or without 5' splice site). Even if this is not the case, the results of Zhang, Kumar et al remain of great interest (e.g. these effects may be happening across internal exons regardless of whether they are regulated or not), but it would be helpful to clarify whether the complexes described support the classical definition of exon definition or they reflect a more general phenomenon across all kinds of internal exons.

2) Related to the previous point, I did not manage to find a description of the sequences of the MINX Exon RNA substrate, and whether this substrate corresponds to a natural viral exon and its natural flanking splice sites, or whether a 5' splice site (which one?) was engineered into a piece of Adenovirus Major Late promoter pre-mRNA. This may be relevant because, if the exon has been engineered, it may not recapitulate the architecture of regulatory elements (e.g. splicing enhancers) that help in exon recognition. Once again, these considerations do not detract from the value of the findings of the paper, but can help to place them in a physiological context.

3) Yet another related point, the length of the exon in the MINX Exon substrate is 40 nucleotides, which is below or near the lower limit observed for sustaining exon definition effects (Hawkins 1988 PMID:

3057449; De Conti et al 2013 PMID: 23044818) and in fact close to the size of microexons, whose inclusion requires a special arrangement of cis-acting sequences and trans-acting factors (Irimia et al 2014; PMID: 25525873). It would be relevant to find whether exon definition effects (see point 1 above) are influenced by the length of the exon. I am not -of course- requesting a repetition of the extensive structural analyses on a longer exon substrate, but rather suggesting that this is an important point to clarify because the vast majority of internal exons are longer than the substrate used in this study. It is in the best interest of the authors to rule out that their complexes can only form on exons of a length shorter than the majority of internal exons, even in mammalian cells.

4) P4: “protein crosslinking coupled with mass spectrometry (CXMS) (Extended Data Fig. 2f and Supplementary Table 2) indicates that U1 and U1-related proteins communicate indirectly with U2 components, mainly via SR proteins that interact with the U2AF1/2 proteins and SF3B1 that are located near the 3' end of the intron, consistent with the idea that U2 and U1 do not directly interact with one another.” It would be good to discuss this in the context of previous interactions proposed to mediate exon definition, e.g. Plaschka et al, 2018 PMID: 29995849; Sharma et al, 2014 PMID: 25403181; Abovich and Rosbash, 1997 PMID: 9150140; Becerra et al 2016 PMID: 26494226).

5) Figure 6b: the scheme is nice but the coordinated action of the two protomers (Figure 1a, section on pages 12-14) is not at all evident in the diagram. As a matter of fact, the diagram seems to suggest that interactions between U2 and tri-snRNP occur at the 3' end of the intron, without any involvement of 5' splice site recognition across the exon.

6) The description of the complex dynamic changes in composition and contacts involved in the different conformational transitions is rigorous and well supported by the data, but I believe it will be challenging to follow for the general reader. I realize that the authors have already made an important effort to highlight the key messages of the manuscript, but an extra effort towards making the manuscript accessible to non-specialists will increase its impact.

Referee #3 (Remarks to the Author):

CE pre-B review

In this manuscript the authors present the purification, structural analysis and functional and biochemical chase of early spliceosomal complexes assembled across an exon and stalled initially at the pre-B stage, before handoff of the 5'SS to the U6 snRNA. The study is highly relevant to the regulation of alternative splicing and the long-debated competing cross-exon and cross-intron assembly pathways.

The striking finding that these CE pre-B complexes can dimerise and the resulting biochemical chases and structural analysis of the resulting pre-B and B complexes allows the authors to propose that cross-exon and cross-intron pathways converge at the pre-B stage through a mechanism in which CE pre-B tri-snRNPs are transferred to a 5'SS bound U1 snRNP in trans.

The manuscript presents a tour de force of biochemical and structural work and provides a provocative, yet compelling model that could explain more generally how alternative splicing factors can regulate e.g. exon skipping by affecting U1 and U2 snRNP binding without necessarily disrupting engagement of the tri-snRNP at any given exon. The model could also explain why, despite numerous attempts so far, no strong and stable structural interactions linking U1 and U2 snRNPs have been observed in either CI or CE A complexes, which appear generally very flexible.

While I generally find the authors' arguments compelling, I believe further explanations and potentially experimental support are necessary to make the proposed model fully convincing and to assess how general it may be. The following points should be addressed.

Major

1. As a general comment, the authors do not present any data on how prevalent the observed dimers are and also to what extent they persist through the various biochemical chases. E.g. if they start with isolated CE pre-B dimers and incubate these with the 5'SS oligo and ATP or just with ATP, does that change at all the proportion of dimers present in the sample? A comprehensive analysis of the relative distribution of dimers vs. monomers in the various samples would be very useful and provide further support for the authors' argument. Presumably this should be easily extracted from the gradient profiles and the cryo-EM initial 2D classes.

2. Relating to the argument on p.4 – can the authors exclude the possibility that their dimers represent complexes in which the substrate is bound to U1 and U2 within the same dimer rather than in different promoters as their model suggests ? i.e., is the supposed 3D arrangement of U1 snRNP from previous structures incompatible with an intra-protomer path ? a simple test of this model would be that RNase-H cleavage of the exon, or of a region in the PY tract that might be accessible even if engaged with SF3B1, should dissociate the dimers or result in dimers with a wider range of different orientations, if their model is correct; minimally, the authors should outline the full proposed path of the substrate and present in a figure whether the available length of the exon between the 5'SS and 3'SS is consistent with the distances observed in their model – it is possible that this exercise might indeed exclude the alternative intra-protomer possibility due to distance constraints, but it would be important to clarify this in a figure

3. Related to p. 5 – the predicted parallel arrangement of the B-like CE complexes suggests a massive rotation from the pre-B state, for which the authors don't provide any clear figure or structural explanation; it is partly this arrangement in B-like complexes that led me to consider the alternative model where the substrate 5'SS and 3'SS are present in the same protomer rather than across protomers in pre-B CE complexes, since the authors' models implies the 5'SS and 3'SS are also situated in the same protomer in the B-like CE complexes. It is in my view hard to imagine how the 5'SS oligo in trans could lead to a B complex where dimers have the exons in the GD1 shown in ED Fig. 3h. The authors do not provide any assessment of the heterogeneity of the protomer arrangement observed in the dimers following conversion from pre-B to B. Do all pre-B dimers observed have an the anti-parallel arrangement and do all resulting B dimers have a parallel arrangement ? Or did the authors only process this category of particles ?

4. Related to p13 – The model for Prp28 transfer in trans is compelling. However, the rationale for excluding the possibility that the observed events are intramolecular isn't entirely convincing in my view; the authors mention structural constraints on Prp28 on p. 12 but they don't show a clear figure explaining this. Additionally they don't provide any clear biochemical test, e.g. trying to cleave the substrate Minx in the pre-BAMPPNP sample in the region between the BP and 5'SS should disrupt the dimers.

5. The authors do not comment at all on whether they believe the observed dimers are necessary for the proposed model of CE to CI conversion. Is the implication that most exons would have such a CE A-like or pre-B complex formed during alternative splicing ?

Minor

6. On p. 8 it is unclear what is different about the sample preparation in the case of pre-B5'SS complexes that leads to only a single oligo being bound ? Is it the absence of the nuclear extract ? If so, does it imply that B complex proteins are important for formation of the extended U6/5'SS when the pairing is limited, as it would be in the native extract ? This is implied later on, but the authors should clarify this point when initially describing the pre-B phases.

Author Rebuttals to Initial Comments:

Referee #1

1. It would be helpful if some aspects of the construct design are moved from the Extended data to the main text, as the general reader is not going to understand a lot of the experimental design here. For example, I would have liked to see ExtData Fig.1a moved to Figure1.

We have moved the depiction of the MINX exon RNA construct from Extended Data Figure 1a to Fig. 1a.

2. Fig 1 and ExtData Fig3: This reviewer would have liked more information on the specific dimer interfaces of these complexes, and some clear indication of any possible functional relevance. It is difficult to see from the figures, and based on the statements on p.12, line 4up (on the RNA-RNA network) and on the last few lines of page 13, this interface is probably functionally important. It would be helpful to create a specific extended data figure just on these interfaces, with the components clearly labeled.

We now include a more detailed, expanded view of the pre-B and B-like molecular interfaces in Extended Data Fig. 1b and Extended Data Fig. 3h, and describe them in more detail in the figure legends. As we are allowed maximally ten Extended Data Figures, it is not possible to add an additional Extended Data figure that deals solely with this aspect.

Also, it would be helpful if this could be addressed head-on in the manuscript: Are these dimer interfaces of functional importance? Aside from the models mentioned in the text above, are there cases where there could indeed be two spliceosomes aligned as shown? Why or why not?

It is unclear whether the dimer interfaces are of any functional importance and any discussion related to this point would be purely speculative. We now mention in the legends to Extended Data Fig. 1 and Extended Data Fig. 3 that “The functional relevance of these interfaces, as well as dimer formation in general, is currently not known”. In a separate paper from our lab (Zhang et al., EMBO J., 2024), in which we analyze the molecular architecture of human B complexes formed on a MINX pre-mRNA substrate (i.e., with two exons separated by an intron), dimers are formed with the same orientation as the B-like dimers and with a similar interface.

3. ExtDataFig1f and Exdata Fig5e: There are some preferred orientation issues with both of these particle types. While this reviewer does not think it's sufficiently problematic to revisit the processing or analysis, it would have been helpful to explain explicitly how it might have impacted specific aspects of the modeling, and in turn, how it impacts final interpretation of the models and mechanism.

The existence of some preferred orientations should have essentially no substantial effect on the final interpretation of the molecular models and mechanisms. Although “top” or “bottom” views of the pre-B complexes are underrepresented, there are a wide range of “side” views and the pre-B complex map does not suffer from preferred orientation issues at the reported resolution. Indeed, the core map shows side-chain information in all orientations (see Figure below showing a region of PRP8 in pre-B). Similarly, the pre-B^{5'ss} complex map does not suffer from the observed, preferred orientations at the reported 4.2 Å resolution at its core. The secondary structures (e.g., alpha helical turns), as well as certain bulky side chains can be well visualized from all orientations, and the carbon backbone can be well-modeled (see Figure below). Although we could not model most of the side chains at this resolution, the main mechanistic insight revealed by the complex is not affected by the relatively lower resolution. For example, at this resolution, we can confidently show that only one copy of the 5'ss-containing oligo binds to the pre-B^{5'ss} complex, and that 5'ss oligo binding triggers a conformational change in PRP8, but is not sufficient to trigger the subsequent, large-scale structural remodeling of the tri-snRNP.

4. Page 5, line 6up: How does the dimer interface differ between these two complexes (see comment above)?

We now show a direct comparison of the interfaces of the B-like and B complex in Extended Data Fig. 3h,i and describe the differences in the legend to the figure.

5. Fig.2 b-d and ExtData F6f: Would you please add a superimposed faint mesh map over these images of the interactions indicated, or provide the density map somewhere else in these figures? It is not clear to this reviewer how good the map is in this region, or how well-resolved the base interactions are. This is important for knowing if the RNA is recognized similarly in other complexes to which these structures are compared.

The local resolution at the RNA core is ~2.9 Å, as shown in Extended Data Fig. 3d and Fig. 5l. At this resolution, the bases are clearly separated, and the protein side chains are well-resolved. New panels showing the fit of RNA bases and protein side chains into the EM density have been added to Extended Data Fig. 3 (the new panel j) and Extended Data Fig. 6 (the new panel h).

6. *Movie 1: Prp8 domains are not labeled in a manner consistent with the text (Prp8Large is not noted - do you mean RT/En domain?).*

We changed the text to match the labeling in Movie 1. PRP8 Large consists of RT/En plus the HB domain.

7. *Movies 1-2. This reviewer is not really seeing evidence for Prp4K moving away from SF3b, and toward Prp6 such that a specific new interface is being made during the structural transition. Indeed, in the movies, it just looks like there is some rotation of the surrounding components while Prp4K fails to make any clear new close interface with a different surface.*

We now include an additional movie (the new Movie 3) that focusses on PRP4K and its repositioning. We also have changed the description in the text such that we now state that “PRP4K and SF3B3 are located farther apart in pre-B^{5'ss} complexes compared to pre-B”, instead of stating that PRP4K has moved away from SF3B3.

8. *Prp4K: Relevant to the above comment: If the authors do not observe specific interfaces involving Prp4K, it does not seem appropriate for them to make specific comments about kinase activation and mechanism later in the text. To that end, if the authors have a better view of Prp4K interfaces and it would strengthen the paper if they were provided as a figure.*

In the right expanded panels of Extended Data Fig. 6c and d in the original manuscript, a comparison of pre-B and pre-B^{5'ss} reveals a more extensive interface between PRP4K and PRP6 in pre-B^{5'ss}. We have tried to make this more evident by including a new panel that shows an overlay of this region, where PRP6 and PRP4K are shown as space filling models (now Extended Data Fig. 6e). We also switched the order of panels c and d in the revised version of the manuscript, such that pre-B comes before pre-B^{5'ss}. Although we clearly see an enhanced contact between PRP4K and PRP6 (one of PRPK4's substrates) in pre-B^{5'ss} complexes, we have softened somewhat our statements in the text that this leads to kinase activation.

9. *There is a typo on page 11, line 2up.*

We changed “BBR2” to “BRR2”.

10. *As a follow-up in future work, how can the complexity of the BRR2-B-specific protein network be sorted out mechanistically? The paper makes it clear that this network is very important, but it's not evident how it can be untangled. Any insights into this would be helpful.*

If it is possible to overexpress and purify full-length versions of the B-specific proteins one could add one or more of them (perhaps in a different temporal order) to purified pre-B complexes in order to try to dissect the specific roles of each protein during the pre-B to B transition.

11. Figure 4: At least for this reader, this figure does not clearly convey how the B-specific proteins are involved in any specific way. It is a complicated figure, and yet it does not really communicate the ideas that the authors seem to be stating, nor does it draw attention to specific players amidst the alphabet soup of proteins. A new type of graphic here would be helpful.

In Fig. 4 we compare the molecular architecture of pre-B^{5'ssLNG+ATP} with the B-like complex in order to illustrate which structural rearrangements do not occur in the absence of the B-specific proteins. Only those proteins relevant at this stage are depicted in the Figure. We have tried to improve this figure by exchanging the complicated cartoon depictions that show simultaneously the roles of most of the B-specific proteins at this stage, with two smaller panels that focus on specific roles of subsets of the B-specific proteins. To save space, we have moved these less-complex cartoons to Extended Data Fig. 8 (the new panels j and k). The roles of the B-specific proteins are also depicted in other figures (e.g., see ED Figs. 4b,c and 8 l-n).

12. Final sentence of the Results section, regarding the handoff, is unclear and should be rewritten for the general reader.

We have changed this sentence and added additional sentences at the end of the Results with the goal of making this part more readily understandable for a general reader.

13. The latter part of the discussion on alternative splicing and splice site choice was outstanding. Having stated these ideas, however, is it now possible to implicate specific B-specific proteins and begin testing their roles in splice site differentiation?

The B-specific proteins are core splicing factors that we and others clearly show play important roles in constitutive pre-mRNA splicing. Our data do not directly implicate the B-specific proteins in regulating alternative splicing events and thus we have not specifically mentioned them in this context. However, as our data indicate that splice site pairing occurs at the B complex stage, proteins that play a role in B complex formation (which include the B-specific proteins) are clearly regulatory candidates. Indeed, RNA-mediated knockdowns of several B-specific proteins have revealed that they modulate multiple alternative splicing events in the cell. For example, Papasaikas et al (Mol. Cell, 2015), showed that knockdown of SMU1 and RED lead to alternative splice site usage and exon skipping. Likewise, the C. elegans homologue of MFAP1 was shown to affect alternative splicing (Ma et al., 2012, PLOS Genet). Due to the lack of space, we have chosen not to discuss the roles of the B-specific proteins in alternative splicing in the main text. However, we now briefly mention this in the legend to Extended Data Fig. 4 (panel c).

Referee #2

1) *The 30 years-old concept of exon definition refers to the mutual stabilization of early splicing factors interacting with the 3' and 5' splice sites across an exonic sequence, e.g. manifested in the observation that mutations in either of these splice sites lead to skipping of the exon rather than retention of the intron harboring the mutation (e.g. Berget et al 1995 PMID: 7852296). These effects are particularly noticeable in alternatively spliced exons rather than in constitutive exons, which has been associated to weaker splice sites flanking regulatable exons (e.g. Xiao et al 2007 PMID: 17998536). Even for alternative exons, the extent to which exon definition effects play a role in their recognition varies (e.g. Buratti et al 2006 PMID: 16855287; De Conti et al 2013 PMID: 23044818). The authors often refer to exon definition in their manuscript, but it is not shown that the exon and flanking splice sites used as the substrate in this study is, strictly speaking, exon defined. It would be important, in my opinion, to show that complex A formation is enhanced by the presence of the downstream 5' splice site (by comparing A complex formation on the substrate with or without 5' splice site). Even if this is not the case, the results of Zhang, Kumar et al remain of great interest (e.g. these effects may be happening across internal exons regardless of whether they are regulated or not), but it would be helpful to clarify whether the complexes described support the classical definition of exon definition or they reflect a more general phenomenon across all kinds of internal exons.*

The MINX exon RNA substrate used in this study was previously used in studies by our lab analysing the formation of cross-exon complexes in HeLa nuclear extracts (see Schneider et al., 2010, Mol. Cell). In the latter paper, it was shown that cross-exon A-like complex formation on the MINX exon RNA was strongly reduced when the 5'ss was mutated, demonstrating that A complex formation is indeed enhanced by the presence of the downstream 5'ss. Furthermore, depletion of either U1 or U2 from the nuclear extract also led in each case to substantial reduction in the formation of the cross-exon A-like complex (Schneider et al., 2010). Thus, the complexes formed on the exon-containing substrate used in this study are indeed exon-defined. We now mention the latter on p. 3 of the main text and include a more detailed description of these previously reported results in the legend to Extended Data Fig. 1.

2) *Related to the previous point, I did not manage to find a description of the sequences of the MINX Exon RNA substrate, and whether this substrate corresponds to a natural viral exon and its natural flanking splice sites, or whether a 5' splice site (which one?) was engineered into a piece of Adenovirus Major Late promoter pre-mRNA. This may be relevant because, if the exon has been engineered, it may not recapitulate the architecture of regulatory elements (e.g. splicing enhancers) that help in exon recognition. Once again, these considerations do not detract from the value of the findings of the paper, but can help to place them in a physiological context.*

As previously described in Schneider et al., 2010, the MINX exon RNA used in this paper was derived from the MINX pre-mRNA, which is a derivative of the Adenovirus Major Late (ADML) pre-mRNA. It was initially generated by Susan Berget's lab (Zillmann et al. 1988). In the MINX pre-mRNA, the 3' exon consists of a truncated version of the ADML exon 2 (TPL2). To generate the MINX exon RNA, the 5' exon and adjacent downstream intron nucleotides of the MINX pre-mRNA were deleted, leaving the 3' exon and 64 nts of the 3' end of the upstream, adjacent ADML intron, which contains an anchoring site followed by the branch site, polypyrimidine tract and 3'ss AG. A 5'ss was introduced at the 5' end of the truncated exon, by adding the last 6 nts of the "wildtype" ADML exon 2 plus 22 nts of the adjacent, downstream intron, that is followed by a short linker and three RNA stem-loops that bind the MS2 protein. The resulting MINX exon RNA, thus consists of a contiguous piece of the ADML transcript with an internally truncated ADML exon 2 flanked by wildtype ADML intronic sequences. The exon is 39 nts and not 40 nts as we reported in the original version of the manuscript. We apologize for this mistake (the G at the 5' end of the intron was erroneously counted as part of the exon) and have corrected this error in the revised manuscript. We now include a more detailed description of the MINX exon RNA substrate used in our studies in the legend to Extended Data Figure 1.

3) Yet another related point, the length of the exon in the MINX Exon substrate is 40 nucleotides, which is below or near the lower limit observed for sustaining exon definition effects (Hawkins 1988 PMID: 3057449; De Conti et al 2013 PMID: 23044818) and in fact close to the size of microexons whose inclusion requires a special arrangement of cis-acting sequences and trans-acting factors (Irimia et al 2014; PMID: 25525873). It would be relevant to find whether exon definition effects (see point 1 above) are influenced by the length of the exon. I am not -of course- requesting a repetition of the extensive structural analyses on a longer exon substrate, but rather suggesting that this is an important point to clarify because the vast majority of internal exons are longer than the substrate used in this study. It is in the best interest of the authors to rule out that their complexes can only form on exons of a length shorter than the majority of internal exons, even in mammalian cells.

As described above in comment #1 above, the substrate used in this study indeed undergoes exon definition, despite the fact that the exon is 39 nts in length. Furthermore, we previously showed in Schneider et al., 2010 that CE pre-B complexes form on other single exon RNA substrates, including one with a substantially longer exon, namely the c-src exon 4 (i.e., BSEx4, Sharma et al 2008), which is 99 nts in length.

4) P4: "protein crosslinking coupled with mass spectrometry (CXMS) (Extended Data Fig. 2f and Supplementary Table 2) indicates that U1 and U1-related proteins communicate indirectly with U2 components, mainly via SR proteins that interact with the U2AF1/2 proteins and SF3B1 that are located near the 3' end of the intron, consistent with the idea that U2 and U1 do not directly

interact with one another.” It would be good to discuss this in the context of previous interactions proposed to mediate exon definition, e.g. Plaschka et al, 2018 PMID: 29995849; Sharma et al, 2014 PMID: 25403181; Abovich and Rosbash, 1997 PMID: 9150140; Becerra et al 2016 PMID: 26494226).

In the legend to Extended Data Fig. 2 where we describe the cross-exon protein interactions, we now mention the findings of Plaschka et al. 2018 and Sharma et al., 2014 which describe interactions involving PRP40 and LUC7, and an interaction between U1 snRNA SL 4 and the U2 SF3A1 protein, respectively. Due to the lack of space, we cannot include this information in the main text. Abovich and Rosbash, 1997 PMID 9150140, describe proteins interacting with the homolog of human SF1 in the yeast commitment complex. As SF1 no longer appears to be present in the CE pre-B complexes, we would argue that the interactions described in Abovich and Rosbash, 1997, are not relevant for our cross exon bridging model and thus we do not mention them. Becerra et al., is a review about PRP40, and instead of citing it, we directly cite papers showing PRP40 interactions with other yeast U1 proteins in early splicing complexes that are consistent with our crosslinks.

5) Figure 6b: the scheme is nice but the coordinated action of the two protomers (Figure 1a, section on pages 12-14) is not at all evident in the diagram. As a matter of fact, the diagram seems to suggest that interactions between U2 and tri-snRNP occur at the 3' end of the intron, without any involvement of 5' splice site recognition across the exon.

Figure 6b depicts the switch from a cross-exon defined to cross-intron defined, monomers. The cross-exon communication between U1 and U2, which is mediated by SR proteins, is indeed depicted in the figure (they are indicated by arrows in the cartoon at the very left). As we mention below (Referee #3, comment 5), dimer formation is not a prerequisite for the CE pre-B complex to interact with an upstream 5'ss that is present in cis (i.e., within the same pre-mRNA substrate), and it is most likely that within the cell the CE to CI switch predominantly occurs in monomeric spliceosomes. We thus do not show the dimers in Fig. 6b.

6) The description of the complex dynamic changes in composition and contacts involved in the different conformational transitions is rigorous and well supported by the data, but I believe it will be challenging to follow for the general reader. I realize that the authors have already made an important effort to highlight the key messages of the manuscript, but an extra effort towards making the manuscript accessible to non-specialists will increase its impact.

Due to the strict limitations on the length of a Nature article it is not feasible to provide extended, more simplistic explanations for all of the new aspects presented in our manuscript. Indeed, this is a very complex story that we have tried very hard to present in a manner “digestible” for non-specialists despite the space limitations imposed by Nature. However, as suggested by Referee #1,

we have changed the last part of the Results section with the goal of making this more readily understandable for a general reader.

Referee #3

1. As a general comment, the authors do not present any data on how prevalent the observed dimers are and also to what extent they persist through the various biochemical chases. E.g. if they start with isolated CE pre-B dimers and incubate these with the 5'SS oligo and ATP or just with ATP, does that change at all the proportion of dimers present in the sample? A comprehensive analysis of the relative distribution of dimers vs. monomers in the various samples would be very useful and provide further support for the authors' argument. Presumably this should be easily extracted from the gradient profiles and the cryo-EM initial 2D classes.

In this study we have focused intentionally on the structural analysis of dimers, as our recent structural studies of human B complex dimers (Zhang et al., EMBO J, 2024) showed that dimerization leads to significantly improved resolution of the protomers. We performed cryo-EM solely with complexes in fractions of the fastest sedimenting peak (typically fractions 14-18), and here we did not detect any monomers. As shown in the EM sorting schemes in the Extended Data figures, except for “bad” particles, we observe predominantly type 1 or type 2 dimers. It is not clear from our gradient analyses to what extent ATP and/or 5'ss oligo addition affects the proportion of dimers, as this would require a rather time consuming cryo-EM analysis of multiple gradient fractions for each of these conditions. We would argue that additional information about the dimers and how ATP or 5'ss oligo addition affects the percentage of dimers/monomers, while potentially interesting, is beyond the scope of this article and would have no impact on the general conclusions that we already make from the large amount of data that we have at hand.

2. Relating to the argument on p.4 – can the authors exclude the possibility that their dimers represent complexes in which the substrate substrate is bound to U1 and U2 within the same dimer rather than in different promoters as their model suggests? i.e., is the supposed 3D arrangement of U1 snRNP from previous structures incompatible with an intra-protomer path? a simple test of this model would be that RNase-H cleavage of the exon, or of a region in the PY tract that might be accessible even if engaged with SF3B1, should dissociate the dimers or result in dimers with a wider range of different orientations, if their model is correct. Minimally, the authors should outline the full proposed path of the substrate and present in a figure whether the available length of the exon between the 5'SS and 3'SS is consistent with the distances observed in their model – it is possible that this exercise might indeed exclude the alternative intra-protomer possibility due to distance constraints, but it would be important to clarify this in a figure.

With the goal of providing additional evidence concerning the molecular architecture of the pre-B dimers and also the location of the U1 snRNP within each protomer, at an earlier stage of this project we performed some initial RNase H digestions with purified pre-B complexes and different DNA oligonucleotides complimentary to the 5' end of the MINX exon (i.e. spanning or

close to the 3'ss). Unfortunately, the conditions used for digestion led to severe dissociation of the protomers and thus subsequent cryo-EM analyses were not possible. Even if it were possible to efficiently cleave the RNA without totally dissociating the protomers due to the harsher digestion conditions required (i.e., incubation at 37°C), RNase H digestions would require an extensive cryo-EM analysis of complexes in several regions of the gradient. Thus, we have chosen not to attempt any additional RNase H digestions with any of the other complexes that in some cases might be less labile.

Concerning the organization of our protomers and whether the length of the exon between the 5'ss and 3'ss is consistent with the distances observed in our model, we now show the distance between where the PPT exits the U2 SF3B1 HEAT domain and where U1 should approximately be located in the adjacent GD or where PRP28, which is directly adjacent to U1 based on the pre-B^{AMPPNP} structure, is located. In Extended Data Figure 2d, we showed that there is continuous density that links the U2 SF3B1 of one protomer to the exon/U1 snRNP-containing globular density (GD) that is attached to the PRP28 of the other protomer. Following this density, the distance between SF3B1 of one protomer and PRP28 of the other protomer would require minimally ca 30 nts of RNA to cover (see Figure below). A similar RNA length would be required to reach the U1 bound to the 5'ss in the adjacent GD. Thus, our placement of U1 is consistent with the 39 nt length of the MINX exon plus ca 10 nts of the PPT/3'ss that extend beyond the SF3B1 HEAT domain.

In the alternative protomer organization, the same U2 snRNP, but instead the U1 from the other GD, would bind to the same MINX Exon RNA substrate (and thus in this case U1 would interact *in cis* with PRP28). This would require that after exiting the SF3B1 HEAT domain, the remaining

PPT/3'ss nts plus the MINX exon would wrap back across the HEAT domain and extend to PRP28 (and to the adjacent U1) within the same protomer. However, this would require a much longer distance between the PPT and downstream 5'ss than that afforded by our substrate. Based on the pre-B 3D model, more than 60 nts would be required to cover this distance, without clashing with any tri-snRNP proteins (see Figure above), which is much longer than the length of our exon (39 nts) plus ca 10 nts of the PPT/3'ss that extend beyond the HEAT domain. Alternative RNA paths that extend along the other side of the complex would be even longer. It needs to be noted that these 60 nts are calculated assuming that the RNA is stretched and thus that RNA-binding proteins do not at all compact the RNA. Therefore, in reality, an exon much longer than 60 nts is probably required to form such an intra-protomer complex. Formation of the latter could potentially lead to circular RNA formation and, indeed, circular RNA that is formed in this manner requires longer exons. That is, Li et al., showed that in yeast, circular RNA formation is abolished when the exon is shortened to 63 nts (Li et al., Nature 2019). Thus, taken together, we are confident that the organization of the tri-snRNP, U2 and U1 in the pre-B protomers is indeed as we have described it in our manuscript. We now include this figure and a description of the exon RNA length requirements for each of the alternative protomer organizations in Extended Data Fig. 2e.

3. Related to p. 5 – the predicted parallel arrangement of the B-like CE complexes suggests a massive rotation from the pre-B state, for which the authors don't provide any clear figure or structural explanation; it is partly this arrangement in B-like complexes that led me to consider the alternative model where the substrate 5'SS and 3'SS are present in the same protomer rather than across protomers in pre-B CE complexes, since the authors' models implies the 5'SS and 3'SS are also situated in the same protomer in the B-like CE complexes. It is in my view hard to imagine how the 5'SS oligo in trans could lead to a B complex where dimers have the exons in the GDI shown in ED Fig. 3h. The authors do not provide any assessment of the heterogeneity of the protomer arrangement observed in the dimers following conversion from pre-B to B. Do all pre-B dimers observed have an anti-parallel arrangement and do all resulting B dimers have a parallel arrangement? Or did the authors only process this category of particles?

It is indeed not clear how the anti-parallel pre-B complex dimers might be converted to a B-like dimer in which the protomers are now aligned in a parallel manner. One possible explanation is that, upon addition of both the 5'ss oligo and ATP (which triggers a major conformational change within the protomers) and leads to displacement of U1 from the 5'ss, the pre-B dimers first partially dissociate, which would allow rotation of one of the protomers such that both protomers are now organized in a parallel manner. Although this is, in principle, pure speculation, we briefly mentioned a potential mechanism whereby an anti-parallel orientation is converted to a parallel one, in the context of describing the RNA network in the pre-B^{ATP} complex, in the legend to Extended Data Fig. 9i in the original manuscript. As shown in the EM sorting schemes, in those gradient fractions that we analysed by cryo-EM, essentially all pre-B and B-like complexes, except for “bad” particles, are organized as dimers. As mentioned above, we would argue that additional

information about the dimers would have no impact on the general conclusions that we can already make based on the current data presented in our manuscript.

4. Related to p13 – The model for Prp28 transfer in trans is compelling. However, the rationale for excluding the possibility that the observed events are intramolecular isn't entirely convincing in my view; the authors mention structural constraints on Prp28 on p. 12 but they don't show a clear figure explaining this. Additionally, they don't provide any clear biochemical test, e.g. trying to cleave the substrate Minx in the pre-BAMPPNP sample in the region between the BP and 5'SS should disrupt the dimers.

As described in detail in comment #2, structural constraints rule out the possibility that PRP28 is acting on the U1 bound to the 5'ss within the same protomer (i.e., in an intramolecular or cis manner). A new panel that has been added to Extended Figure 2e shows that the MINX Exon RNA is simply too short to form a protomer where U1 and PRP28 interact with one another *in cis* instead of *in trans* as we have concluded. We now refer to this figure where we mention in the main text that structural constraints rule out that PRP28 is acting *in cis*.

5. The authors do not comment at all on whether they believe the observed dimers are necessary for the proposed model of CE to CI conversion. Is the implication that most exons would have such a CE A-like or pre-B complex formed during alternative splicing?

We do not believe that dimer formation is a prerequisite for the conversion of a cross-exon assembled spliceosome to a cross-intron assembled one. As depicted in Figure 6b, the CE to CI conversion should be possible with a monomeric spliceosome if the tri-snRNP of the CE pre-B complex interacts across an intron with U1 bound to an upstream 5'ss. We now explicitly state in the main text on p. 13 that dimer formation is not a prerequisite for the CE to CI switch.

Minor

6. On p. 8 it is unclear what is different about the sample preparation in the case of pre-B5'SS complexes that leads to only a single oligo being bound? Is it the absence of the nuclear extract? If so, does it imply that B complex proteins are important for formation of the extended U6/5'SS when the pairing is limited, as it would be in the native extract? This is implied later on, but the authors should clarify this point when initially describing the pre-B chases.

This is indeed due to the absence of nuclear extract and thus the absence of the B-specific proteins. We now mention this in the text on p.8.

We thank the referees for their very constructive criticisms. We hope our manuscript is now suitable for publication in Nature.

We realize that our manuscript likely exceeds Nature's limit of 8 printed pages, but to what extent is currently not clear to us. In the event that our manuscript is accepted by Nature we will make every effort to shorten the manuscript as much as possible. However, given the large amount of data presented in our manuscript, we would hope that you would consider allowing the printed version to exceed 8 pages.

Sincerely,

Reinhard Lührmann
Holger Stark

Reviewer Reports on the First Revision:

Referees' comments:

Referee #1 (Remarks to the Author):

The authors did an excellent job answering all of this reviewer's questions. I think this outstanding work should now be published without further modification.

Referee #2 (Remarks to the Author):

The authors have clarified the issues raised in my previous report and I am happy to recommend the manuscript for publication in Nature. It will have a significant impact in the RNA processing field.

Referee #3 (Remarks to the Author):

I commend the authors for trying to answer all points raised during the review process in a constructive manner.

I believe my previous concerns have been sufficiently addressed for the purposes of this manuscript. I appreciate that a more detailed analysis of dimer formation is beyond the scope of the current focus of the manuscript and will likely be more informative for other more "native" substrates in future work.

I think the manuscript should be a landmark paper for understanding coupling of 5'SS selection with spliceosome activation. Further shortening seems essentially impossible without losing readability, so I encourage Nature to publish it in its current form.